# AMPK induces degradation of the transcriptional repressor PROX1 impairing branched amino acid metabolism and tumourigenesis

Yanan Wang [1,10], Mengjun Luo[2], Fan Wang[3], Yu Tong[3], Linfeng Li[3], Yu Shu[3], Ke Qiao[2], Lei Zhang[4], Guoquan Yan[4], Jing Liu [2], Hongbin Ji [5], Youhua Xie [2,6] ✉, Yonglong Zhang [7] ✉, Wei-Qiang Gao [3,8] ✉ & Yanfeng Liu [3,9,10] ✉

Tumour cell metabolic plasticity is essential for tumour progression and therapeutic responses, yet the underlying mechanisms remain poorly understood. Here, we identify Prospero-related homeobox 1 (PROX1) as a crucial factor for tumour metabolic plasticity. Notably, PROX1 is reduced by glucose starvation or AMP-activated protein kinase (AMPK) activation and is elevated in liver kinase B1 (LKB1)-deficient tumours. Furthermore, the Ser79 phosphorylation of PROX1 by AMPK enhances the recruitment of CUL4-DDB1 ubiquitin ligase to promote PROX1 degradation. Downregulation of PROX1 activates branched-chain amino acids (BCAA) degradation through mediating epigenetic modifications and inhibits mammalian target-of-rapamycin (mTOR) signalling. Importantly, PROX1 deficiency or Ser79 phosphorylation in liver tumour shows therapeutic resistance to metformin. Clinically, the AMPK-PROX1 axis in human cancers is important for patient clinical outcomes. Collectively, our results demonstrate that deficiency of the LKB1-AMPK axis in cancers reactivates PROX1 to sustain intracellular BCAA pools, resulting in enhanced mTOR signalling, and facilitating tumourigenesis and aggressiveness.

Tumour cells are frequently exposed to environmental conditions, such as energy stresses and therapeutic drugs, throughout tumour initiation and progression[1]. Tumour cells engage multiple strategies to tailor metabolic programmes, allowing the flexibility of a broad range of metabolic pathways. This event is termed metabolic plasticity. It plays a critical role in supporting high energy demands and facilitates the fitness and survival of tumour cells in response to low nutrient availability[2,3]. Thus, metabolic plasticity is an emerging hallmark of human cancers, that dictates tumour transformation, progression and therapeutic response.

[1]Department of Laboratory Medicine, Renji Hospital, School of Medicine, Shanghai Jiao Tong University, Shanghai, China. [2]Key Laboratory of Medical Molecular Virology (MOE & MOH), Institute of Biomedical Sciences, Shanghai Medical College, Fudan University, Shanghai, China. [3]State Key Laboratory of Oncogenes and Related Genes, Renji-Med-X Clinical Stem Cell Research Center, Ren Ji Hospital, School of Medicine, Shanghai Jiao Tong University, Shanghai, China. [4]Institute of Biomedical Sciences, Shanghai Medical College, Fudan University, Shanghai, China. [5]Shanghai Institute of Biochemistry and Cell Biology, Chinese Academy of Sciences, University of Chinese Academy of Sciences, Shanghai, China. [6]Children's Hospital, Shanghai Medical College, Fudan University, Shanghai, China. [7]Central Laboratory, Shanghai Jiao Tong University Affiliated Sixth People's Hospital, Shanghai, China. [8]School of Biomedical Engineering & Med-X Research Institute, Shanghai Jiao Tong University, Shanghai, China. [9]Department of Liver Surgery, Ren Ji Hospital, School of Medicine, Shanghai Jiao Tong University, Shanghai, China. [10]These authors contributed equally: Yanan Wang, Yanfeng Liu. ✉e-mail: yhxie@fudan.edu.cn; yonglongzhang@126.com; weiqgao@yahoo.com; lyf7858188@163.com

The energy sensor AMP-activated protein kinase (AMPK) is regarded as a master regulator that orchestrates the fine-tuning of cellular metabolism upon nutrient fluctuations. AMPK is a hetero-trimeric complex consisting of an α catalytic subunit and β and γ regulatory subunits[4]. Upon energetic stress, AMPK is activated by the phosphorylation of Thr172 in the α subunit[5] through an incompletely defined mechanism involving an upstream kinase of LKB1[6–8] or CaMKKβ[9,10] to redirect cellular metabolism. In particular, LKB1 is one of the leading somatically mutated genes in multiple cancer types including lung cancer[11]. By phosphorylating AMPK and other substrates, LKB1 is regarded as a major tumour suppressor and regulator of metabolism that acts mainly by inhibiting the mammalian target-of-rapamycin (mTOR) pathway[12–14]. Emerging evidence has revealed that the LKB1-AMPK axis potently restricts tumourigenesis and dictates metabolic plasticity and therapeutic resistance[15,16]. These findings demonstrate that the therapeutic manipulation of this pathway has implications for novel intervention strategies against cancers. Despite improved understanding, the underlying mechanisms by which LKB1-AMPK controls cellular metabolic plasticity remain largely unknown.

Branched-chain amino acids (BCAAs), including valine (Val), leucine (Leu) and isoleucine (Ile), are essential amino acids. BCAAs are important nutrition signals and play a crucial role in the activation of the mTOR pathway[17]. In particular, emerging evidence has indicated that altered gene expression in BCAA metabolism is observed in various cancer types and correlated with tumour aggressiveness and therapeutic resistance[18–20]. However, it remains unclear how BCAA metabolism alterations in cancers are regulated, particularly in contexts where nutrients are deficient. Revealing these mechanisms could facilitate the discovery of potential therapeutic targets against human cancers.

Prospero-related homeobox 1 (PROX1) is an essential lineage-specific transcription corepressor regulator for the development of multiple organs and tissues, including the liver[21–27]. Homozygous Prox1 knockout mice are embryonic lethal owing to multiple developmental defects. Emerging evidence has indicated that PROX1 expression is important for patient clinical outcomes in various cancers[28–32]. We have previously found that high levels of PROX1 in hepatocellular carcinoma (HCC) promote tumour growth and metastasis, and show poor prognosis and a high recurrence rate in HCC[30,33]. However, the mechanism by which PROX1 is abnormally expressed remains poorly understood. The relationship between PROX1 and cancer development is complex, and the functional role and mechanism of PROX1 in tumour progression are largely unknown.

In this study, we revealed that PROX1 serves as an important factor of tumour metabolic plasticity downstream of the LKB1-AMPK axis that coordinates tumour cell anabolism during insufficient nutrient availability. Mechanistically, we revealed that AMPK phosphorylates PROX1. This decreases BCAA levels and suppresses mTOR signalling activity and overall energy metabolism. Thus, therapeutic resistance to metformin or phenformin develops. This mechanism links LKB1-AMPK to BCAA metabolism and provides therapeutic implications for cancer treatment.

## Results

### PROX1 is a crucial factor for AMPK-mediated tumour metabolic plasticity

Tumour metabolic plasticity represents a leading challenge for cancer therapy. To dissect the molecular determinants governing metabolic plasticity, we performed TMT (tandem mass tag) labelling and quantitative proteomics in Huh7 cells exposed to metabolic stresses, such as glucose deprivation (Fig. 1a and Supplementary Fig. 1a). Consequently, several pathways including TGF-β, terpenoid backbone biosynthesis, steroid biosynthesis, pyrimidine metabolism and AMPK signalling were enriched, as analysed using KEGG (Fig. 1b). Interestingly, PROX1 expression was reduced by glucose starvation and was

ranked as the top enriched candidate (Fig. 1c and Supplementary Fig. 1b, c). Therefore, it was selected for further investigation. In support of this, higher levels of PROX1 were observed in KRAS-driven LKB1-mutant (KL), the upstream kinase for AMPK, lung tumours, compared with either lung cancer subtypes of KRAS (K) or concomitant KRAS and TP53 mutation (KP) from TCGA database (Fig. 1d). This unique expression pattern in both AMPK-proficient and deficient contexts suggests that PROX1 might be an important factor for tumour metabolic plasticity. Indeed, the PROX1 protein abundance was reduced in response to glucose deficiency, metformin or MK8722 (a selective on-target synthetic AMPK activator) treatment in liver cancer cells and mouse embryonic fibroblasts (MEFs) (Fig. 1e–g and Supplementary Fig. 1d–g). We further confirmed this observation in vivo. Consistently, PROX1 was decreased in the liver sections by fasting or metformin treatment, as revealed by IHC and western blot analysis (Fig. 1h, i and Supplementary Fig. 1h). To address whether the alteration of PROX1 was involved in tumour metabolic flexibility, we examined the survival capacity of HCC cells following metabolic stress. As expected, PROX1 depletion attenuated tumour cell death upon glucose starvation (Fig. 1j, k and Supplementary Fig. 1i–k). More importantly, we further found that concomitant loss of Ampkα and Prox1 largely rescued the apoptosis signature observed in Ampkα deficient MEFs (Fig. 1l and Supplementary Fig. 1l). These findings suggest the important role of PROX1 in AMPK-mediated metabolic plasticity under energy stress and the AMPK likely signals through other factors that also contribute to cell survival.

Given the role of PROX1 in AMPK-mediated cell survival during glucose starvation, we reasoned that AMPK might regulate PROX1 expression. As expected, PROX1 expression was upregulated in AMPKα knockdown HepG2 cells (Fig. 1m). Furthermore, only wild-type AMPKα2 (WT-AMPK) and constitutively active AMPK (CA-AMPK), but not the dominant-negative mutant (Dn-AMPK), were able to reduce PROX1 protein levels (Fig. 1n, o). These findings indicate that the kinase activity of AMPK is indispensable for controlling the steady-state expression of PROX1. Additionally, we used MEFs with homozygous knockout (KO) of AMPKα to further examine these results. Consistently, an upregulation of PROX1 expression was found in AMPKα KO MEFs (Fig. 1p). More importantly, we found that the glucose deprivation-mediated reduction in PROX1 expression was abrogated in AMPKα KO MEFs (Fig. 1p), indicating that the reduction in PROX1 expression during glucose starvation depends on AMPK. Furthermore, we verified the above observation in de novo tumour tissues derived from Kras^G12D, Kras^G12D; Lkb1^flox/flox and Kras^G12D; Ampkα^-/- (KA) mice in which AMPKα was deleted via lenti-Cre infection by nasal inhalation. IHC analysis showed an upregulation of PROX1 expression following the genetic loss of either Lkb1 or Ampkα (Fig. 1q, r and Supplementary Fig. 1m). Additionally, AMPKα2 dose-dependently reduced PROX1 expression in a manner that appeared to occur posttranslationally (Supplementary Figs. 1n, 2a, b). This was further supported by the observation that glucose limitation suppressed the protein stability of PROX1 (Supplementary Fig. 2c–e). Collectively, these findings demonstrate that the downregulation of PROX1 expression by AMPK is an important event for tumour metabolic adaptations during nutrient deficiency.

### PROX1 is directly phosphorylated by AMPK at Ser79

Given that AMPK kinase activity is required for the control of PROX1 protein abundance, we reasoned that AMPK might directly phosphorylate and decrease the expression of PROX1. To this end, using immunoprecipitation (IP)-based mass spectrometry, we identified the phosphorylation of Ser79 in the amino terminus of PROX1 as the conserved AMPK substrate motif site (Fig. 2a, b and Supplementary Fig. 2f), suggesting that PROX1 might be a potential substrate of AMPK. Since AMPKα2 has been shown to be preferentially expressed in the nucleus[34], where PROX1 is also localized[35], we examined whether

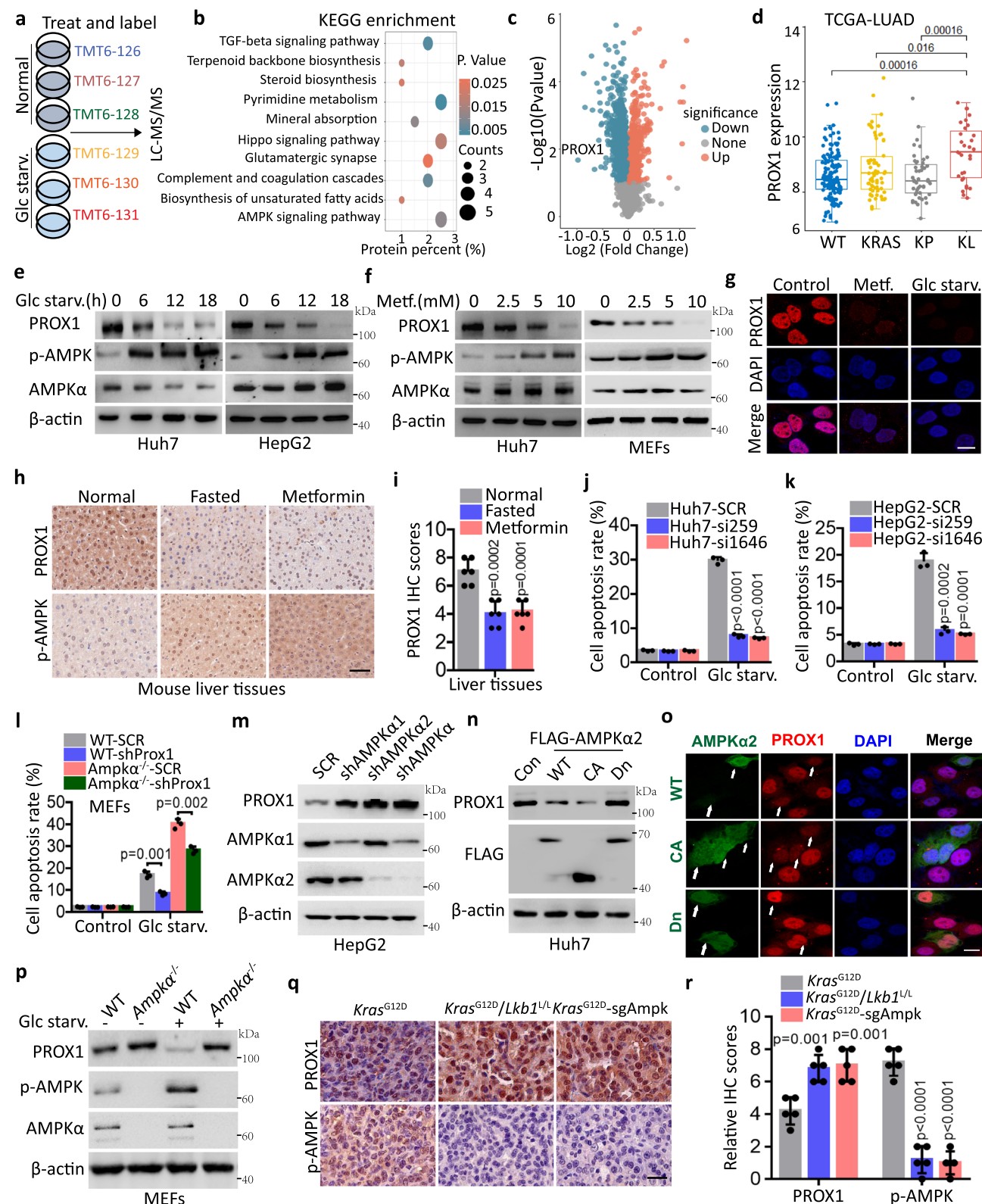

PROX1 can directly interact with AMPKα2. GST-AMPKα2 was expressed and purified in Escherichia coli. Full-length PROX1 was produced via in vitro translation. GST pulldown assays indicated that PROX1 directly interacted with AMPKα2 (Fig. 2c). Moreover, endogenous AMPKα2 colocalized with PROX1 in Huh7 cells (Fig. 2d). In addition, glucose starvation facilitated PROX1-AMPKα2 binding (Fig. 2e).

To determine the regions mediating this interaction, we prepared constructs encoding various GST-tagged truncated PROX1 proteins to test the interaction with AMPKα2 individually. The N-terminal region spanning residues 1 to 337 of PROX1 (P1) was required for the interaction with AMPKα2 (Fig. 2f, g). Additionally, FLAG-tagged PROX1 fragments were produced in HEK293T cells to examine its interaction with GST-AMPKα2. In support of the observed finding, the N-terminus of PROX1 was also responsible for the interaction with GST-AMPKα2 (Supplementary Fig. 2g). Reciprocally, the N-terminal kinase domain (aa 1–312) of AMPKα2 mainly mediated the association with PROX1

**Fig. 1 | PROX1 functions as a feedback mechanism mediates tumour cell metabolic plasticity upon glucose starvation. a, b** TMT6 labelled quantitative proteomics (**a**) to investigate the mechanisms for tumour cell metabolic plasticity and KEGG pathway enrichment analysis (**b**) the differential protein in the Huh7 cells upon glucose starvation (Glc starv.) for 12 h. **c** Volcano plot shows the total proteins upon glucose starvation by LC-MS/MS. **d** Comparison of the PROX1 mRNA level between KRAS/TP53/LKB1 wild-type (WT), single KRAS mutation (KRAS), KRAS/TP53 both mutation and KRAS/LKB1both mutation in the lung adenocarcinoma (LUAD) from TCGA database. WT/KRAS/KP/KL: $n = 138/66/47/31$, maximum = 11.54/ 12.69/11.74/11.57, upper quartile = 8.95/9.11/8.76/10.28, median = 8.06/8.36/8.10/ 9.32, lower quartile = 7.63/7.64/7.47/8.17, minimum = 6.16/6.75/6.31/7.24. **e, f** Western blot analysis the cell lysates upon glucose starvation (**e**) and metformin treatment (**f**) as indicated. **g** Immunofluorescence analysis the Huh7 cells as indicated ($n = 3$ independent experiments). Scale bar, 10 μm. **h, i** Liver tissues from normal, fasted and metformin (500 mg/L) treatment mice ($n = 6$) are subjected to immunohistochemistry (**h**) and the corresponding quantified graph of liver tissues

(**i**) are shown. Scale bar, 50 μm. **j, k** The apoptotic analysis of the Huh7 (**j**) and HepG2 (**k**) cells were infected with the lentivirus either expressing *PROX1* siRNA (si259 or si1646) precursor or scrambled siRNA precursor (SCR) by flow cytometry ($n = 3$ independent experiments). **l** The apoptotic analysis of the wild-type (WT) and AMPKα knockout (Ampkα$^{-/-}$) MEFs were infected as indicated ($n = 3$). **m** Immunoblot analysis the cell lysates as indicated. **n** Western blot analysis the cell lysates overexpression of the wild-type AMPKα2 (WT-AMPK), the constitutively active AMPK (CA-AMPK) and the dominant-negative AMPK (Dn-AMPK). **o** Representative confocal images of Huh7 cells transfected with WT-, CA- and Dn-AMPKα2 plasmids ($n = 3$ independent experiments). Scale bar, 10 μm. **p** Western blot analysis the cell lysates as indicated. **q, r** Representative IHC staining images (**q**) and statistical data (**r**) in the murine lung tumour tissues from *Kras*$^{G12D}$/*Lkb1*$^{L/L}$ and *Kras*$^{G12D}$-sgAmpk mouse ($n = 5$). Scale bar, 50 μm. The immunoblots are repeated independently with similar results at three times. For **i–l** and **r**, data represent the mean ± SD. Statistical significance was assessed using two-tailed unpaired Student's *t*-test. Source data are provided as a Source Data file.

(Supplementary Fig. 2h). We next sought to examine whether AMPK-PROX1 binding stimulated PROX1 phosphorylation. Using an in vitro kinase assay, we found the N-termini of PROX1, but not other fragments, were specifically recognized by the phosphor-AMPK substrate motif antibody. The replacement of S79 with Ala (A) abolished the phosphorylation signal (Supplementary Fig. 2i), implying that a phosphorylation event occurs in this region. Using an antibody that specifically recognizes PROX1-Ser79 phosphorylation (Supplementary Fig. 2j, k), we found that the purified N-termini of PROX1 (P1-WT) but not PROX1 (P1-S79A) from bacteria were efficiently phosphorylated by the purified active AMPK complex (α2/β1/γ1). This finding suggests a direct role for AMPK in phosphorylating PROX1 at Ser79 (Fig. 2h). Consistent with this finding, either the phosphor-deficient PROX1-S79A (Ser79Ala) or the phosphor-mimetic PROX1-S79E (Ser79Glu) compromised its interaction with AMPKα2 compared to the wild-type PROX1-WT (Fig. 2i).

In addition, PROX1, but not the S79A mutant, underwent rapid Ser79 phosphorylation following glucose deficiency or when the wild-type AMPKα2 (WT-AMPK) and constitutively active AMPK (CA-AMPK) were expressed compared to dominant-negative AMPKα2 (DN-AMPK) (Fig. 2j, k). However, the levels of Ser79 phosphorylation were diminished in AMPKα-depleted Huh7 cells (Fig. 2l). Additionally, we used MEFs with homozygous knockout (KO) of AMPKα to further examine these results. Consistently, a reduction in PROX1-Ser79 phosphorylation was found in the AMPKα KO MEFs (Fig. 2m). More importantly, we also found that glucose deprivation-mediated PROX1-Ser79 phosphorylation was abrogated in AMPKα KO MEFs (Fig. 2m). A similar finding was also observed in tissue sections of KL and KA mice relative to those of K mice (Fig. 2n). Consistently, the PROX1-Ser79 phosphorylation level was increased in the liver sections after fasting or metformin treatment, as revealed by IHC analysis (Fig. 2o). Taken together, these results indicate that AMPK directly phosphorylates PROX1 at Ser79 in vitro and in vivo.

### Ser79 phosphorylation destabilizes PROX1 in a CUL4-DDB1 E3 ligase-dependent manner

As described above, AMPK was found to interact with and phosphorylate PROX1 to destabilize its protein level, so we next sought to address how PROX1 phosphorylation is linked to altered protein degradation. Given the rapid degradation due to glucose starvation, we assumed that PROX1 might be degraded through ubiquitination and the proteasome pathway. Indeed, treatment with MG-132, a proteasome inhibitor, reversed PROX1 degradation caused by glucose deprivation (Figs. 2p, q), implying a potential role of the proteasome pathway. To provide further evidence that links AMPK-dependent Ser79 phosphorylation and the proteolytic destruction of PROX1, coimmunoprecipitation (Co-IP)/mass spectrometry (MS) was conducted to identify potential E3 ligase(s) that target PROX1 for

degradation during glucose starvation in the presence of MG132. Intriguingly, several members of the CUL4-DDB1 E3 ubiquitin ligase complex (CUL4A, CUL4B and DDB1) were identified as PROX1 binding partners (Supplementary Fig. 3a, b). Next, we systematically examined a possible interaction between the cullin family members (CUL1–5) and PROX1 and found that PROX1 exhibited specific binding to CUL4A and CUL4B (Fig. 3a). In agreement with this observation, only the dominant-negative mutants of CUL4A and CUL4B (dnCUL4A and dnCUL4A) could upregulate PROX1 protein levels in Huh7 cells (Fig. 3b). Moreover, we further demonstrated that PROX1 interacts with CUL4A, CUL4B and DDB1, but not with CUL1, CUL2, CUL3 and CUL5 (Fig. 3c and Supplementary Fig. 3c), and these associations with PROX1 were enhanced following glucose starvation (Fig. 3d). These findings suggest that PROX1 indeed interacts with the CUL4-DDB1 E3 ubiquitin complex and responds to metabolic stresses. In addition, PROX1 colocalized with DDB1 in Huh7 cells, as evidenced by immunofluorescence staining (Fig. 3e). Next, we determined the specific regions required for this interaction. FLAG-PROX1 fragments were cotransfected with HA-CUL4A, HA-CUL4B and DDB1 in HEK293T cells. The results of the Co-IP assays indicated that the N-terminus of PROX1 was responsible for the interactions with CUL4A, CUL4B and DDB1 (Supplementary Fig. 3d–f). Furthermore, GST pulldown assays showed that the phosphomimetic PROX1-S79E bound to DDB1 with a stronger intensity than that of the wild-type PROX1. However, the phosphordeficient PROX1-S79A attenuated its association with DDB1 (Fig. 3f). Collectively, these results demonstrate that the Ser79 phosphorylation of PROX1 is an upstream event that recruits the CUL4-DDB1 E3 ligase complex.

We next examined whether the CUL4-DDB1 E3 ligase was implicated in PROX1 degradation. As expected, we depleted the expression of CUL4A, CUL4B and DDB1 in HepG2 cells individually and discovered that PROX1 expression was upregulated (Supplementary Fig. 3g). For visible evaluation of these effects, we performed an immunofluorescence assay and revealed that PROX1 expression was reduced by overexpression of the wild-type CUL4A, CUL4B or DDB1 in Huh7 cells and was increased when the dominant-negative CUL4A or CUL4B was expressed (Fig. 3g). In support of this notion, the results indicated that the half-life of PROX1 was impaired in Huh7 cells ectopically expressing CUL4A or DDB1 (Supplementary Fig. 3h–k), whereas *CUL4B* or *DDB1* knockdown in HepG2 cells prolonged the protein turnover of PROX1 (Fig. 3h–k). In further support of this observation, we demonstrated that individual knockdown of CUL4A, CUL4B or DDB1 could reduce the ubiquitination of PROX1 (Supplementary Fig. 3l). Furthermore, we found that the ubiquitination of PROX1 was increased by glucose starvation (Supplementary Fig. 3m). Next, we determined whether Ser79 phosphorylation links PROX1 to its ubiquitination and degradation through the CUL4-DDB1 E3 ligase. The steady-state level of PROX1 was enhanced in PROX1-S79A compared to the wild-type

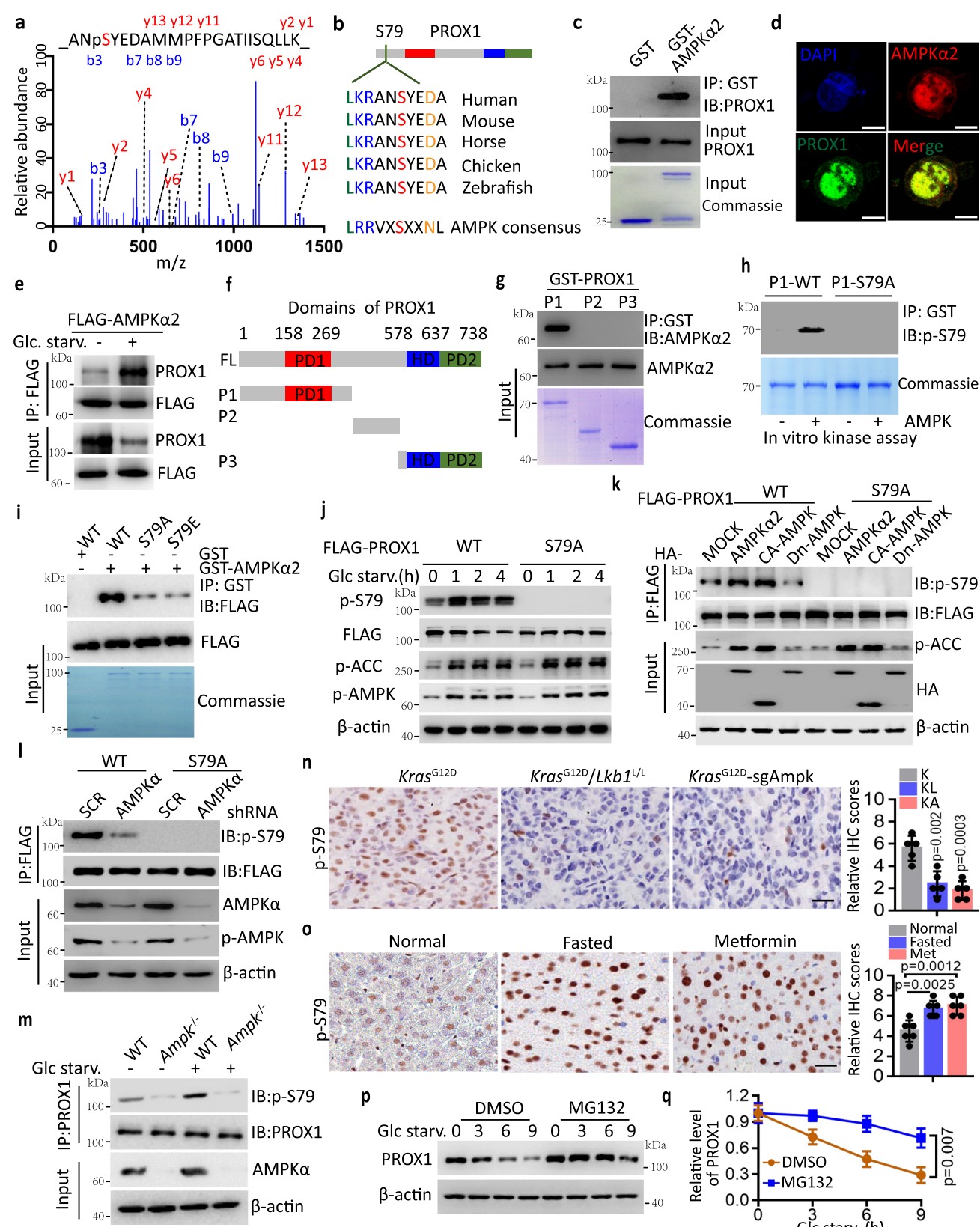

PROX1. In contrast, the PROX1-S79E mutation, which mimics the phosphorylated form of PROX1, facilitated its protein degradation (Fig. 3l, m). In further support of this observation, we demonstrated that the ubiquitination of PROX1-S79A was reduced by glucose starvation compared with wild-type PROX1 or the PROX1-S79E mutation (Fig. 3n). In addition, ectopic expression of CUL4A, CUL4B or DDB1

reduced the levels of abundance of PROX1-WT and PROX1-S79E, but not PROX1-S79A (Fig. 3o). To provide clinical relevance, we performed IHC staining to examine their expression patterns and correlation using a tissue microarray containing samples from 90 HCC individuals. A strong negative correlation between PROX1 and CUL4A, CUL4B or DDB1 levels was observed (Pearson $r = -0.52$, $r = -0.622$ and $r = -0.508$,

**Fig. 2 | AMPK directly phosphorylates PROX1 at Ser79. a** The Ser79 phosphorylation modification of PROX1 peptides identified through liquid chromatography-tandem mass spectrometry. **b** The substrate motif of AMPK kinases is shown (lower), and the Ser79 site of PROX1 is conserved in vertebrate. **c** Coomassie blue staining of GST and GST-AMPKα2 incubated with in vitro translated PROX1, PROX1 was detected by anti-PROX1 antibody after GST-pulldown. **d** Endogenous PROX1 and AMPKα2 in the Huh7 cells were visualized under fluorescent microscopy ($n = 3$ independent experiments). Nuclei were stained with DAPI. Scale bar, 10 μm. **e** Western blot analysis Huh7 cell lysates as indicated. **f** The domain organization of PROX1 and the deletion constructs. PD1, prospero domain1; HD, homeodomain; PD2, prospero domain 2. **g** Input, coomassie blue staining of each GST-PROX1 fragment incubated with in vitro translated AMPKα2. AMPKα2 was detected by anti-AMPKα2 antibody after GST-pulldown. **h** The P1-WT and P1-S79A of PROX1 as indicated was detected using a phosphor-specific antibody against Ser79 of PROX1. **i** HEK293T-expressed wide type (WT) and replacement of S79 with Ala (A) or Glu (E)

in the FLAG-PROX1 (S79A and S79E) incubated with GST-AMPKα2. **j** Western blot analysis HEK293T cell lysates as indicated. **k** Immunoblot analysis of HEK293T cell lysates transfected with the indicated PROX1 and HA-AMPKα2 plasmids. **l** Immunoblot analysis of the FLAG-IP and cell lysates from transfected with the indicated constructs. **m** Immunoblot analysis of MEFs cell lysates as indicated. **n** Representative IHC staining images and statistical data of the murine lung tumour tissues from $Kras^{G12D}$ (K), $Kras^{G12D}/Lkb1^{L/L}$ (KL) and $Kras^{G12D}$-sgAmpk (KA) mouse ($n = 5$). Scale bar, 50 μm. **o** Representative IHC staining images and statistical data of the liver tissues from normal, fasted and metformin (500 mg/L) treatment mice ($n = 6$). Scale bar, 50 μm. **p, q** Representative western blot (**p**) and the corresponding quantified graph (**q**) of Huh7 cell lysates are shown. $n = 3$ independent experiments. IB, immunoblot; IP, immunoprecipitation. The immunoblots are repeated independently with similar results at three times. For **n, o** and **q**, data represent the mean ± SD. Statistical significance was assessed using two-tailed unpaired Student's $t$-test. Source data are provided as a Source Data file.

$p < 0.01$) (Fig. 3p, q). Taken together, these results suggest that AMPK-mediated PROX1 Ser79 phosphorylation promotes its protein degradation by recruiting a CUL4-DDB1 E3 ligase complex.

## PROX1 couples epigenetic cues to suppress BCAA degradation

To understand the physiological consequences of AMPK-mediated PROX1 phosphorylation and degradation, we comparatively analysed the RNA-sequencing (RNA-seq) data of liver sections from wild-type ($Prox1^{f/f}$) and Prox1 liver specific knockout (Alb-Cre; $Prox1^{f/f}$) mice. These data were then cross-referenced to ChIP-seq data of $Prox1$ in mouse livers[36] to identify the potential signalling pathways. In total, 328 genes were identified to be commonly modulated. Intriguingly, KEGG pathway enrichment analysis revealed that valine, leucine and isoleucine degradation pathways (BCAA metabolism) were significantly enriched (Fig. 4a). Most key genes involved in this pathway were upregulated in the Prox1 liver-specific knockout (Prox1-cKO) group and PROX1-knockdown Huh7 cells (Fig. 4b and Supplementary Fig. 4a). Moreover, GSEA also identified BCAA metabolism enrichment in the Prox1-cKO mice (Fig. 4c) and PROX1-depleted Huh7 cells (Supplementary Fig. 4b). In addition, we comparatively analysed the RNA-seq data from the lung tumour tissues from $Kras^{G12D}$ (K) and $Kras^{G12D}$; $Lkb1^{flox/flox}$ (KL) mouse models[37] and found a similar result. That is, BCAA metabolism was enriched in K tumours in contrast to KL tumours that exhibited high PROX1 levels (Fig. 4d), suggesting PROX1 deficiency promoted BCAA degradation.

We then verified the RNA-seq results using real-time PCR. The genetic loss of Prox1 in mice or PROX1 ablation in Huh7 and HepG2 cells indeed increased the expression of most genes implicated in BCAA metabolism (Fig. 4e and Supplementary Fig. 4c, 4d). Moreover, the expression of certain key genes involved in BCAA metabolism (BCKDHB, ACADSB, ACADM, EHHADH, HSD17B10, ABAT, ACAA2, HADHB and HMGCS2) was decreased in AMPKα knockdown Huh7 cells (Fig. 4f). Importantly, PROX1 deficiency-mediated gene transcription of the above BCAA metabolism-related factors in Huh7 cells was enhanced by glucose starvation or in mice following fasting (Fig. 4g and Supplementary Fig. 4e). In agreement with this finding, ChIP assays using a PROX1 antibody revealed the recruitment of PROX1 at the promoter regions of selected genes implicated in BCAA metabolism in in vitro and in vivo models (Fig. 4h, i and Supplementary Fig. 4f). To clarify how BCAA metabolism-related gene transcription programmes were activated, we evaluated chromatin accessibility using ATAC-seq (Assay for Transposase-Accessible Chromatin with high throughput sequencing) in Prox1-deficient mice. ATAC-seq revealed increases in the accessibility of proximal regulatory elements in mice following PROX1 loss relative to the control group. Moreover, KEGG pathway enrichment analysis revealed that the valine, leucine and isoleucine degradation pathway (BCAA metabolism) was enriched and that the accessibility of proximal regulatory elements in numerous genes involved in BCAA metabolism was increased in the Prox1-deficient

group (Fig. 4j, k and Supplementary Fig. 4g, h). This process is accompanied by alterations in the histone markers H3K4me3, H3K9me3 and H3K27ac at multiple promoters of certain key genes involved in BCAA metabolism (Fig. 4l, m and Supplementary Figs. 4i, j, 5a–c), as evidenced by ChIP-qrtPCR in mice and HCC cell lines, respectively. We next examined the effect of PROX1 Ser79 phosphorylation on BCAA metabolism. To this end, we generated stable Huh7 cells in which endogenous PROX1 was depleted by si1646, while approximate levels of wild-type PROX1 or its S79A/S79E mutants were achieved (Supplementary Fig. 5d). Using real-time PCR and ChIP-qrtPCR assays, we found that phosphor-deficient PROX1-S79A could reduce the expression of certain key genes involved in BCAA metabolism (BCKDHB, ACADSB, ACADM and EHHADH) and increased the recruitment of H3K9me3 and PROX1 at the promoter regions of these genes compared with the phosphor-mimetic S79E mutant (Fig. 4n, o and Supplementary Fig. 5e). Collectively, these results demonstrate that PROX1 reduces BCAA degradation partially through epigenetic modifications.

We further confirmed these results in HCC cells and liver sections using western blot analysis. Consistently, the protein levels of these genes were elevated when PROX1 was depleted in Huh7 and HepG2 cells, or in mouse livers (Fig. 4p and Supplementary Fig. 5f). To further confirm their relevance in HCC tissues, we performed IHC staining to examine their expression patterns. As expected, we found that the levels of PROX1 were negatively correlated with selected genes encoding key enzymes (BCKDHB, ACADSB, ACADM, EHHADH, DLD and HMGCL) of BCAA metabolism and positively correlated with BCAT1 expression (Figs. 4q, r and Supplementary Fig. 5g). Furthermore, we also examined the relationship between p-AMPK and the above key enzymes of BCAA metabolism in HCC tissues. Consistently, we found that the levels of p-AMPK were positively correlated with the above key enzymes (BCKDHB, ACADSB, ACADM, EHHADH, DLD and HMGCL) of BCAA metabolism and negatively correlated with BCAT1 expression (Supplementary Fig. 5g and h). Moreover, systematic TCGA database interrogation indicated that high expression of BCAT1 and low expression of several selected key enzymes (BCKDHB, ACADSB, ACADM, EHHADH, DLD, HMGCL and HMGC2) in BCAA metabolism of HCC were associated with unfavourable patient survival (Supplementary Fig. 5i–p). Taken together, these observations indicate that PROX1 inhibits BCAA metabolism in relation to tumour progression and patient survival.

## PROX1 activates mTOR signalling by sustaining intracellular BCAA pools

Given that PROX1 suppresses BCAA metabolism, we then examined whether PROX1 could alter the intracellular BCAA concentrations. First, we performed targeted metabolomic analyses for amino acids and found that BCAAs, including valine, leucine, and isoleucine, were indeed decreased in mouse livers with Prox1 deletion relative to wild-

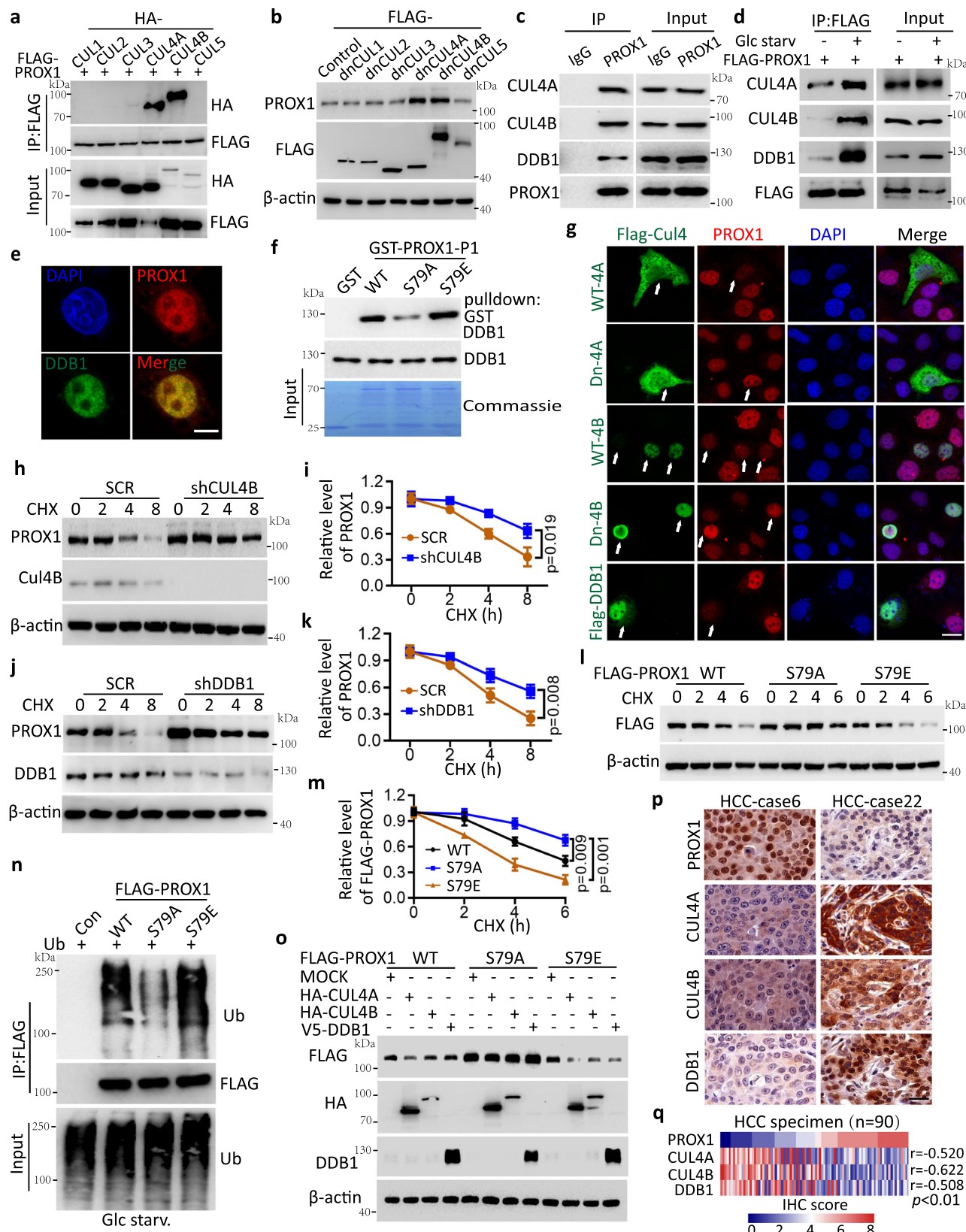

type mice (Fig. 5a). Similarly, Huh7 cells with PROX1 knockdown showed reduced BCAA concentrations compared to control cells (Fig. 5b). The overall lower levels of BCAA upon PROX1 loss in mice and HCC cells support this notion (Fig. 5c and Supplementary Fig. 6a). We next examined the effect of PROX1 Ser79 phosphorylation on BCAA metabolism. We generated stable Huh7 cells in which endogenous

PROX1 was depleted by si1646 while approximate levels of wild-type PROX1 or its S79A/S79E mutants were achieved (Supplementary Fig. 5d). Using these cell lines, we found that BCAA reduction by PROX1 depletion could be rescued by restoring the expression of wild-type PROX1 and the S79A mutant but not the phosphomimetic S79E mutant. These findings demonstrate that PROX1 phosphorylation

**Fig. 3 | Ser79 phosphorylation destabilizes PROX1 in a CUL4-DDB1 E3 ligase-dependent manner. a** Interaction between PROX1 and CUL proteins were analyzed in the HEK293T cells. IP, immunoprecipitation. **b** Immunoblot analysis of the cell lysates from Huh7 cells transfected with the indicated dominant-negative CUL (dn-CUL1, 2, 3, 4 A, 4B and 5) constructs. **c** Endogenous PROX1 in the Huh7 cells treated with MG132 (4uM) was immunoprecipitated using anti-PROX1 antibody or isotype IgG control and subjected to immunoblot analysis. **d** HepG2 cells transfected with FLAG-PROX1 were deprived of glucose for 8 h in the presence of MG132. **e** Representative confocal images from Huh7 cells ($n = 3$ independent experiments). Scale bar, 10 μm. **f** Purified recombinant GST-PROX1 (P1-WT, S79A and S79E) was incubated with DDB1 individually. **g** Representative confocal images of PROX1 expression in the Huh7 cells transfected with WT- and Dn-CUL4A, 4B constructs, and Flag-DDB1 as the indicated ($n = 3$ independent experiments). Scale bar, 20 μm. **h, i** HepG2 cells infected with shRNA lentivirus encoding scramble (SCR) and shCUL4B were treated cycloheximide (CHX, 100 mg/mL). Representative western blot (**h**) and the corresponding quantified graph (**i**) are shown ($n = 3$ independent experiments). **j, k** HepG2 cells infected with the lentivirus as indicated. Representative western blot (**j**) and the corresponding quantified graph (**k**) are shown ($n = 3$ independent experiments). **l, m** HEK293T cells transfected with FLAG-PROX1 variants upon glucose starvation were subject to CHX (100 mg/mL) treatment. Representative western blot (**l**) and the corresponding quantified graph (**m**) are shown ($n = 3$ independent experiments). **n** Ubiquitination levels of FLAG-PROX1 variants in the HEK293T cells upon glucose starvation were immunoprecipitated using anti-FLAG mAb and subjected to immunoblot analysis. **o** Immunoblot analysis of the cell lysates from HEK293T cells co-transfected plasmids as indicated. **p, q** Representative IHC staining images (**p**) and the heatmap (**q**) of IHC score (by Pearson's) between PROX1, CUL4A, CUL4B and DDB1 expression in HCC tissues ($n = 90$). Scale bar, 50 μm. The immunoblots are repeated independently with similar results at three times. For **i, k** and **m**, data represent the mean ± SD. Statistical significance was assessed using two-tailed unpaired Student's $t$-test. Source data are provided as a Source Data file.

alters its protein activity, which impairs BCAA metabolism (Fig. 5d). We then extended this investigation using isotope labelling and tracing to evaluate metabolic interconversion in Huh7 cells cultured in the presence of [$^{15}N_1,^{13}C_5$]-Gln and $^{13}C_6$-Leu, respectively. On the one hand, [$^{15}N_1,^{13}C_5$]-Gln tracing suggested that PROX1 ablation impaired the conversion of Gln to Glu (M + 6) and BCAAs (M + 1) catalysed by glutaminase (GLS) and subsequent BCAT1/2, as evidenced by lower fractional labelling (Figs. 5e and f). We further found that the phosphor-deficient PROX1-S79A could enhance the conversion of Gln to BCAAs (M + 1) compared with the phosphor-mimetic S79E mutant upon glucose starvation (Fig. 5g). On the other hand, an increased level of deaminated KIC (M + 6) was observed following PROX1 knockdown in Huh7 cells (Figs. 5h and i), while the generation of other metabolites including αKG, Ace-CoA, Citrate and succinate, was not altered significantly regardless of different labelling (Supplementary Fig. 6b). Consistently, we demonstrated that the phosphor-deficient PROX1-S79A reduced the level of KIC compared to the phosphor-mimetic S79E mutant upon glucose starvation (Fig. 5j). In addition, we further found that the knockdown of PROX1 in Huh7 cells did not alter the expression of GLS, GLS2 or SLC7A5 (Supplementary Fig. 6c). Collectively, these results demonstrate that PROX1 increases BCAA production, a potent nutrient signal that activates mTOR signalling. Therefore, we next determined whether altered BCAAs driven by PROX1 affected mTOR signalling activity. As expected, RNA-seq data showed compromised enrichment of the mTOR signature in the livers of *Alb-Cre*; *Prox1*$^{f/f}$ mice relative to *Prox1*$^{f/f}$ mice (Fig. 5k). Indeed, depletion of PROX1 in HCC cells resulted in impaired phosphorylation of ribosomal S6 kinase (S6K), the well-established substrate of mTOR, and increased the phosphorylation level of AMPK (Fig. 5l). To determine whether the observed decrease in mTOR activity was due to perturbed BCAA production, we manipulated BCAA levels in Huh7 cells through an additional treatment with leucine, the core amino acid sensed by mTORC1. Strikingly, leucine supplementation almost abrogated mTOR signalling downregulation upon PROX1 knockdown (Fig. 5m). Additionally, a similar result was found in the *Prox1*$^{f/f}$ and *Alb-Cre*; *Prox1*$^{f/f}$ mice fed a diet with or without high BCAA contents (Fig. 5n), indicating that altered BCAA metabolism by PROX1 is required to activate mTOR activity.

Next, we examined the expression of PROX1 and p-S6K in HCC tissues ($n = 90$). As expected, the levels of PROX1 and p-S6K in HCC tissues were positively correlated (Fig. 5o). Consistent with the existence of the AMPK-PROX1 axis, our findings indicate that the S79E mutant compromised the PROX1 function for mTOR signalling activation compared with the wild-type PROX1 and S79A mutant. (Fig. 5p and q). Since mTOR is a master regulator of anabolic processes, we then examined whether activated mTOR signalling affects other metabolic pathways, such as glucose metabolism. In line with the observation that the expression pattern of glycolysis was diminished in *Alb-Cre*; *Prox1*$^{f/f}$ mice (Supplementary Fig. 6d), impaired glucose consumption and lactate production were observed in HCC cells with PROX1 deletion (Supplementary Fig. 6e). Furthermore, HCC cells overexpressing PROX1-S79A exhibited increased glucose metabolism compared to PROX1-WT cells. This effect was significantly impaired in HCC cells when PROX1-S79E was ectopically expressed (Supplementary Fig. 6f–6h), demonstrating that PROX1 Ser79 phosphorylation resulted in its altered ability to rewire glucose metabolism.

Thus, AMPK-mediated PROX1 phosphorylation dictates BCAA metabolism and mTOR signalling to limit glucose consumption, possibly facilitating tumour cell adaptation to metabolic stresses. To test this hypothesis, we examined cell viability under glucose deprivation by monitoring mTOR signalling activity. As expected, rapamycin, an mTOR inhibitor, considerably reversed Huh7 cell death under glucose-deprived conditions, an effect comparable to that of cells with PROX1 depletion (Supplementary Fig. 6i), and rapamycin had a similar effect in PROX1-depleted Huh7 cells with ectopic expression of PROX1 (Supplementary Fig. 6j). These findings suggest that the inactivation of mTOR by the AMPK-PROX1 axis is critical for tumour cells to establish metabolic adaptation.

## The AMPK-PROX1 axis controls tumourigenesis and therapeutic response

We next explored whether Ser79 phosphorylation affects the function of PROX1 in HCC progression. As expected, HCC cells overexpressing PROX1-S79A, but not PROX1-S79E, promoted HCC cell proliferation compared to the wild-type PROX1 group (Supplementary Fig. 7a–7c). Furthermore, we next tested the in vivo effect of the PROX1-BCAA axis during tumourigenesis and progression and found that the phenotypes of the mice bearing PROX1-deficient Huh7 cells were rescued after treatment with leucine, at least partially. Processes affected by this included the proliferative activity, p-S6K and Ki-67 activity, and overall survival times (Supplementary Fig. 7d–7f). To further confirm these results in the Prox1 liver-specific knockout mice, we used a DEN-induced liver cancer mouse model and observed that the dietary intake of high BCAA promoted liver tumourigenesis in the *Prox1*$^{f/f}$ mice and abolished the tumour incidence caused by *Alb-Cre*; *Prox1*$^{f/f}$ mice, as the tumour number and liver-body ratio did not yield a difference between *Prox1*$^{f/f}$ and *Alb-Cre*; *Prox1*$^{f/f}$ mice following BCAA treatment (Fig. 6a–c). Notably, Prox1 deficiency was correlated with the upregulated expression of key enzymes implicated in BCAA metabolism, including Bckdhb, Dld, Acadsb, Acadm, Ehhadh, Hibadh and Hmgcl (Supplementary Fig. 7g and 7h). These findings reinforce the notion that PROX1 promotes liver tumourigenesis via BCAA metabolism. To link the role of PROX1 to its phosphorylation by AMPK, we next determined whether the S79E mutant of PROX1 also rendered a disadvantage in tumour growth in the DEN-induced liver cancer mouse model. Using adeno-associated virus (AAV) serotype 8-mediated liver-specific expression of S79A/S79E and Cre in *Prox1*$^{f/f}$ mice, we found that the mice harbouring the nonphosphorylated PROX1 mutant

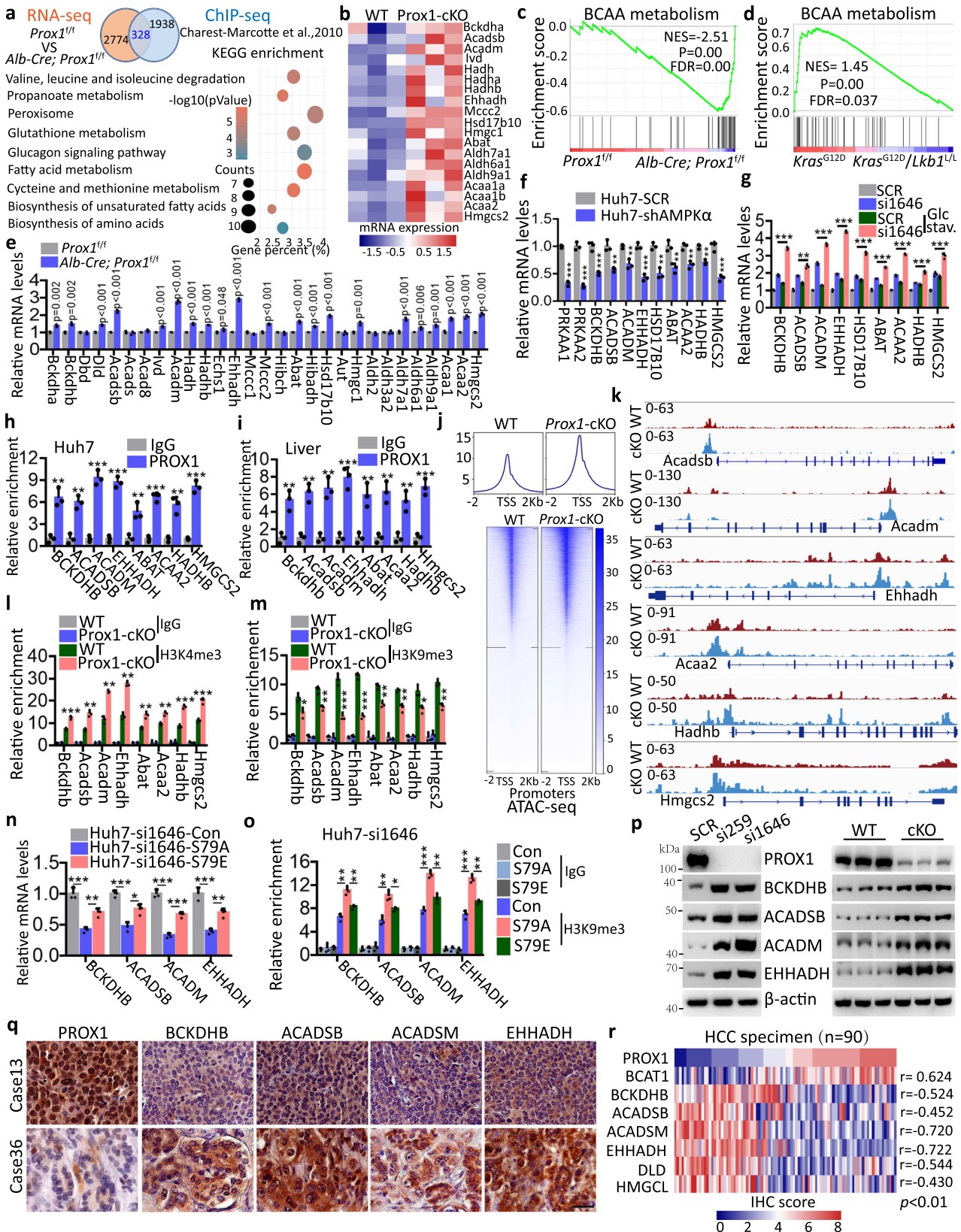

(S79A) exhibited more tumour numbers and an increased liver-body ratio compared to the control and S79E mice (Fig. 6d–f). Thus, AMPK phosphorylates PROX1 to impair its function in vivo.

Considering the potent role of PROX1 in the modulation of overall anabolism via BCAAs, we determined whether PROX1-driven tumour-igenesis establishes metabolic vulnerability that is sensitive to therapeutic drugs, such as metformin. As expected, the loss of PROX1 rendered HCC cells resistant to metformin in vitro and in vivo, and compromised the overall survival times in nude mice (Fig. 6g and Supplementary Fig. 7i–k). In agreement with these findings, IHC staining revealed that metformin treatment upregulated p-AMPK and PROX1 Ser79 phosphorylation levels, and reduced Ki-67 levels in

**Fig. 4 | PROX1 inhibits BCAA metabolism via mediating an epigenetic modification. a** Venn diagram showing the number genes in the mouse liver with PROX1 binding and displaying expression changes in Prox1 liver specific knockout mice (Alb-Cre; *Prox1*[f/f]). KEGG pathway enrichment analysis the above overlapping genes as indicated. **b** Heatmap demonstration of the gene expression related to valine, leucine and isoleucine degradation (BCAA metabolism) from WT and Prox1-cKO mice (*n* = 3). **c** GSEA shows the enrichment of BCAA metabolism in Alb-Cre; *Prox1*[f/f] mice. **d** GSEA shows the enrichment of BCAA metabolism in *Kras*[G12D] mice compared with the *Kras*[G12D]/*Lkb1*[L/L] mice. Statistical significance was assessed using Permutation test. **e** Real-time PCR analysis the relative mRNA levels of BCAA metabolism genes from *Prox1*[f/f] and Alb-Cre; *Prox1*[f/f] mice (*n* = 3). **f** Real-time PCR analysis the relative mRNA levels from Huh7 cells as indicated (*n* = 3). **g** Real-time PCR analysis the relative mRNA levels of Huh7 cells as indicated (*n* = 3). **h, i** ChIP-qrtPCR was performed with sonicated chromatins immunoprecipitated from Huh7 cells (**h**) and mouse liver tissue (**i**) by anti-PROX1 antibody or preimmune IgG (*n* = 3).

**j** Heatmap of the gene peaks by ATAC-seq from WT and Prox1-cKO mouse liver tissues. **k** Density maps for ATAC-seq in liver tissues from WT and Prox1-cKO mice. **l, m** ChIP analysis of H3K4me3 (**l**) and H3K9me3 (**m**) enrichment in mouse liver tissues as indicated (*n* = 3). **n** Real-time PCR analysis the relative mRNA levels as indicated (*n* = 3). **o** ChIP analysis of H3K9me3 enrichment at the BCAA embolism genes promoter as indicated (*n* = 3). **p** Immunoblot analysis cell lysates from Huh7 cells and liver tissues from WT and Alb-Cre; *Prox1*[f/f] (cKO) mice as indicated (*n* = 3). **q, r** Representative IHC staining images (**q**) and the heatmap of IHC score (by Pearson's) (**r**) between PROX1 and several proteins as indicated in HCC tissues (*n* = 90). Scale bar, 50 μm. The immunoblots are repeated independently with similar results at three times. *n* was biological replicates for all experiments. For **e**–**i** and **l**–**o**, data represent the mean ± SD and **P* < 0.05, ***P* < 0.01, ****P* < 0.001. Statistical significance was assessed using two-tailed unpaired Student's *t*-test. Source data are provided as a Source Data file.

PROX1 expressing tumour tissues. Despite the activation of AMPK in PROX1 knockdown xenografts, Ki-67 activity was not altered. This is possibly due to deficient PROX1 expression and Ser79 phosphorylation (Supplementary Fig. 7l and m). Consistent with this notion, therapeutic resistance was correlated with S79 phosphorylation of PROX1. Metformin also reduced the tumour weight of the PROX1-WT and PROX1-S79A groups. However, no significant difference was found in the PROX1-S79E group (Fig. 6h and Supplementary Fig. 7n), which had an effect comparable to that of PROX1 depletion. These findings suggest that PROX1 blunts the therapeutic response of HCC cells to metformin, which is dependent on AMPK.

Given that LKB1, an upstream kinase of AMPK, is the leading mutated gene in lung adenocarcinoma, we sought to further confirm the above results using the LKB1-deficient KRAS-driven lung cancer model (KL). We first verified whether Ser79 phosphorylation influenced the lung tumourigenesis of PROX1 in KL mice. Of note, the lentivirus-mediated delivery of PROX1-S79A in the lung through nasal inhalation, resulted in a larger tumour number and increased tumour burden relative to the vector control and S79E mice (Fig. 6i–l). In agreement with these findings, IHC staining revealed that the PROX1-S79A group showed a high level of p-S6K and the low expression of key enzymes implicated in BCAA metabolism (Bckdhb, Acadsb and Ehhadh) compared with the vector control and S79E groups (Fig. 6m and Supplementary Fig. 8a). Due to metabolic vulnerability, phenformin has been identified to selectively target LKB1-deficient KRAS-driven lung tumours[16], in which PROX1 is markedly elevated. We therefore further investigated whether PROX1 rewired the therapeutic response of KL tumour to phenformin. Indeed, KL tumours were sensitive to phenformin administration, and decreases in tumour burden, tumour numbers and the percentage of Ki-67 positive cells were observed. This effect was largely compromised in *Prox1*-deficient KL tumours showing resistance to phenformin therapy (Fig. 6n–r). As expected, the levels of key enzymes implicated in BCAA metabolism were increased, and the level of p-S6K was reduced in KL tumours following Prox1 deletion compared to those derived from KL tumours (Supplementary Fig. 8b, c). The low expression of several enzymes, such as BCKDHB, ACADSB and HMGSC2, was correlated with worse patient survival in lung adenocarcinomas (Supplementary Fig. 8d–f). In summary, there results demonstrate that the AMPK-PROX1 axis controls tumourigenesis and the therapeutic response in HCC and lung cancers.

**The AMPK-PROX1 axis is important for patient outcomes**

To determine the importance of AMPK-PROX1 for clinical patient outcomes, we extended our investigation to analyse the expression pattern and associations of the AMPK-PROX1 axis in cancer specimens. We examined the expression of p-AMPK, PROX1 and Ser79 phosphorylation in HCC (*n* = 90) and lung adenocarcinoma (*n* = 90). Based on immunostaining and grading, PROX1 expression was conversely correlated with p-AMPK levels in HCC specimens. High expression of

PROX1 predicted unfavourable prognosis, and high expression of p-AMPK predicted a better prognosis in patients with HCC (Supplementary Fig. 9a–d). Strikingly, the phosphorylation levels of AMPK and PROX1 showed a positive correlation in HCC tissues (Fig. 7a, b), and higher Ser79 phosphorylation levels were associated with a better clinical outcome in HCC patients (Fig. 7c). Consistently, p-AMPK and Ser79 phosphorylation were observed to be reduced in cancerous tissues relative to adjacent tissues of HCC patients, and higher Ser79 phosphorylation was correlated with low tumour stage in HCC (Supplementary Fig. 9e–i). HCC patients stratified by the signature of p-AMPK[high]/S79p[high] predicted the best clinical outcome, while patients with p-AMPK[low]/S79p[low] showed the shortest survival times (Fig. 7d). Similar results were found in human lung adenocarcinoma (LUAD). A higher p-AMPK level was positively associated with Ser79 phosphorylation and was negatively associated with to PROX1 expression (Figs. 7e, f and Supplementary Fig. 9j, k). Moreover, the overall survival times and disease-free survival appeared to be shorter in lung cancer patients with higher PROX1 and were prolonged in cases of higher Ser79 phosphorylation (Fig. 7g–j). Intriguingly, the expression patterns of p-AMPK[high]/S79p[high] and p-AMPK[low]/S79p[low] showed maximum performance in stratifying patients into favourable and unfavourable prognoses, and predicted tumour relapse (Fig. 7k and l). Taken together, these results suggest that the AMPK-PROX1 axis serves as an important molecular switch that modulates BCAA metabolism and mTOR signalling and, dictates overall anabolism and tumourigenesis (Fig. 7m). Thus, our data suggest that Ser79 phosphorylation serves as a biomarker for the prediction of patient prognosis.

## Discussion

Metabolic plasticity has been recognized as an emerging hallmark of human cancers that can establish metabolic adaptations that favour tumour survival in the face of fluctuating stressful conditions, such as energy stresses and therapeutic drugs[1]. Despite the central role of AMPK in metabolic plasticity, the underlying mechanisms by which AMPK orchestrates this process remain largely unknown. In this study, we clearly demonstrated that PROX1 is an important factor for tumour metabolic plasticity based on quantitative proteomics and biochemical and functional analyses. First, PROX1 is reduced in multiple organs and cancer cells in response to metabolic stress or AMPK activation. Second, an increased PROX1 expression pattern specifically occurs in *KRAS*-driven *LKB1*-mutant (KL) lung tumours, but not KRAS alone or *KRAS*-driven *TP53*-mutant (KP) tumours. Consistent with this finding, loss of PROX1 largely rescues cell death in *Ampkα*-deficient MEFs upon glucose starvation. Third, glucose deprivation induces the rapid degradation of PROX1 by AMPK by directly phosphorylating its Ser79. Fourth, PROX1 deficiency or Ser79 phosphorylation renders tumour cells metabolically invulnerable and therapeutically resistant to metformin. Finally, genetic ablation of PROX1 renders LKB1-deficient KRAS-driven lung cancer resistant to phenformin treatment.

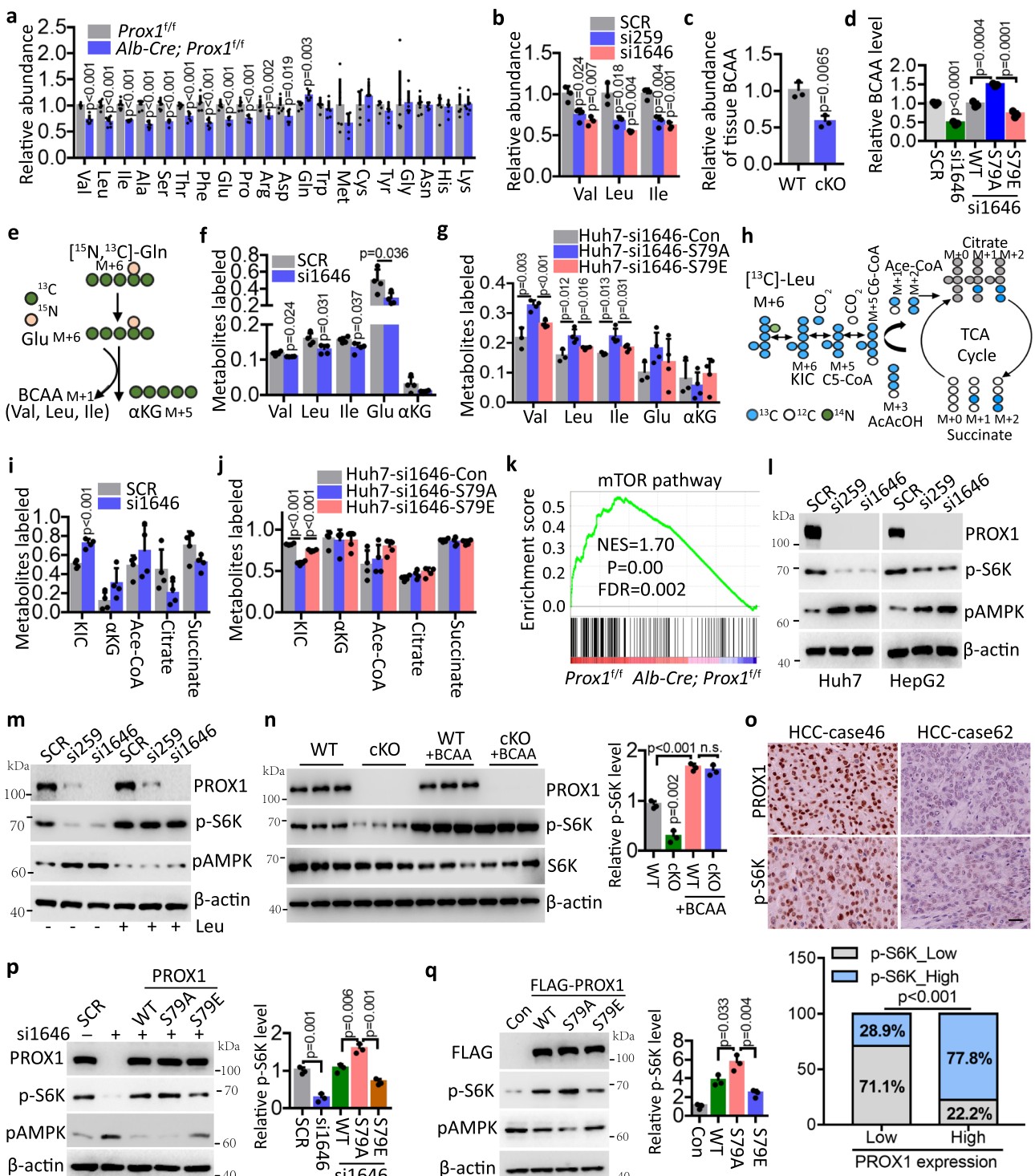

Therefore, we identified PROX1 as a critical factor for the LKB1-AMPK axis that mediates tumour metabolic plasticity to direct the therapeutic response to metformin.

Another important finding of the current study the functional link between the LKB1-AMPK axis and BCAA metabolism in governing tumour metabolic plasticity. LKB1 deficiency or mutations contribute to oncogenesis, and this effect mainly depends on its mediation mTOR signalling activation via multiple characterized mechanisms[12–14]. Although it is well known that BCAAs can directly activate mTOR signalling[17], whether LKB1 deficiency plays a role in BCAA metabolism remains unclear. In this study, by integrated analyses of multiple omics datasets, we found a metabolic signature of impaired BCAA

metabolism in *Lkb1*-deficient *Kras*-driven lung tumours relative to *Kras* mutant tumours. The loss of LKB1 results in abrogated AMPK and hyperactivation of mTOR, which is potently activated by BCAAs[17]. Thus, we propose that somatically acquired mutations of LKB1 are a driver of the metabolic shift to reprogramme BCAA metabolism and to sustain TOR activation. This model is strongly supported by findings in the current study that indicate that AMPK depletion represses BCAA-related genes and the clinical relevance of AMPK with BCAA gene expression in specimens of HCC. First, the present experiments reveal the molecular mechanisms by which the LKB1-AMPK axis mediates BCAA metabolism. Prox1 deletion in HCC and lung tumours increases the expression of the key enzymes of BCAA metabolism in part by

**Fig. 5 | PROX1 activates mTOR pathway via sustaining intracellular BCAA pools. a** Relative abundance of amino acids by LC/MS in liver tissues of *Prox1*^f/f (WT) and Alb-Cre; *Prox1*^f/f (Prox1-cKO) mice (*n* = 6). **b** Relative abundance of valine, leucine and isoleucine by LC/MS in Huh7 cells infected with the lentivirus either expressing *PROX1* siRNA (si259 or si1646) precursor or scrambled siRNA precursor (SCR) (*n* = 3). **c** Relative abundance of BCAA in the liver tissues as indicated (*n* = 3). **d** Relative abundance of BCAA in the Huh7 cells as indicated (*n* = 3). **e, f** Schematic of isotope tracing in Huh7 cells were traced 24 h with [$^{15}$N, $^{13}$C]-Gln (**e**), followed by LC/MS analysis of the labelled metabolites (**f**) (*n* = 4). **g** LC/MS analysis of the labelled metabolites in Huh7 cells upon glucose starvation were traced 24 h with [$^{15}$N, $^{13}$C]-Gln (*n* = 4). **h, i** Schematic of isotope tracing in Huh7 cells were traced 24 h with [$^{13}$C]-Leu (**h**), followed by LC/MS analysis of the labelled metabolites (**i**) (*n* = 4). **j** LC/MS analysis of the labelled metabolites in Huh7 cells as indicated upon glucose starvation were traced 24 h with [$^{13}$C]-Leu (*n* = 4). **k** GSEA shows the enrichment of

mTOR pathway in the *Prox1*^f/f mice. Statistical significance was assessed using Permutation test. **l** Immunoblot analysis of the cell lysates as indicated. **m** Immunoblot analysis of Huh7 cells with or without Leu (200 μM) as indicated. **n** Immunoblot analysis of the lysates from the liver tissues with or without BCAA (200%) as indicated, and the relative p-S6K level are shown (*n* = 3). **o** Representative IHC staining images (upper) and statistical data (down) of p-S6K and PROX1 expression in HCC tissues (*n* = 90). Scale bar, 50 μm. **p** Immunoblot analysis of Huh7 cell lysates as indicated and the relative p-S6K level are shown (*n* = 3). **q** Immunoblot analysis of SMMC-7721 cell lysates as indicated and the relative p-S6K level are shown (*n* = 3). The immunoblots are repeated independently with similar results at three times. n was biological replicates for all experiments. For **a–d**, **f–g**, **i–j**, **n** and **p–q**, data represent the mean ± SD. Statistical significance was assessed using two-tailed unpaired Student's *t*-test. Source data are provided as a Source Data file.

mediating epigenetic modifications. Second, PROX1 increases the intracellular BCAA concentrations in an AMPK-dependent manner. Third, LKB1-AMPK axis-mediated PROX1 phosphorylation results in BCAA degradation, which suppresses mTOR signalling activity and facilitates tumour cell adaptation to metabolic stresses. Other studies also support our findings, as these studies reported that the inhibition of mTOR signalling facilitates the adaptation of tumour cells to unfavourable conditions[38,39]. Together, these observations underline the context-dependent regulation of metabolic drivers that mediate an unexpected role to facilitate tumour survival during metabolic stresses[1,38–42]. Thus, the current work establishes a crucial molecular connection between LKB1 and BCAA metabolism and that the important link is PROX1. PROX1 can be phosphorylated to exert its function on AMPK to switch off mTOR signalling. These findings suggest an alternative mechanism for this process, in addition to the well-documented interaction between the AMPK and mTOR pathways via twin mechanisms involving the direct phosphorylation of mTORC1 binding partners, such TSC2 and Raptor, by AMPK[13,43]. Indeed, AMPK can inactivate mTORC1 independently of directly targeting TSC2 and Raptor[44]. It has been demonstrated that RAGA/B deficiency in GTPase activity largely abrogates mTORC1 inhibition during glucose starvation, but AMPK is still activated[44,45]. Therefore, our work has enhanced the understanding of the molecular switch of AMPK/mTOR, and supports the notion that AMPK both directly and indirectly interacts with the mTOR signalling cascade during nutrient shortage, at least partially.

Consistent with the role of PROX1 in the activation of mTOR signalling via BCAA metabolism, PROX1 indeed alters the expression of numerous genes in the BCAA metabolic pathway. Our findings indicate that PROX1 acts to increase intracellular BCAA levels by inhibiting BCAA catabolism activity but also by increasing BCAA synthesis. The glutamine level increased, the incorporation of labelled glutamine into BCAAs decreased, and increased KIC from labelled leucine upon PROX1 deficiency. Thus, it is reasonable that loss of PROX1 both, increases the expression of genes implicated in BCAA degradation and results in the downregulation of certain genes, such as BCAT1-2, leading to BCAA synthesis suppression. These findings are in line with the bidirectional roles of BCAT1/2 that catalyzes the reversible interconversion of BCAAs[17,20]. We also examined the expression of GLS, GLS2 and SLC7A5 and ruled out the possibility of changes in glutamine uptake in Huh7 cells with PROX1 depletion. Indeed, the levels of these genes were not significantly altered, further supporting the dual role of PROX1 in BCAA metabolism. It is well documented that transcriptional activation is often associated with reduced trimethylation of H3K9 and the increased trimethylation of H3K4 and acetylation of H3[46–50]. Despite being encouraged by the altered enrichment of promoters of BCAA metabolic genes by H3K9me3, H3K4me3, and H3K27ac, we have not determined which epigenetic writer (s) or eraser (s) are potentially involved in selectively altering BCAA metabolic gene expression in the context of

PROX1 or AMPK. It has been previously reported that PROX1 is implicated in the nucleosome remodelling and deacetylase (NuRD) complex, which consists of numerous different protein subunits, including HDAC1/2 and lysine specific demethylase 1 (LSD1). PROX1 may act via the NuRD complex via direct binding to HDAC1/2 or LSD1, which was also reported in our previous works[30,51]. However, experimental work is warranted in the future to link these events. Additionally, it would be more relevant to identify PROX1 as a factor that interacts with the ERRα/PGC-1α complex to control energy homoeostasis[52]. More importantly, AMPK directly and indirectly increases PGC-1α activity to initiate the expression of several key players in mitochondrial and glucose metabolism[53–55]. Whether the phosphorylation of PROX1 by AMPK involved in this process require further confirmation in the future studies.

Even though PROX1 has been demonstrated to play an important role in the development of human cancers[30,35,56,57], whether it exerts an oncogenic or a tumour suppressor function is complex and depends on cancer types. Our current findings that PROX1 can be directly phosphorylated by AMPK to facilitate PROX1 degradation by recruiting a CUL4-DDB1 ligase complex shed an important light on the function of PROX1. However, we have not identified the substrate receptor of the CUL4-DDB1E3 ubiquitin complex for PROX1, and what part(s) is implicated in substrate recognition and ubiquitination. These questions remain to be explored in the future. Notably, PROX1 elicits a potent oncogenic function by sustaining intracellular BCAA pools. This results in enhanced mTOR signalling, which is constrained by AMPK via Ser79 phosphorylation of PROX1. In addition, the phosphorylation of PROX1, which impairs its oncogenic function, is markedly reduced in cancers, and lower Ser79 phosphorylation levels are associated with unfavourable patient survival. Thus, our study clearly illustrates that Ser79 nonphosphorylated PROX1 is an oncogenic driver in tumourigenesis and aggressiveness, at least in HCC and lung cancers.

Finally, the present work shows that PROX1-driven BCAA production is indispensable for the subsequent activation of mTOR signalling to boost overall anabolism and tumourigenesis, thereby establishing a direct regulatory mechanism that links these metabolic regulators. Consistent with this role, higher expression levels of PROX1 in HCC show therapeutic sensitivity to metformin. Similarly, *Lkb1*-deficient *Kras*-driven lung tumours with higher PROX1 are efficiently targeted by phenformin, owing to defects in the intact metabolic regulatory axis. Considering that the high expression of PROX1 imposes selective vulnerability via BCAAs, dietary BCAA limitation may be a potential intervention for treating HCC and lung cancer patients with aberrant PROX1 expression. Collectively, the deficiency of the LKB1-AMPK axis in cancers reactivates PROX1 to sustain intracellular BCAA pools and mTOR signalling, facilitating tumourigenesis and aggressiveness. Perturbating such metabolic reprogramming could be a promising therapeutic approach for the treatment of cancer.

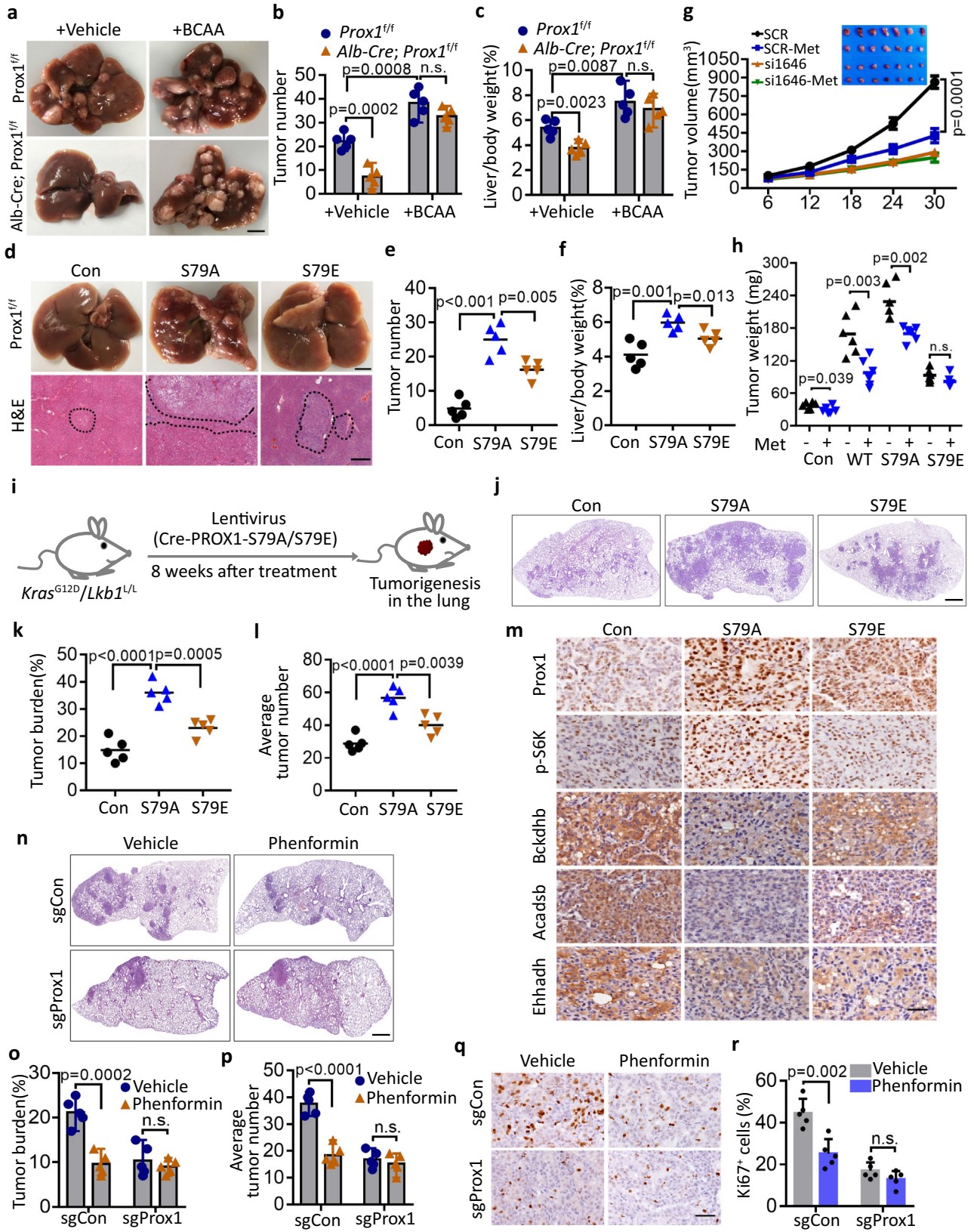

## Methods

### Plasmid constructs

Flag-tagged full-length *PROX1* and *AMPKα2* cDNAs were cloned in pcDNA3 (Invitrogen, USA) to create pFlag-Prox1 and pFlag-AMPKα2 respectively. Lentiviral vectors pLKO.1 TRC (RRID: Addgene_10878) and pWPI (RRID: Addgene_12254) were used for producing recombinant lentiviruses. For RNA interference of *PROX1*, DNA fragments encoding hairpin precursors for si259 (5'-TTTCCAGGAGCAAC-CATAATT-3', corresponding to nt259–279 of *PROX1* ORF) and si1646 (5'-GGCTCTCCTTGTCGCTCATAA-3', corresponding to nt1646–1666 of *PROX1* ORF) were inserted into pLKO.1 TRC respectively. A scrambled siRNA precursor (Scr) of similar GC-content to si259 and si1646

**Fig. 6 | AMPK-PROX1 axis impairs tumourigenesis and dictates therapeutic response. a–c** Analysis of *Prox1*[f/f] (WT) and Alb-Cre; *Prox1*[f/f] (Prox1-cKO) mice with DEN-induced liver cancer with normal or high BCAA (200%) diets. Representative images of livers (**a**) and the number of tumours (**b**), and the Liver/body weight (**c**) of the mice (*n* = 5) as indicated. Scale bar, 1 cm. **d–f** Representative images and H&E staining of livers (**d**) and the number of tumours (**e**), and the Liver/body weight (**f**) of the mice (*n* = 5) as indicated. **g** Huh7 cells stably expressing the indicated siRNAs were subcutaneously injected into in nude mice respectively with or without metformin (500 mg/L) treatment. Shown are average tumour volumes over time (*n* = 7). Data are presented as mean ± SEM. **h** SMCC-7721 cells stably overexpressing the PROX1 variants were subcutaneously injected into nude mice respectively with or without metformin (500 mg/L) treatment as indicated, and the weights of tumours are shown (*n* = 6). **i** Schematic model of lung tumourigenesis from *Kras*[G12D]/*Lkb1*[L/L] mice treated with a lenti-Cre-PROX1 virus. Eight weeks after nasal

inhalation, mice were killed and analysed. **j–k** Representative H&E image (**j**) of lung tumour are shown. Scale bar, 500 μm. The tumour burden (**k**) and average tumour numbers (**l**) were calculated and plotted (*n* = 5 mice for each group). **m** Immunohistochemistry analysis of the levels of Prox1, p-S6K, Bckdhb, Acadsb and Ehhadh in the above tissues as indicated. Scale bar, 50 μm. **n–p** Representative the H&E image (**n**) of lung tumour from *Kras*[G12D]/*Lkb1*[L/L] mice treated with a lenti-Cre-sgPROX1 and sgSCR virus as indicated with or without phenformin (1.5 g/L) treatment (*n* = 5 mice for each group). Scale bar, 500 μm. Statistical analysis of the tumour burden (**o**) and average tumour numbers (**p**) were shown.
**q, r** Representative IHC staining images (**q**) and statistical data (**r**) of Ki67 expression in the above tissues (*n* = 5) as indicated. Scale bar, 50 μm. For **b–c, e–f, h, k–l, o–p** and **r**, data represent the mean ± SD. Statistical significance was assessed using two-tailed unpaired Student's *t*-test. n.s. not significant. Source data are provided as a Source Data file.

but no sequence identity with *PROX1* was used as negative control. For overexpression of *PROX1*, Flag-tagged *PROX1* cDNA was cloned in pWPI. For overexpression of si1646-resistant *PROX1*, synonymous mutations were introduced into the target sequence of si1646 (5'-GGCTCTC<u>A</u>TT<u>A</u>TC<u>A</u>CTCATAA-3', mutations underlined) in *PROX1* cDNA. pEGFP-Prox1 was generated by inserting *PROX1* cDNA into pEGFP-C2 (Clontech). HA tagged human *Cul4A*, *Cul4B* and *DDB1* cDNAs were cloned into pcDNA3 respectively. *AMPKα2* cDNA was also inserted into pWPI and pGEX-4T1 (Amersham).

### Cell lines, transfection and lentiviruses
Human HCC cell lines Huh7 (SCSP-526), HepG2 (SCSP-510), SMCC-7721 (TCHu 52) and embryo kidney cell line HEK293T (SCSP-502) were obtained from Cell Bank of Shanghai Institutes of Biological Sciences, Chinese Academy of Sciences (Shanghai, China). These cells were maintained in Dulbecco's modified Eagle medium (DMEM, Gibco, USA) supplemented with 100 U/ml penicillin G/streptomycin sulfate and 10% (v/v) foetal bovine serum (FBS, Gibco, USA), and cultured at 37 °C with 5% $CO_2$.

Helper plasmids pSPAX2 (Addgene plasmid 12260) and pMD2.G (Addgene plasmid 12259) were co-transfected with pLKO.1- or pWPI.1-based plasmids into HEK293T cells to package recombinant lentiviruses. Supernatants from co-transfections were used for infection of cultured cells.

### Cell apoptosis assay
Huh7 and HepG2 cells were treated with or with glucose starvation for 24 h, and these cells were collected and analyzed using the Annexin V-FITC/PI apoptosis detection kit (BD Biosciences) according to manufacturer's instructions. All flow cytometry data were analyzed by Flow Jo software.

### Immunoprecipitation and mass spectrometry
HEK293T cells transfected with FLAG-PROX1 or control plasmid were lysed used the Pierce RIPA buffer (Thermo Scientific, Prod#89901). The supernatant of the cell lysate was incubated with anti-FLAG M2-agarose beads (Sigma, M8823). The anti-FLAG M2-agarose beads were washed six times with Pierce RIPA buffer. The beads were boiled in protein loading buffer, resolved on SDS-PAGE and coomassie blue stained. The mass spectrometry analysis procedures were performed as previously described[30].

### Antibodies
Primary antibodies against the following proteins were obtained from Cell Signaling Technology: p-AMPK (#2535, 1:1000), p-AMPK substrate motif (#5759, 1:1000), AMPKa (#2532, 1:2000), AMPKa2 (#2757, 1:2000), p-mTOR (#5536, 1:1000), mTOR (#2983, 1:2000), p-4EBP1 (#2855, 1:1000), p-S6K (#9205, 1:1000) and S6K (#9202, 1:2000); From Upstate: PROX1 (07-537, 1:2000); from Abcam: AMPKa1 (ab32047, 1:2000) and DDB1 (ab109027, 1:2000); from Sigma: Flag M2 (F3165,

1:5000) and β-Actin (A1978, 1:10000); from Thermo Fisher Scientific: HA (SG77) (71-5500, 1:3000); from Abclonal: p-S79 (customization, 1:1000), ACADM (A4567, 1:2000) and EHHADH (A13488, 1:2000); from Proteintech: CUL4A (14851-1-AP, 1:2000), CUL4B (12916-1-AP, 1:2000), BCKDHB (13685-1-AP, 1:2000) and ACADSB (13122-1-AP, 1:2000); from Cell Signaling Technology: anti-rabbit IgG, HRP-linked Antibody (#7074, 1:5000) and anti-mouse IgG, HRP-linked Antibody (#7076, 1:5000).

### Phospho-specific p-S79 PROX1 antibody
The phospho-specific p-S79 PROX1 antibody was prepared from a supplier (Abclonal) and generated by immunizing rabbits with the synthetic phospho-peptide, covalently cross-linked to keyhole limpet haemocyanin (KLH), and was purified through subsequent positive and negative selection using the KRAN(S-p)YED phosphor peptide and the unphosphorylated KRANSYED peptide, respectively. The phospho-specific immunoreactivities of the antibodies were detected by dot plot using both phosphorylated and nonphosphorylated peptides, and further confirmed by western bot using the cell lysates derived from the expression of PROX1 and S79A mutant constructs in HEK293T cells.

### In vitro kinase assay
Active recombinant AMPK (α2, β1, γ1) was purchased from Sigma (A1733). The recombinant GST-PROX1 fragment and variants (1–337, 335–570, 544–738 and 1-337-S79A) proteins purified from *Escherichia coli* were incubated with recombinant AMPK in a 20 μl reaction volume containing 50 mM Tris, 10 mM MgCl₂, 5 mM DTT and 200 μM ATP. The reactions were incubated at 30 °C for 1 hr and stopped by addition of protein loading buffer. The samples were subjected to western blot experiment with the phospho-AMPK substrate motif antibody and a phosphor-specific antibody against Ser79 of PROX1.

### RNA isolation and quantitative real-time PCR (qrtPCR)
Total RNA was extracted from HCC cells or mouse tissues using TRIzol Reagent (Invitrogen) according the manufacturer's instructions. The total cDNA was generated by the Prime-Script RT reagent Kit (TaKaRa, Dalian, China). The real-time PCR assay was performed using SYBR Premix Ex Taq (TaKaRa, Dalian, China) according the manufacturer's instructions. Relative mRNA levels were calculated using *β-actin* as control as previously described[30]. Primer sequences are presented in Supplementary Table 1.

### ATAC-seq analysis and chromatin immunoprecipitation
Assay for transposase-accessible chromatin with high throughput sequencing (ATAC-seq) was performed as the previously described[58]. MAnorm and integrative genomics viewer (IGV) software was used to compare ATAC-seq signal intensities and enriched peaks between *Prox1*[f/f] and *Alb-Cre*; *Prox1*[f/f] mice.

Anti-PROX1 antibody was used to immunoprecipitate sonicated chromatin prepared from Huh7 cells or mouse liver tissues.

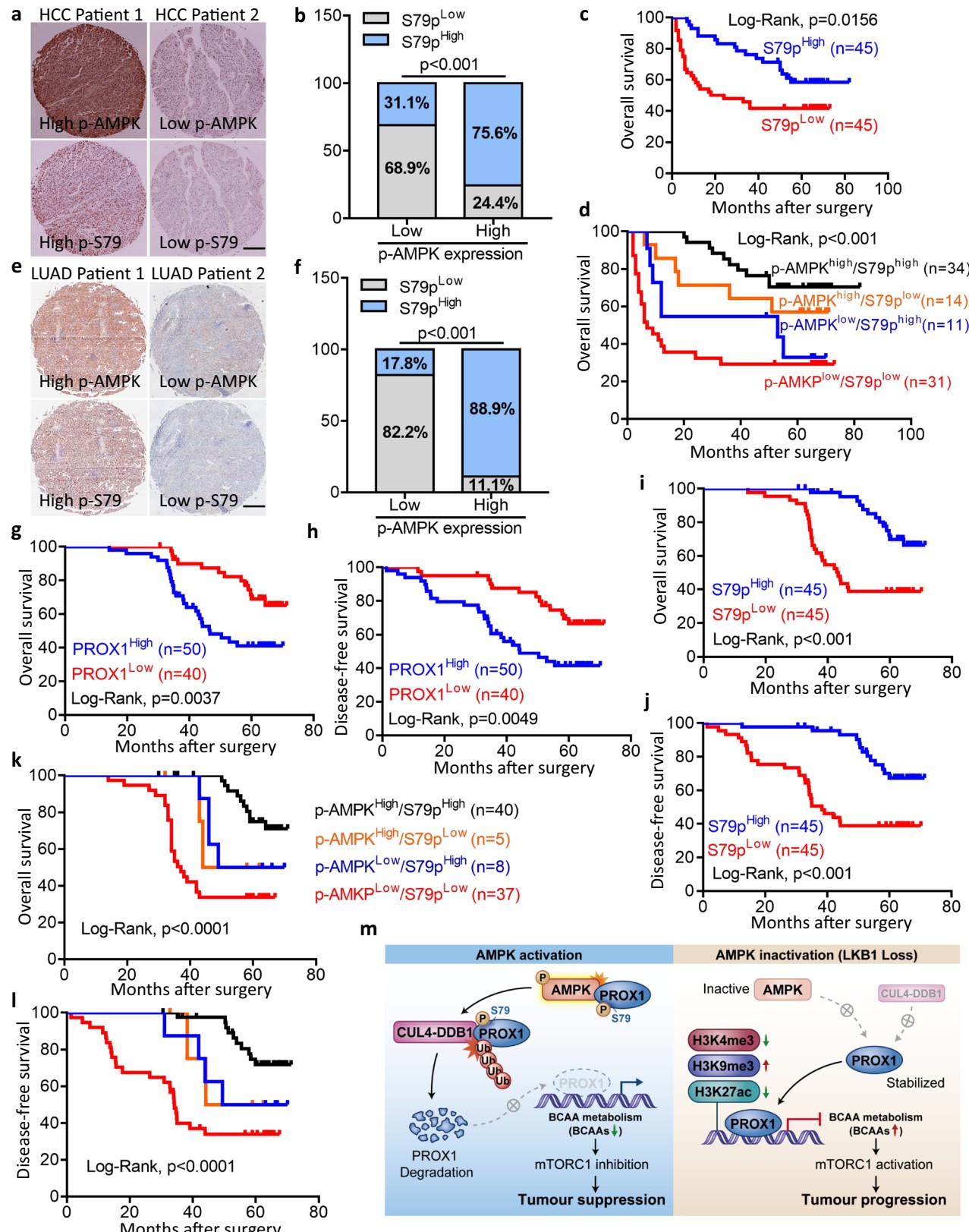

Five percent of post-sonication sample were saved as input control and pre-immnue IgG was used for specificity control. DNA extracted from precipitated chromatin were quantitated using qrtPCR. DNA extracted from saved input sample were quantitated in parallel (Ct [Input]) and results for IP by Anti-PROX1 Ab or pre-immnue IgG (Ct [IP]) were then used to calculate relative specific occupancy and non-specific background using the equation: 2^ (Ct [IP]-Ct [Input]) *100%.

### BCAA, glucose and lactate detection

BCAA concentration was detected using the BCAA kit (Sigma, MAK003) according to the manufacturer's instructions. The cells

**Fig. 7 | AMPK-PROX1 axis in human cancers is associated with patient prognosis. a, b** Representative IHC staining images (**a**) and statistical data (**b**) of p-S79 and p-AMPK expression in HCC tissues ($n = 90$). Scale bar, 500 μm. **c** Kaplan−Meier analysis of overall survival probability of p-S79 levels in HCC patients. The statistical significance was assessed using log-rank test according to HCC patients with low or high expression of p-S79. **d** Kaplan−Meier analysis of overall survival in the HCC patients ($n = 90$) according to combined expression status of p-S79 and p-AMPK. **e, f** Representative IHC staining images (**e**) and statistical data (**f**) of p-S79 and p-AMPK expression in lung adenocarcinoma (LUAD) tissues ($n = 90$). Scale bar, 500 μm. **g, h** Kaplan−Meier analysis of overall survival (**g**) and disease-free survival (**h**) probability of PROX1 levels in LUAD patients ($n = 90$). **i, j** Kaplan−Meier analysis of overall survival (**i**) and disease-free survival (**j**) probability of p-S79 levels in LUAD patients. **k, l** Kaplan−Meier analysis of overall survival (**k**) and disease-free survival (**l**) in NSCLC patients according to combined expression status of p-S79 and p-AMPK. **m** Glucose deprivation activated AMPK directly phosphorylated PROX1 at Ser79, allowing a rapid recruitment of Cul4-DDB1 E3 ubiquitin ligase complex to promote PROX1 degradation, a critical event that activates BCAA metabolism to suppress mTOR signalling pathway. Conversely, the deficient-LKB1-AMPK axis in cancers reactivates PROX1 to dictate BCAA catabolism and mTOR signalling, facilitating tumourigenesis and aggressiveness. For **b** and **f**, statistical significance was assessed using two-sided Chi-square test. Source data are provided as a Source Data file.

were seeded in the six-well plates for 24 h, the cell numbers were counted, collected and be rapidly homogenized in 100 μL of cold BCAA Assay buffer for detecting BCAA. Concentration of BCAA is calculated by the formulation: $C = S_a/S_v$ ($S_a$ = Amount of BCAA in unknown sample (nmole) from standard curve, $S_v$ = Sample volume (mL) used). For determining glucose and lactate, Huh7 and HepG2 cells were seeded in the six-well plates. After 24 h, The cell culture medium was collected. Glucose and lactate were determined using the glucose assay kit (Sigma, MAK181) and lactate assay kit (Sigma, MAK064) respectively according the manufacturer's instructions. Glucose consumption was determined by subtracting the final medium glucose concentration from the initial medium glucose concentration in the culture medium.

### Stable isotope labelling

The Huh7 cells were plated in the 10 cm dish for culture 24 h. Then, the medium of cells was replaced with fresh medium with carbon ($^{13}C_6$)-labelled leucine. After 24 h, the cells were collected, the intracellular metabolites were extracted for liquid chromatography-mass spectrometry analysis.

### Mice

*Prox1*^f/f and *Alb-Cre*; *Prox1*^f/f mice were prepared from the Shanghai Model Organisms Center (Shanghai, China). The *Prox1*^f/f and *Alb-Cre*; *Prox1*^f/f male mice at 14 day old were injected with 25 mg/kg dose of DEN to establish the liver cancer models as previously described[19]. *Kras*^G12D and *Lkb1*^L/L mice were originally generously provided by T. Jacks (Koch Institute for Integrative Cancer Research, Cambridge, MA, USA) and R. Depinho (MD Anderson Cancer Center, Houston, TX, USA) respectively, and a gift from Prof. Hongbin Ji (Shanghai Institute of Biochemistry and Cell Biology, Center for Excellence in Molecular Cell Science, Chinese Academy of Sciences, Shanghai, China) for current study. *Kras*^G12D/*Lkb1*^L/L mice were randomly group and treated with a lentivirus expressing Cre recombinase by nasal inhalation to establish tumour mouse models as previous described[15,59]. All these mouse strains were generated in a C57BL/6 background. All mice were born and maintained under pathogen-free conditions and were housed with a 12-h light/dark schedule at $25 \pm 1$ °C and were fed an autoclaved chow diet (XieTong Biology, Cat#1010013) and water ad libitum. The maximal tumour size/burden was not exceeded, all of the mouse experiments were approved by the Institutional Animal Care and Use Committee at Ren Ji Hospital affiliated to School of Medicine of Shanghai Jiao Tong University.

### Clinical samples and immunohistochemistry

This study was approved by the institutional review board at Ren Ji Hospital affiliated to School of Medicine of Shanghai Jiao Tong University. Tissue microarray of tumour samples were collected at Ren Ji Hospital and prepared from Shanghai Zuocheng Biotech (Shanghai, China) and Shanghai Outdo Biotech Co., Ltd. (Shanghai, China). The informed consent was acquired from the patients and patients' parties. The diagnosis of human cancers or normal tissue was confirmed based on histological findings by independent pathologists.

Immunohistochemical (IHC) staining procedures was performed as previously described[30]. Primary antibodies against the following proteins were obtained from Cell Signaling Technology: p-AMPK (#2535, 1:200); from Abcam: DDB1 (ab109027, 1:200); from Abclonal: p-S79 (customization, 1:100), ACADM (A4567, 1:200), EHHADH (A13488, 1:200) and HIBADH (A19871, 1:200); from Proteintech: PROX1 (11067-1-AP, 1:200), CUL4A (14851-1-AP, 1:200), CUL4B (12916-1-AP, 1:200), BCAT1 (13640-1-AP, 1:200), BCKDHB (13685-1-AP, 1:200), ACADSB (13122-1-AP, 1:200), DLD (16431-1-AP, 1:200) and HMGCL (16898-1-AP, 1:200). Signals were detected using Envision-plus detection system (Dako, Carpinteria, CA, USA) and visualized following incubation with 3,3'-diaminobenzidine. The staining intensity was graded from 0 to 2 (0, no staining; 1, weak; 2, strong). The staining extent was graded from 0 to 4 based on the percentage of immunoreactive tumour cells (0%, 1%–5%, 6%–25%, 26%–75%, 76%–100%). A score ranging from 0 to 8 was calculated by multiplying the staining extent score with the staining intensity score, resulting in a low (0–4) level or a high (6–8) level for each sample. Staining was independently assessed based on histological findings by two experienced pathologists.

### Tumour xenograft mouse models

Male athymic BALB/c nude mice (6 weeks old, Chinese Academy of Sciences) were raised in specific pathogen-free conditions. Mice were housed with a 12-h light/dark schedule at $25 \pm 1$ °C and were fed an autoclaved chow diet (XieTong Biology, Cat#1010013) and water ad libitum. The maximal tumour size/burden was not exceeded, and the animal care and experimental protocols were in accordance with guidelines established by the Institutional Animal Care and Use Committee at Ren Ji Hospital affiliated to School of Medicine of Shanghai Jiao Tong University. Tumour xenograft mouse models were established as previous described[30,33,35].

SMCC-7721 cells ($2 \times 10^6$ cells/mouse) stably overexpressing the PROX1 variants (WT, S79A and S79E) were subcutaneously injected in nude mice (six male mice/group, 6 weeks old) respectively. Metformin treatment was once started after inoculation and mice were randomly assigned to receive with or without the metformin dissolved in the drinking water (500 mg/L) daily until the end of experiments. Four weeks post-injection, the mice were sacrificed under anaesthesia, the tumour samples were then collected for further analysis.

Huh7-SCR and Huh7-si1646 cells ($3.0 \times 10^6$ cells/mouse) were subcutaneously implanted into the nude mice (seven male mice/group, 6 weeks old), respectively. Metformin treatment was once started after inoculation and mice were randomly assigned to receive with or without the metformin dissolved in the drinking water (500 mg/L) daily until the end of experiments. Tumour sizes were measured every 6 days. After 30 days, the tumour samples were then collected for further analysis.

### Statistics and reproducibility

Statistical analysis was performed with SPSS 20, GraphPad Prism 7 and GSEA 3.0. No statistical method was used to estimate sample size. Pearson $\chi^2$-test, Chi-square test, Permutation test and the Student $t$-test

were used for compare the differences between the variables. The immunoblots are repeated independently with similar results at three times. The replicates of all experiments were described in the corresponding legends. Kaplan–Meier analysis was used to determine survival. Log-rank test was used to compare survival between groups. The $p$-value $< 0.05$ was considered statistically significant.

### Reporting summary
Further information on research design is available in the Nature Portfolio Reporting Summary linked to this article.

## Data availability
The RNA-seq and ATAC-seq data have been deposited in GEO (https://www.ncbi.nlm.nih.gov/geo/) with the accession numbers GSE214192 and GSE214193, and the proteomics data have been deposited to the ProteomeXchange Consortium via PRIDE partner repository with the dataset identifier PXD036977 (http://www.ebi.ac.uk/pride). The data of RNA-seq, ATAC-seq and proteomic datasets generated in this study can also be viewed in NODE (https://www.biosino.org/node) by pasting the accession (OEP003076) into the search box. The TCGA data can been obtained from the TCGA database (https://xenabrowser.net/datapages/?hub=https://tcga.xenahubs.net:443). The main data are available in the article, supplementary information and Source Data file. Source data are provided with this paper.

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

## Acknowledgements

This work was supported by the Chinese Ministry of Science and Technology Grant (2017YFA0102900 to W.-Q.G.); National Natural Science Foundation of China Grants (81874174 and 82073190 to Y.L.; 81630073 and 81872406 to W.-Q.G.; 82073258 and 81702864 to Y.Z.); Science and Technology Commission of Shanghai Municipality (20JC1417600 to W.-Q.G.), the 111 project (B21024 to W.-Q.G.) and the KC Wong foundation to W.-Q.G.; Shanghai Rising-Star Program (18QA1402600 to Y.L.); Innovative Research Team of High-level Local Universities in Shanghai (SHSMU-ZLCX20211602 to Y.L.); Shanghai Municipal Commission of Health and Family Planning (2018YQ12 to Y.L.); The author gratefully acknowledges the support of SA-SIBS scholarship programme (to Y.Z.); the Excellent Youth Program of the Sixth People's Hospital, School of Medicine, Shanghai Jiao Tong University (ynyq202105 to Y.Z.).

## Author contributions

Y.L., Y.W., Y.X., Y.Z. and W.-Q.G. contributed to the design of experiments, preparation and revision of the article. Y.L. Y.W., M.L., G.Y., K.Q., L.Z. and Y.Z. performed the biochemical studies. Y.L., Y.W., M.L., F.W., and Y.Z. performed animal studies. Y.W., F.W., Y.T., Y.S., assisted with xenograft and IHC assays. Y.L., Y.W., M.L., F.W., Y.T., L.L., Y.S., L.Z., G.Y., K.Q., J.L., H.J. and Y.Z. contributed to the acquisition of the data. Y.L., Y.W., Y.X, Y.Z. and W.-Q.G. contributed to the data analyze and writing of the manuscript. All authors commented on the manuscript.

## Competing interests

The authors declare no competing interests.
