## [Peer Review File · Nature Communications]

Title: AMPK induces degradation of the transcriptional repressor PROX1 impairing branched amino acid metabolism and tumourigenesisREVIEWER COMMENTS

Reviewer #1 (Remarks to the Author):

Wang et al designed the study to ask the question on how cancer cells adapt their metabolism under nutrient deprivation. The core of the manuscript describes following chain of molecular events: 1) under glucose deprivation, nutrient sensor Ser-Thr protein kinase AMPK gets activated and phosphorylates the transcriptional repressor PROX1 protein; 2) in turn, this phosphorylation of PROX1 results in its poly-ubiquitination by CUL4-DDB1 E3 ligase complex leading to PROX1 degradation by proteasome; 3) as a result of PROX1 degradation, the transcription of genes, that would be repressed by PROX1 (directly or indirectly), gets re-activated and among these genes the authors found the metabolic enzymes for BCAA catabolism, a finding that is supported by lower levels of BCAA in cells depleted of PROX1; 4) given that BCAAs are essential amino acids that serve as metabolic cues to activate mTORC1 nutrient sensing signal transduction pathway, authors also show that mTORC1 signaling is inhibited in PROX1-depleted models. All these findings they collect in different cell and animal models in the context of liver cancer, lung cancer and in non-tumoral settings in vitro and in vivo. Moreover, they re-analyze available transcriptional data sets and analyze histological samples of liver cancer and lung cancer patients to show the correlations between PROX1 levels, BCAA enzyme expression, AMPK activation and patient survival. Thus, the authors report a novel regulatory link between pro-catabolic LKB-AMPK and pro-anabolic mTORC1 signaling in the conditions of glucose deprivation. In this mechanism, phosphorylation of transcriptional repressor PROX1 by AMPK releases the transcription of metabolic enzymes that degrade branched amino acids and thus it would limit their availability for mTORC1 activation, suggesting oncogenic role for PROX1.

Overall, although it is an interesting report, in its current form, it has a major incurable problem - it is overloaded with data that are assembled with limited explanations. This makes the story uncoherent, dispersed, incremental and at places hardly comprehensive. Notably, the manuscript contains 130 labeled figure panels in 7 main figures and 86 labeled figure panels in 9 supplementary data figures. Given that considerable portion of the figure panels both in main and in supplementary data also contain the subpanels (multiple IHC staining, pictures and graphs, different cell types) this manuscript lacks sufficient descriptions, author's interpretations and guidance of reader from question to question through the story. Sadly, the wording and language does not convey clear messages. The formulations are vague, the definitions of the terms are missing, there is no sufficient information about how experiments were conducted and the author's interpretations are limited. Beyond these presentation issues, some of the claims especially regarding the role of AMPK-PROX1 upstream of "tumor metabolic plasticity" need further experimental evidence. Moreover, it is not clear how this mechanism will be activated in physiological and pathophysiological settings and how selective it is for lung and liver cancer. At least for liver cancer it was shown that chronic mTORC1 inhibition enhanced HCC development in cancer model induced by DEN+HFD treatment (PMID: 24910242). Thus, it would be important to contextualize the novel findings of this manuscript in relation to already available bulk knowledge on the crosstalk of AMPK-mTORC1 in physiology and in cancer such as HCC. Importantly, the choice of models, conditions and the links between different analyses should be directed by clear

hypotheses but in its current form the manuscript hardly has any explanations on the use of the models/conditions that were employed. To this end, the U-turn switches between HCC, lung cancer, MEFs, normal non-tumoral liver mutants are puzzling. It is not clear how the analyses in non-malignant models support the conclusions of the authors on the role of PROX1 in the controlling cancer cell metabolism and vice versa it is not evident how findings in cancer context would enrich our understanding of PROX1 functions in physiology. Finally, a number of important controls in different models/analyses that are used are missing. The specific comments regarding the experimental work are detailed below.

Comments on the experimental work:

- 1) Study was initiated with TMT labeling in HUH7 cells. The reasoning for performing these analyses in HCC line, the selection of the glucose starvation and the evidence of the metabolic stress should be provided. Regarding glucose starvation conditions used throughout the study, it is not clear why the timing of treatment is different in different figures (6h,12h,16h, 24hrs), it should be explained.
- 2) The controls of effective AMPK activation should be presented in in vivo analyses when fasting or metformin treatment is employed. The same for PROX1 protein levels, the IF and IHC analyses should be backed by immunoblot of nuclear fractions (e.g. Fig.1h). How authors interpret that IF analyses in Fig1g show effective decrease of PROX1 staining already with 5mM metformin treatment, yet it was not associated with decrease in total protein levels in Fig.1f?
- 3) Explanation of cell death rate calculations as % should be provided (Fig.1j-1l). The controls of PROX1 depletion should be presented for these analyses. The impact of PROX1 depletion on cell viability should be expanded to cell proliferation and apoptosis. When measuring the proliferation the authors should complement their findings with metabolic kits with cell number measurements, BrdU labeling, pH3 IF or FACS. The same for apoptosis (FACS, immunoblot of cleaved caspase and PARP). These analyses should be done in cancer cell and non-cancer cell types to clarify the specificity of the mechanism.
- 4) The impact of AMPK mediated phosphorylation of PROX1 on its poly-ubiquitination should be demonstrated (in vitro Ub assay of PROX1 mutants, Ub immunoblot, Tube pull-down). The analyses presented on Suppl Fig.2g with Mg132 treatment should be extended to later time points as 6hours is the earliest time when degradation of PROX1 protein might happen as judged from Fig.1.
- 5) Throughout the study the analyses of PROX1 protein levels should be quantified and average of several independent repeats presented. For example, in relation to Fig.1m authors mention that PROX1 protein levels were significantly upregulated in HepG2 cells upon knock-down of AMPK, yet the blot panel of PROX1 is unconvincing. Similarly, in Supp Fig.1j, 1k.
- 6) Fig.1r the controls of treatment should be presented (LKB and AMPK depletion). The IHC of PROX1 presented is not evident and should be ameliorated. The analyses in Tumoral vs non-Tumoral area would be interesting to show.
- 7) The labeling of WB panels on Fig.2d, 2f, 2h, 2k is not clear (2d-what is PROX1 in middle panel?; 2f- what is PROX1 on top panel? 2h- what is AMPKa2 on middle panel? 2k-what is the FLAG on middle panel and what serves the Coomassie gel?).
- 8) Fig.2k, it is not clear what analyses are presented. If it is co-IP then the authors should include

essential controls for non-specific binding of PROX1 proteins to GST-beads before concluding on the effect of S79 mutation on the interaction with AMPK (which seems unchanged by the mutation contrary to the conclusion in line 228-230). Fig. 2l, 2m controls of AMPK activity should be presented. Also, in case of the IF analyses on Fig. 2e, it would be more relevant to perform those under glucose starvation and to establish dynamics of phosphorylation, nuclear export and degradation of PROX1.

9) In regards to PROX1 phosphorylation by AMPK, the analyses of known AMPK substrates in the employed experimental conditions would be insightful to understand the dynamics of AMPK activity towards PROX1 in comparison to other known targets (autophagy, metabolism, mTOR signaling). To this end, how authors explain that S79 PROX1 is highly phosphorylated at basal state in the untreated cells (Fig. 2i)? In the same line, this phosphorylation is also present in AMPK^{KO} MEFs and it is sensitive to glucose starvation in this KO model (Fig. 2o). The later is contrary to what is commented in the text line 238-240. The IHC captures on Fig. 2p are too small and not informative, thus, the presentation of those analyses should be ameliorated.

10) Focusing on PROX1 phosphorylation and polyubiquitination, the link between these two events has to be better supported (PROX1 Poly-Ub analyses in the models of AMPK activation/inhibition). Do authors suggest that CUL4-DDB1 activity is stimulated by Glu starvation and AMPK activation? How it is modified in cancer vs non-transformed cells?

Specifically, in relation to AMPK mediated phosphorylation and stability of PROX1, it would be important to assess the half-life (under CHX treatment) of S79A and S79E mutants compared to WT PROX1 under glucose starvation. The analyses in Fig. 3j, 3k, 3m and presented in Suppl. Fig. 3h, 3i should be quantified and the half-life of PROX1 protein calculated to be able to draw the conclusion on the dynamics of PROX1 protein turnover. Moreover, it is unclear how authors arrived to a conclusion that S79A mutation attenuated interaction with DDB1 (Fig. 3g, line 274). The quantification of several independent analyses should be presented. Similarly, the quantification of multiple analyses of protein levels of PROX1 upon knockdown of CUL4-DDB1 should be provided (e.g. Suppl. Fig. 3e, Fig. 3n). IF analyses are not adapted for quantitative evaluation of protein levels (Fig. 3h, line 283-286).

Moreover, the analyses of dose response of CUL4-DDB1 overexpression are not clear (Suppl. Fig. 3f, 3g). To this end, while the Flag-PROX1 levels respond to the overexpression of E3 ligase complex, the overexpression of E3 enzyme seems not gradual. Thus, how to interpret this gradual effect on PROX1 levels? How exogenous PROX1 protein is degraded if AMPK is not activated? The analyses of PROX1 polyubiquitination could be informative in this case.

Finally, the heat-map on Fig. 3p is derived from analyses on Fig. 3o, however it is not clear what tumoral area and how it was quantified. It would be interestingly to discuss what are those HCC tumors in which PROX1 levels are high.

11) The authors employed a murine model of liver specific knockout of PROX1 to perform RNA seq (Fig. 4) in order to understand the physiological consequence of AMPK dependent PROX1 phosphorylation. It is not clear how those are related to metabolic rearrangements in cancer under AMPK activation (lung cancer and liver cancer evoked in previous analyses). In the same line, the gene signature and pathway analyses in ATAC-Seq should be discussed (Fig. 4j)? How specific it would be to BCAA metabolism? The relevance of these analyses to metabolic adaptation in cancer or under glucose

deprivation challenge or under AMPK activation should be explained. To this end, the correlative analyses presented in Fig.4s should be extended to pAMPK as the upstream player of PROX1 in the mechanism that is studied.

12) It would be important to at least to discuss what are the mechanisms of differential methylation of H3K4me3, H3K43me and H3K27Ac in PROX1-depleted cells/livers? How selective it is to the genes involved in degradation of BCAA?

13) What is the subcellular localization and chromatin recruitment of PROX1 S79 point mutants especially the resistant to degradation PROX1-S79A? What is their effect on BCAA metabolic gene expression (recruitment and modification in promoters)?

14) Given the diurnal control of the AA degradation, authors should specify the time of tissue collection for metabolic and molecular analyses. It is not clear how authors derived the graph on Fig.5c if on graph Fig.5a none of the BCAA reach differences of 50% decrease. The methodology of relative BCAA level calculation should be better described. As previous analyses with the point mutants of PROX1 show that its degradation was promoted by the glucose starvation and AMPK activation, the similar experimental settings should be employed in these analyses.

15) The important analyses in HUH7 cells presented on Fig.5e should be complemented with the controls of PROX1 expression.

16) In the metabolic flux analyses, the scale in graphs of Fig.5g and 5i are not clear. Were the values of heavy isotopologues normalized to the total metabolite levels? The glutamine levels should be measured to understand if the uptake is different. The conclusion that authors make about PROX1 required for BCAA production through transamination needs further support regarding the expression of GLS2 and the role of PROX1 in its control. Given that the focus of this study is BCAA degradation upon AMPK activation, the flux analyses should be performed under glucose starvation and KIC, C5-Coa and the TCA intermediates should be analyzed.

17) The suggestion that findings of lower BCAA levels in Fig.5g would impact mTORC1 activation need further support. Given that the PROX1-related mechanism will be triggered by active AMPK, the contribution of other AMPK driven mechanisms upstream of mTORC1 should be studied/discussed. Overall, the analyses of TORC1 activation should be quantified (e.g. Fig.5k, 5l, Supp Fig. 6b and 6c) and they should be complemented by analyses of AMPK status.

18) The description of Fig.5m is missing. It is not clear how the treatment was done and what question this experiment addresses. It would be important to elaborate on how the rescue with exogenous Leu or BCAA would work if it is degradation of BCAA that is activated in cells depleted of PROX1. These metabolite rescue analyses should be complemented by metabolomics analyses.

19) The link of mTORC1 to glycolysis in the context of "AMPK->hPROX1->PROX1 degradation->increased BCAA catabolism" needs clarification. The experiments presented on Fig.5o-x require controls of treatments (levels of PROX1 expression/knockdown/rescue). The metabolic flux with labelled glucose should be performed as from available in material and methods information, the enzymatic kits that were used are not adapted for measurement of consumption.

20) What would be the explanation of increased proliferation in cells expressing S79E PROX1 mutant (Fig.6)? Importantly, the proliferation was assessed by CCK8 assay which is based on the formazan reduction and, given the metabolic changes that occur upon PROX1 downregulation, it is not an

adequate approach. Should be supplemented with BrdU labeling and by cell counting.

21) The histological images on Fig.6y should be increased in size and the description of the quantification methodology should be detailed. The scale bar should be included. The quantification of lesions needs to be normalized to area of tissue analyzed and the grading of lesions must be explained (Fig.6z).

22) Leu treatment in vivo, needs further controls including the effect on TORC1 signaling, proliferation, metabolite levels. The analyses in Supp Fig 7h-7m are mentioned in one sentence, thus it is difficult to understand what they represent. The metabolic profiling in the tumors should be performed to characterize how PROX1-S79E and metformin treatment affect BCAA metabolism. The phosphorylation of p-Ser79 PROX1 and its total protein levels as well as polyubiquitination should be analyzed in course of tumorigenesis together with status of AMPK to draw clear parallels between these molecular events.

23) The model sgPROX1 on Fig 6r-u should be introduced. As for any other treatment, the controls of phenformin efficacy and analyses of PROX1 phosphorylation as well as protein levels should be included. In relation to this main figure, the choice of the enzymes in Suppl Fig.8a-d should be explained. The correlation with the TORC1 signaling should be established in the sgPROX1 conditions and in tumors that are characterized by “high” expression of metabolic enzymes in BCAA metabolism.

24) Given that the authors link the pro-tumorigenic effect of PROX1 expression to repression of BCAA catabolism, it would be important to perform the correlation analyses with TORC1 activation by analyzing its phospho-targets (Fig7).

25) The schema of the study should be modified to better convey the take home message.

Comments on the text and formatting:

1) The authors are encouraged to revise the English to define the terms and concepts that they use. They also should employ the action verbs that are specific. To this end, avoid the unclear verbs such as “regulate, impact, control” and instead use “activate, repress, increase/decrease” so that the message is evident.

The vague phrasing examples that need definitions in different parts of the text are: broadly coordinate (line 12), essential determinant (line 138), considerably impaired (line 391), aggressive phenotype of Prox1 (line 422); central switch that orchestrates fine-tuning (line 94-95), aberrant BCAA metabolism (line 112), malignant phenotypes (line 113), intrinsic cancer properties (line 114), property of metabolic plasticity (line 131), the metabolic fitness (l165), BCAA metabolism reprogramming, metabolic vulnerability to therapeutic drugs, reprogramming switch... etc.

The examples of terms that need definition are: metabolic plasticity, tumor transformation, metabolism reprogramming, property of cell metabolic plasticity (line 128), metabolic therapy (line 140), metabolic invulnerability (line 534)

2) the abbreviations in the title and the abstract should be limited and explained (PROX1, LKB1, BCAA, CUL-DDB1, HCC, AMPK, mTOR). The take-home message of the story should be clarified. In the title, it would be advisable to make clear what is PROX1, what is BCAA and how exactly it is impacted by LKB-AMPK. In abstract, it should be clear what is the cellular function of PROX1, how it is relevant to tumor metabolism in general and BCAA metabolism specifically (current state of the art). Their relevance in cancer should be stressed.

3) The introduction would benefit from including more information on what is known about the role of PROX-1 in tumorigenesis (expand section lines 120-126). The take-home message in the concluding paragraph of the introduction should be edited as it is not clear in its current formulation.

4) Figure legends should be revised and when missing added (e.g. Fig4j and 4g). The minimal required information to understand the experimental conditions such as type of treatments, measurements and the type of cell line/tissue should be indicated in the legend.

5) The presentation of histology analyses should be ameliorated as captures are hardly visible. Scale bar and the insets with higher magnification should be added.

6) As for discussion, the main conclusion that “PROX1 is an essential determinant for LKB-AMPK axis that mediates tumor metabolic plasticity to control therapeutic response (line 536-538)” needs further experimental evidence. To this end no flux metabolomics analyses were performed to define the metabolic rearrangements downstream of activated or inhibited AMPK in cancer cells. The selectivity of this identified mechanism to the cancer context should be established. Authors also write that they have discovered “the molecular mechanism of LKB-AMPK axis mediated BCAA metabolism reprogramming” (line 540). “PROX1 controls the intracellular BCAAs concentrations in an AMPK-dependent manner” (line 554). However, the analyses presented in Fig.5 do not address the role of AMPK in BCAA degradation in cancer context and the authors do not measure the levels of BCAA under conditions of increased or inhibited AMPK. Another conclusion that authors discuss is that “PROX1 deletion in HCC and lung tumor significantly increases expression of the most majority of key enzymes for BCAA metabolism through mediating epigenetic modifications” (line 552-554). This is an overstatement as, first, epigenetics marks were analyzed only in HCC cell model (HUH7 cells). Second, although ATAC-seq in non-tumoral PROX1 LKO mice (Fig.4) showed marked changes in chromatin accessibility (suggestive of active transcription) this was less evident in HUH7 cells depleted of PROX1 (Suppl.Fig.4). As for epigenetic defects in PROX1 depleted cells, those were suggested by CHIP-qPCR in some regions of selected genes with very modest enrichment around 10folds and about 20-30% increase/decrease upon knockdown or knockout of PROX1. Thus, if it is the point that the authors want to stress, they should perform unbiased genomic analyses of epigenetic landscape in PROX1 mutant cells and investigate what is the functional link between PROX1 and epigenetic control of transcription.

Reviewer #2 (Remarks to the Author):

In this manuscript, the authors discover PROX1 transcription factor as a substrate of AMPK, and that AMPK-dependent phosphorylation of PROX1 on Ser79 results in its ubiquitin-dependent degradation, which requires the Cul4A/Cul4B/DDB1 complex. The authors go on to demonstrate that PROX1 is suppressing mRNA expression of core branch amino acid metabolic enzymes, and in PROX1 KO cells and tissues, there is a upregulation of those same enzymes at the protein level (ACADSB, ACADM, BCKDHB, etc). IHC on human liver samples supports an inverse relationship between PROX1 levels and levels of these BCAA enzymes. The authors next examine whether the changes in BCAAs is sufficient to alter mTOR signaling, which is known to be very reliant on amino acid levels. siRNA to PROX1 modestly lowers P-S6K and 4ebp1 indicative that mTOR signaling is impacted by the lowered levels of BCAAs. The authors go on to define the impact of sgPROX1 in Kras Lkb1 mutant GEMM models of NSCLC and the

impact of enforced expression of S79A and S79E cDNAs of PROX1 in Kras Lkb1 mutant mice. Finally IHC and comparison of gene expression in human samples make this a very thorough and very compelling study.

The authors have made what appears to be a significant finding in decoding new targets of LKB1 and new regulators of cancer metabolism more broadly. The extensive data across many systems makes the paper almost acceptable in its current form, but there are a few minor experiments the authors could do which would greatly enhance the manuscript and will not require that much effort.

Major points:

1. In Figure 1, the authors use glucose starvation and metformin as AMPK activators and their data is convincing but in the last 5 years there are now a number of selective on-target synthetic AMPK activators that work readily in cells, i.e. 991 or MK8722. The authors should examine whether these direct pure AMPK activators can also trigger PROX1 degradation in cells. They almost certainly will but this would be an important point for the authors to experimentally address.
2. The effects of the 79A and 79E mutants used throughout (e.g. across Figure 5) is presumably due to the protein stability of these two PROX1 mutants. However the data on the mutants in Figure 3M is not very convincing there is a major difference, in fact the data in Figure 3N is more convincing. WT vs 79A vs 79E PROX1 levels are needed for interpreting Figure 5E, 5Q, 5R, 5S, 5T, 5U, etc. A western blot of PROX1 levels is needed from the cells used in Figure 5 panels.
3. IHC for PROX1 or its targets in Figure 6M, 6N (like authors did in Supplemental Figure 8A) would also help explain the behavior of the S79A and S79E alleles in the mouse tumors in vivo. Such IHC should go in the main figure of the paper.
4. In Figure 6 regarding the phenformin experiments, while it is correct that sgPROX1 is rendering the tumor cells less sensitive to phenformin, a complicating factor is that the tumors lacking PROX1 are not dividing as much to begin with. I would simply change the wording from “dictates therapeutic response” to “blunts therapeutic response”. There are likely many targets of AMPK that play a role in mediating phenformin response. This data with PROX1 is surprising and exciting but given that the tumors are growing slower and are smaller to begin with, one cannot so directly compare them.

Minor points:

1. A lighter exposure of the PROX1 immunoblot in Figure 1M would help make the point more clearly. The AMPKa1 shRNA did not work very well in this experiment, authors might want to redo the experiment.
2. In Fig. 3f the HA-Cul4 input is not shown. The figure legend nor the figure panel mention what is HA-tagged.

Reviewer #3 (Remarks to the Author):

The manuscript by Wang et al. investigates the role of Prospero-related homeobox 1 (PROX1) in coordinating hepatic and lung tumor metabolic plasticity through the modulation of the BCAA metabolism in response to AMPK signaling. The authors first observed that PROX1 levels are decreased due to impaired protein stability in response to glucose starvation, P-AMPK (LKB1-AMPK) or metformin. They went on to show mechanistically that these metabolic perturbations lead to the AMPK-dependant phosphorylation of Serine 79 on PROX1, which targets PROX1 for degradation through the CLU4-DDB1 E3 ubiquitin-ligase complex. The loss of PROX1 transcriptionally modulates the expression of multiple enzymes that belong to various metabolic pathways, including branch chain amino acid metabolism (BCAA). They show that high levels of PROX1 in response to LKB1-AMPK axis increases BCAAs levels in hepatic and lung cancer models, contributing to mTOR activity, cancer progression and resistance to metformin or phenformin treatment and contributes to clinical outcomes.

The data reported in this manuscript is very relevant to the field of tumour metabolism and therapeutics and the conclusions improve the mechanistic understanding of AMPK-dependant metabolism. However, some mechanistic studies lack essential controls, and some conclusions are not fully supported experimentally as currently reported. The data presented also focusses on a specific metabolic aspect of the observed phenotype and would benefit from efficiently integrating the findings with the previously reported metabolic roles of PROX1. The data presented is also very dense and the description of results sometimes lacks precision: the manuscript would benefit from a thorough and more detailed description of the rationales and results. Below are some major and minor points that would need to be addressed to improve these aspects.

Major points

1- The cell and tumour models used throughout the manuscript get interchanged between lung and hepatic tumour models. It should be specified in the text why the focus is put on these two cancer models and the rationale for investigating one or the other at specific instances (do they expect same PROX1 functions if different tumour types?). In addition, 2 different cell lines are used as models for hepatic cancers (none for the lung models) and they are being used alternatively without justifications, with several of the results only reported for one of the two cell lines. It is important that the authors justify reason for switching between models (or cell lines from a same model) and report differences observed between them, as it seems that some differences are indeed observed.

2- In figure 3, the authors report that the CLU4-DDB1 E3-ubiquitin ligase complex mediates decrease in PROX1 protein levels/stability upon glucose starvation or AMPK activation. Yet it is not clear whether this complex is expressed/active in all models studied, as they need to ectopically overexpress the complex in Huh7 cells to detect it and to study its effect on PROX1 levels (was the interaction study done in HEK293 cells, it is not specified?). Why not kd the endogenous CUL4-DDB1 in Huh7 cells as they did in HepG2? Is it because it is not expressed endogenously in Huh7 cells? If that is the case, it partly

invalidates this mechanism as the main ligase responsible for PROX1 degradation in response to low glucose (since PROX1 decrease upon low glucose is also observed in Huh7 cells). Also, ectopic expression could lead to overexpression of proteins and facilitate non-functional interactions. The levels of ectopic proteins should be quantified, compared between each other and related to endogenous protein levels. Also are endogenous levels of CUL4-DDB1 proteins modulated upon glucose starvation in the different models? The authors should clarify and comment on this.

3- None of the western blots is quantified and none of them shows molecular weight markers. This should be corrected. Many conclusions are drawn from small changes in band intensity that are not always obvious (eg. Fig.3h; 5k-l-m; 6b-c; Supp Fig.3e, etc).

4- Confirmations of KD or KO should be shown for each si-, sh- and sg-RNA used. Eg. Figure 1j and k first introduce PROX1 kd but no confirmation that the protein is actually depleted is shown at that point. (shown later in the manuscript for some models, but not for all the lines, etc). The same is expected for all proteins that are depleted or deleted.

5- It is not shown in figure 2o whether glucose starvation induces S79 phosphorylation on endogenous PROX1 in MEF (as it is an IP) so the statement on p, 9 line 13 is not supported by figure 2o. To support this statement, figure 2p (blot for p-S79-Prox1) should be repeated including the mice treated conditions from figure 1h (fasted and metformin treated). In figure 2p, IHC for AMPK-P and AMPK and PROX1 total should be shown as well.

6- In Figure 3c, blotting for other CULs factors (eg CUL5) should be included as it was done in Figure 3b to show these are specific interactions.

7- The co-localization experiments only show that these proteins are located in the same organelles and should not be used to conclude endogenous interactions (the text implies that).

8- In Figure 3c, the DDB1 levels are affected (decreased) by glucose starvation. How does this affect the role of this complex in regulating PROX1? In Figure 3b is this at basal levels? What about in low glucose? In figure 3f, Why blot DDB1 in input and not only HA as it is done in the supplemental? Is this a mistake?

9- In figure 4, there are a few issues with the transcriptional/epigenetic analyses:

- a different set of genes regulated by PROX1 is shown for each model used. The same list of genes should be used in all the models to show expression.
- The GSEA result implies downregulation of the BCAA pathway in PROX1 KO condition (4c) but the heatmap (4b), figure 4d and qPCR validations (4e-f-g, etc) and epigenetic data imply that low PROX1 upregulates a subset of these genes. Is the NES really negative in 4c?
- Are these transcriptional/epigenetic reprogramming of BCAA genes modulated by AMPK itself? It is not shown directly but is implied in the text and should be experimentally demonstrated in this figure.
- What is going on with BCAT1-2 (Supp Fig4c and Figure 4s)? These genes show a different trend than other genes (downregulated upon loss of PROX1). What is the consequence of this different trend as

they are the enzymes that initiate the catabolism of BCAA? This should be shown in main figure and discussed.

10- In figure 4j, include stats about differences in accessible chromatin between WT and Prox1 KO for upper panels. Importantly, the authors should show a similar panel restricting the analysis only to the ATAC results at the promoters of the BCAA genes that they study. The genomic representations shown in figure 4 and in supplemental 4k do not convincingly show that chromatin is “more opened” at promoters of genes coding for BCAA enzymes. Do the sites that show increased opening in figure 4j actually include BCAA genes?

11- In figure 5, can the author better describe why the BCAA-related metabolites are decreased upon loss of PROX1 while most genes regulating them are upregulated? Could this be linked to the trend of BCAT1-2? Is it because the BCAT enzymes catabolism deplete BCAA-derived metabolites? The authors should discuss this. Few points that could help this discussion:

- In 5f-1: Decreased incorporation of labelled Gln into BCAA but increased KIC from labelled leucine. Blots or RNA should be provided to show GLS and BCAT1/2 levels upon PROX1 inhibition to better understand this result.
- BCAT1/2 is bidirectional but the breakdown of BCAA to KIC seems to be preferred upon PROX1 ablation (according to the data 5f-i)? Is PROX1 acting to increase intracellular BCAA levels due to inhibited BCAA catabolism activity but also due to increasing BCAA synthesis? (From 5f-g?)
- In Fig 5I-M: Are extracellular BCAAs contributing to mTORC1 activation? Could they be taken up by LAT1? Gln level shows opposite trend to Leu level in Fig 5A, which could suggest LAT1 activity upon modulation of PROX1?

12- The conclusion of Figure 5 does not seem to be fully supported: “PROX1 activates the mTOR pathway via BCAA to promote tumour progression”. It is clear that altered PROX1 levels modulates mTOR signalling (which occurs at least partly through modulation of BCAA levels) but the link that this is responsible for tumour aggressiveness is not clear. Supplemental Fig 6d is the only figure actually testing mTOR modulation. However its conclusion is not convincing: “rapamycin, a mTOR inhibitor, considerably reversed Huh7 cell death under glucose-deprived conditions, an effect comparable to that of cells with PROX1 depletion (Supplementary Fig. 6d), suggesting that inactivation of mTOR by AMPK-PROX1 axis is critical for tumor cells to establish metabolic adaptation” the fact that it is “comparable” to PROX1 depletion does not mean that the effect of PROX1 on cell growth is actually going through mTOR inhibition. Also, how does rapamycin affect PROX1 levels? What is the status of PROX1, pS6 and p4EBP upon glucose starvation? Glucose starvation should decrease PROX1 (and mTOR?) so then why further inhibiting them prevents the apoptosis? This part is confusing (more rationale needed?). Maybe an approach using ectopic expression of PROX1 in PROX1KO cells and rapamycin treatment would better support this? (blots for pS6 should be quantified, eg. effect in 5I is not clear). Also, 5X-Z: Staining of pS6 (on top of Ki67 in S6M) on these tumours would better link the aggressiveness with PROX-1 dependant mTORC1 activation.

13- In figure 6k-l, the PROX1 KD already has a very significant effect on tumor growth (orange and green

curves in 5k) so it is very difficult to conclude whether loss of PROX1 makes HCC cells resistant to metformin - can the growth be even further decreased? Same observations in r-s-t-u. Can the authors show that PROX1 is necessary for the metformin effect instead?

- In the same line of thoughts, metformin decreases PROX1 as shown in figure 1, so does this imply that the loss of PROX1 is necessary for metformin effect on mTOR and tumor growth? This is not clearly shown here but should be the purpose of these experiments. Can they re-express PROX1 as rescue upon metformin treatment? Or use the S79A mutant to show this?

14- Regarding the previously reported role of PROX1 as a transcriptional co-repressor interacting with ERR-PGC1 axis (in bioenergetic metabolism and circadian clock), or NuRD complex (to repress Notch pathway) and other reported mechanisms, can the authors discuss whether these TFs might be involved in mediating the PROX1 transcriptional activity on BCAA metabolic enzymes? Is the effect directly due to PROX1 recruitment to the BCAA gene promoters or could this happen indirectly as well? Can the effect on mTOR be linked to the PROX1-ERR-PGC1 axis through modulation of bioenergetic capacity? Likewise, could other metabolic pathways/functions altered by the loss of PROX1 (eg fatty acid metabolism, glutathione, peroxisome, etc as shown in figure 4a) be affected by the AMPK-PROX1 axis and contribute to the PROX1-mTOR effect? Better integration of previously described PROX1 roles should be discussed.

Minor points

1- Some parts of intro would benefit from clarifications or rephrasing:

- Introduction line 21: “(...) imposing a metabolic invulnerability to therapeutic drugs”. Do they mean resistance to therapeutic drugs?
- Also in introduction, the part “we uncovered that PROX1 phosphorylation by AMPK drove BCAA metabolism reprogramming to suppress mTOR signaling activity and overall energy metabolism” is also confusing as it suggests that PROX1 activity suppresses mTOR activity (we don't know yet that phosphorylation of PROX1 targets it for degradation). However, mTOR activity is known to be linked to drug resistance in several tumor types, which makes the rest of the sentence about invulnerability to therapeutic drugs confusing.

2- The cell models used for some experiments are not systematically reported (eg. Co-IP-MS in figure 3, etc). This should be corrected throughout the manuscript.

3- Mistake in Figure 2f? Where is AMPK? Is it FLAG AMPK α 2? Or blotted AMPK? The text refer to AMPK2 α .

4- In Supp Figure 2f, DAPI imaging is not scaled according to the other images. Include the right image.

5- The expression “therapeutic drugs” should be changed to refer to metformin or phenformin throughout the manuscript.

6- p6 line 3, define TMT (undefined acronym).

7- In figure 4a, the reference for PROX1 ChIP-seq is Charest-Marcotte et al., 2010 and not "Alexis et al., 2011". Also Fatty acid and not "Fathy acid".

8- Gene names when referring to mouse models should not be all capitalized. Change in Supplementary Figure 7B and supplementary Figure 8A.

9- Wording for some conclusions is very strong (eg. "drive" and "essential event"). Eg. in "Collectively, these findings demonstrate that downregulation of PROX1 by AMPK is an essential event for tumor metabolic adaptations during nutrient deficiency." This figure suggests that increase in PROX1 contributes in part to the effect of decreased AMPK on survival and apoptosis as kd of PROX does not completely rescue this effect. AMPK likely signals through other factors that also contribute to the survival and this should be nuanced in the conclusion/text.

10- Supplementary Figure 2g is important and links figure 2 to figure 3: it should be in the main figures and part of Figure 3.

11- In all the GSEA plots, the p-values should not be 0. Actual adjusted p-value should be reported.

12- in figure 4j: the legend does not report "j" panel; mistaken with "g"?

13- In Figure 5 c-d, what does BCAA means in the graph axis? Does it mean all BCAAs compiled together? That should be detailed.

14- Figure 5e should show levels of PROX1 (ectopic) as they state that they reached a level equivalent to endogenous but it is not shown.

15- On p.6 line 26-27: the authors state that "Given the role of PROX1 in AMPK-mediated metabolic plasticity (...)". The authors have not shown a role of PROX1 on metabolic "plasticity" yet so it is not clear what they are referring to. Rephrase?

16- in general, figures are very dense and convey multiple messages (eg. Figure 5 starts with BCAA metabolomics and ends with pS79 effect on migration, invasion and tumor volume (and going though glucose consumption/mTOR activation). The link between these multiple messages is not always clear or supported and figures could be split (or re-ordered) to better convey the main conclusions. In line with this, better explanations of the large variety of results obtained is needed.

Response to reviewers:

We thank the reviewers for their interests in our work and their insightful and constructive comments and criticisms. Accordingly, we have performed a large number of new experiments and extensively revised the manuscript. We have also expanded the Discussion sections. Please find below our responses to all comments and criticisms. The data generated during the revision are highlighted in red color in the figures and the text.

REVIEWER COMMENTS

Reviewer #1 (Remarks to the Author):

Wang et al designed the study to ask the question on how cancer cells adapt their metabolism under nutrient deprivation. The core of the manuscript describes following chain of molecular events: 1) under glucose deprivation, nutrient sensor Ser-Thr protein kinase AMPK gets activated and phosphorylates the transcriptional repressor PROX1 protein; 2) in turn, this phosphorylation of PROX1 results in its poly-ubiquitination by CUL4-DDB1 E3 ligase complex leading to PROX1 degradation by proteasome; 3) as a result of PROX1 degradation, the transcription of genes, that would be repressed by PROX1 (directly or indirectly), gets re-activated and among these genes the authors found the metabolic enzymes for BCAA catabolism, a finding that is supported by lower levels of BCAA in cells depleted of PROX1; 4) given that BCAAs are essential amino acids that serve as metabolic cues to activate mTORC1 nutrient sensing signal transduction pathway, authors also show that mTORC1 signaling is inhibited in PROX1-depleted models. All these findings they collect in different cell and animal models in the context of liver cancer, lung cancer and in non-tumoral settings in vitro and in vivo. Moreover, they re-analyze available transcriptional data sets and analyze histological samples of liver cancer and lung cancer patients to show the correlations between PROX1 levels, BCAA enzyme expression, AMPK activation and patient survival. Thus, the authors report a novel regulatory link between pro-catabolic LKB-AMPK and pro-anabolic mTORC1 signaling in the conditions of glucose deprivation. In this mechanism, phosphorylation of transcriptional repressor PROX1 by AMPK releases the transcription of metabolic enzymes that degrade branched amino acids and thus it would limit their availability for mTORC1 activation, suggesting oncogenic role for PROX1.

Overall, although it is an interesting report, in its current form, it has a major incurable problem - it is overloaded with data that are assembled with limited explanations. This makes the story uncoherent, dispersed, incremental and at places hardly comprehensive. Notably, the manuscript contains 130 labeled figure panels in 7 main figures and 86 labeled figure panels in 9 supplementary data figures. Given that considerable portion of the figure panels both in main and in supplementary data also contain the subpanels (multiple IHC staining, pictures and graphs, different cell types) this manuscript lacks sufficient descriptions, author's interpretations and guidance of

reader from question to question through the story. Sadly, the wording and language does not convey clear messages. The formulations are vague, the definitions of the terms are missing, there is no sufficient information about how experiments were conducted and the author's interpretations are limited. Beyond these presentation issues, some of the claims especially regarding the role of AMPK-PROX1 upstream of "tumor metabolic plasticity" need further experimental evidence. Moreover, it is not clear how this mechanism will be activated in physiological and pathophysiological settings and how selective it is for lung and liver cancer. At least for liver cancer it was shown that chronic mTORC1 inhibition enhanced HCC development in cancer model induced by DEN+HFD treatment (PMID: 24910242). Thus, it would be important to contextualize the novel findings of this manuscript in relation to already available bulk knowledge on the crosstalk of AMPK-mTORC1 in physiology and in cancer such as HCC. Importantly, the choice of models, conditions and the links between different analyses should be directed by clear hypotheses but in its current form the manuscript hardly has any explanations on the use of the models/conditions that were employed. To this end, the U-turn switches between HCC, lung cancer, MEFs, normal non-tumoral liver mutants are puzzling. It is not clear how the analyses in non-malignant models support the conclusions of the authors on the role of PROX1 in the controlling cancer cell metabolism and vice versa it is not evident how findings in cancer context would enrich our understanding of PROX1 functions in physiology. Finally, a number of important controls in different models/analyses that are used are missing. The specific comments regarding the experimental work are detailed below.

Comments on the experimental work:

1) Study was initiated with TMT labeling in HUH7 cells. The reasoning for performing these analyses in HCC line, the selection of the glucose starvation and the evidence of the metabolic stress should be provided. Regarding glucose starvation conditions used throughout the study, it is not clear why the timing of treatment is different in different figures (6h,12h,16h, 24hrs), it should be explained.

Response:

We thank the reviewer for his/her insightful comments. We apologize that we did not make these issues very clear in the previous version of the manuscript. We initially aimed to screen novel regulator(s) in HCC during metabolic stress, such as glucose starvation and therapeutic drug (e.g., Metformin/ phenformin). Both conditions of glucose starvation and therapeutic drug are clinically relevant in cancers, including HCC, as insufficient glucose supply is frequently found in the tumor microenvironment (1, 2) or metformin treatment of cancers results in glucose insufficiency. Given that glucose starvation treatment to HCC cells generally, at least partially, recapitulates the physiological status of tumors in patients with HCC, "glucose starvation" is therefore selected to identify potential regulators associated with metabolic stress. As the reviewer pointed out, the timing for glucose starvation is indeed not consistent in

different experiments. Actually, we have tested serial time courses of glucose starvation in various cell lines, and some of these results were shown in Fig 1e. For cells used in this study, treatment of glucose starvation over 12h is sufficient to cause metabolic stress in cells, as determined by the levels of phosphor-AMPK (T172). In Fig. 1a-1c, 12h was selected for screen assay. In Fig. 1J and 1k, for evaluation of cell apoptosis in response to glucose starvation, 12h is indeed sufficient to activate metabolic stress and AMPK, but is insufficient to induce robust cell apoptosis (less than 10%). 24h is optimized time course for determining cell apoptosis. In Fig. 1l, originally, the timing is still 24h for cell apoptosis. However, it was suggested to perform time-course dependent effect in MEFs upon glucose deficiency during previous submission to certain journal. To avoid dense data in figures, we have removed the results of other time points. We do appreciate the reviewer's comments and we will try to design, process and present data in a more reasonable way in the future.

2) The controls of effective AMPK activation should be presented in in vivo analyses when fasting or metformin treatment is employed. The same for PROX1 protein levels, the IF and IHC analyses should be backed by immunoblot of nuclear fractions (e.g., Fig.1h). How authors interpret that IF analyses in Fig1g show effective decrease of PROX1 staining already with 5mM metformin treatment, yet it was not associated with decrease in total protein levels in Fig.1f?

Response:

As suggested, we have now presented additional data for phos-AMPK, phos-ACC and PROX1 levels in supplemental Fig. 1h, and see also for convenience. Given the exclusive nuclear distribution of PROX1 in HCC cells and murine liver cells, the protein abundance of PROX1 was determined based on the total protein. For the reviewer's concern with respect to the discrepancy between PROX1 abundance evaluated by two separate approaches (IF and WB), we feel that different assays might give rise to the similar conclusion, but not always to the same extent, partly owing to distinct contrast in each assay. Indeed, PROX1 level decrease in IF analyses was robust, and moderate in WB analyses with the same metformin treatment. By band density normalization, more than 50% decrease of PROX1 was seen. Overall, we feel that metformin treatment of Huh7 cells with 5mM did lead to PROX1 degradation regardless of different assays. More important, this experiment has been repeated many times, we are fully confident of the results and conclusions in this manuscript. Again, we appreciate the reviewer for the strict and professional advice about data presentation.

3) Explanation of cell death rate calculations as % should be provided (Fig.1j-1l). The

controls of PROX1 depletion should be presented for these analyses. The impact of PROX1 depletion on cell viability should be expanded to cell proliferation and apoptosis. When measuring the proliferation the authors should complement their findings with metabolic kits with cell number measurements, BrdU labeling, pH3 IF or FACS. The same for apoptosis (FACS, immunoblot of cleaved caspase and PARP). These analyses should be done in cancer cell and non-cancer cell types to clarify the specificity of the mechanism.

Response: We thank the reviewer for his/her valuable comments. Precisely, data presented is cell apoptotic rate, which is calculated by the percentage of late phase of cell apoptosis analyzed by PI/Annexin-V and FACS, and statistical analysis was shown. Also, similar results regarding early phase of cell apoptosis rate were observed. Due to limited space, this part of results was not presented, and could be provided if necessary. Representative FACS images were included in supplemental Fig. 1k and 1l. PROX1 depletion efficiency in Huh7 and HepG2 was confirmed by immunoblot and relevant data was shown in supplemental Fig. 1i. Given that the effect of PROX1 on HCC cell proliferation has been investigated by our previous work (3), we have used BrdU Cell Proliferation ELISA kit to evaluate cell proliferative capability in cells upon PROX1 depletion to compare the results with that of CCK8 based approach. Results of these two experiments were comparable, and were shown below for your reference. Our findings indicate that AMPK-dependent metabolic regulation of PROX1 appears to be a common mechanism in both cancer cells and non-cancer cell, like MEFs and murine liver and lung cells. We felt that these data are generally sufficient to support the conclusions in the manuscript, considering the figures with overloaded data, as mentioned by the reviewer.

4) The impact of AMPK mediated phosphorylation of PROX1 on its poly-ubiquitination should be demonstrated (in vitro Ub assay of PROX1 mutants, Ub immunoblot, Tube pull-down). The analyses presented on Suppl Fig.2g with Mg132 treatment should be extended to later time points as 6 hours is the earliest time when degradation of PROX1 protein might happen as judged from Fig.1.

Response: We appreciate this constructive comment from biochemical perspective. However, the full-length PROX1 is technically difficult to purify using the prokaryotic system, which contains the structural insight of PROX1. Although partial PROX1 can now be purified, it is not clear which part is implicated in ubiquitination. Therefore, there is no currently suitable way to perform a meaningful in vitro Ub assay. Cellular ubiquitination and immunoblot assays of PROX1 and its mutants have been done as requested by the reviewer above. Results have been shown in Fig. 3m and Fig. 2r and

below for reference. We appreciate the reviewer for the suggestion.

5) Throughout the study the analyses of PROX1 protein levels should be quantified and average of several independent repeats presented. For example, in relation to Fig.1m authors mention that PROX1 protein levels were significantly upregulated in HepG2 cells upon knock-down of AMPK, yet the blot panel of PROX1 is unconvincing. Similarly, in Supp Fig.1j, 1k.

Response: We thank the reviewer for the comments. We have quantified the band density of PROX1 immunoblot. Relative abundance of PROX1 in each independent repeat was significantly increased upon AMPK depletion. Meanwhile, the other reviewer of this manuscript also concerns about the depletion efficiency of AMPK in Fig.1m, we therefore have repeated this assay. The original data in Fig.1m has now been replaced with the new one and was also shown below for your reference.

6) Fig.1r the controls of treatment should be presented (LKB and AMPK depletion). The IHC of PROX1 presented is not evident and should be ameliorated. The analyses in Tumoral vs non-Tumoral area would be interesting to show.

Response: IHC staining of LKB and AMPK have been done, and representative images have been shown in supplemental Fig. 1m. The quality of IHC staining of PROX1 and phosphor-AMPK in Fig.1r has been improved, with more evident images. Since sections of lung cancer used here is initiated in *Kras*^{LSL-G12D/+} (K), *Kras*^{LSL-G12D/+}; *Lkb1*^{fllox/fllox} (KL) mouse model, in which the oncogenic *Kras*G12D allele is conditionally activated following Ad-Cre infection by nasal inhalation. LKB1 or AMPK is only deficient in KRAS-activated tumor tissues. Thus, we feel that analyses in non-tumoral tissues might be less informative.

7) The labeling of WB panels on Fig.2d, 2f, 2h, 2k is not clear (2d-what is PROX1 in middle panel?; 2f-what is PROX1 on top panel? 2h- what is AMPK α 2 on middle panel?)

2k-what is the FLAG on middle panel and what serves the Coomassie gel?).

Response: We appreciate the reviewer for pointing out the mistakes. We are sorry that these panels were mislabeled, and have now been corrected as shown in Fig. 2d, 2f, 2h and 2k.

8) Fig.2k, it is not clear what analyses are presented. If it is co-IP then the authors should include essential controls for non-specific binding of PROX1 proteins to GST-beads before concluding on the effect of S79 mutation on the interaction with AMPK (which seems unchanged by the mutation contrary to the conclusion in line228-230). Fig.2l, 2m controls of AMPK activity should be presented. Also, in case of the IF analyses on Fig.2e, it would be more relevant to perform those under glucose starvation and to establish dynamics of phosphorylation, nuclear export and degradation of PROX1.

Response: We feel sorry for misleading data presentation. Data shown in Fig.2k was GST pulldown assay using purified GST-AMPK incubated with cell lysate of FLAG-PROX1 or its mutant to determine whether PROX1 phosphorylation impacts its binding to AMPK. The specific interaction of PROX1/AMPK has been confirmed in vitro and by pulldown assay (Fig 2d and 2f). Nonetheless, we agree with the reviewer's point. We have repeated this experiment with GST control, and new figure has been shown in Fig.2k. The same for AMPK activity, immunoblot of p-AMPK and its substrate p-ACC has been measured and shown in Fig. 2l and 2m. We appreciate your insightful suggestion. Actually, we previously performed real-time fluorescent imaging of PROX1/AMPK, and PROX1 degradation in the nucleus, but not nuclear export, was observed upon glucose starvation. However, the image quality was low, and the image has not been shown. In addition, data in our hands did not support a potential nuclear export of PROX1 for degradation. It was probably degraded in the nucleus, where PROX1 interacts with CUL4/DDB1 E3 ligase complex and potentially nuclear ubiquitin-proteasome system.

9) In regards to PROX1 phosphorylation by AMPK, the analyses of known AMPK substrates in the employed experimental conditions would be insightful to understand the dynamics of AMPK activity towards PROX1 in comparison to other known targets (autophagy, metabolism, mTOR signaling). To this end, how authors explain that S79 PROX1 is highly phosphorylated at basal state in the untreated cells (Fig.2i)? In the same line, this phosphorylation is also present in AMPK α KO MEFs and it is sensitive to glucose starvation in this KO model (Fig.2o). The later is contrary to what is commented in the text line238-240. The IHC captures on Fig.2p are too small and not informative, thus, the presentation of those analyses should be ameliorated.

Response: We appreciate the reviewer for the insightful questions. The analyses of known AMPK substrate p-ACC has been measured and shown in Fig. 2l and 2m. AMPK is in fact activated at basal state in the untreated cells, please see this in Fig. 1e, 1f and 2i. We feel that the reason that S79 PROX1 phosphorylation is basally phosphorylated lies in the presence of activated AMPK in cells under normal condition. Also, we have provided evidence to demonstrate PROX1 as one of the AMPK substrates during

energetic stress. However, we can not rule out the possibility of other kinases potentially involved. Based on this potential possibility, we think that PROX1 phosphorylation still occurs in the absence of AMPK. Not surprisingly, PROX1 phosphorylation is detectable in MEFs with AMPK deficiency. Additionally, similar result was presented in published literature (4)(see Fig.2f). We have replaced Fig. 2o with a more representative one. The IHC images of p79-PROX1 has been improved and shown in Fig.2p.

10) Focusing on PROX1 phosphorylation and polyubiquitination, the link between these two events has to be better supported (PROX1 Poly-Ub analyses in the models of AMPK activation/inhibition). Do authors suggest that CUL4-DDB1 activity is stimulated by Glu starvation and AMPK activation? How it is modified in cancer vs non-transformed cells?

Response: Thanks for the comments. We have no evidence to suggest the potential regulation of CUL4-DDB1 activity upon glucose starvation and AMPK activation. We proposed that PROX1 is destroyed by CUL4-DDB1 activity under normal condition, and is greatly enhanced following metabolic stresses, by which PROX1 is phosphorylated via AMPK at S79. Such a tag promotes PROX1/CUL4-DDB1 interaction and subsequent ubiquitination and degradation. Also, we have presented new data showing lower ubiquitination levels of PROX1-S79A than WT PROX1 or PROX1-S79E under glucose starvation (Fig. 3m). More importantly, PROX1 ubiquitination was present under normal glucose supply and was increased upon glucose insufficiency (supplemental Fig. 3i). PROX1 phosphorylation has been decreased in cancers (supplemental Fig. 10g and 10h). However, PROX1 ubiquitination is technically difficult to measure in cancers vs non-cancer cells.

Specifically, in relation to AMPK mediated phosphorylation and stability of PROX1, it would be important to assess the half-life (under CHX treatment) of S79A and S79E mutants compared to WT PROX1 under glucose starvation. The analyses in Fig.3j, 3k, 3m and presented in Suppl.Fig.3h, 3i should be quantified and the half-life of PROX1 protein calculated to be able to draw the conclusion on the dynamics of PROX1 protein turnover. Moreover, it is unclear how authors arrived to a conclusion that S79A mutation attenuated interaction with DDB1 (Fig.3g, line 274). The quantification of several independent analyses should be presented. Similarly, the quantification of multiple analyses of protein levels of PROX1 upon knockdown of CUL4-DDB1 should be provided (e.g. Suppl. Fig.3e, Fig. 3n). IF analyses are not adapted for quantitative evaluation of protein levels (Fig.3h, line 283-286).

Response: We appreciate the thoughtful suggestions. We have assessed the half-life of S79A and S79E mutants compared to WT PROX1 under glucose starvation, and shown this results in the revised Fig.3i (replaced the original Fig. 3m). Relevant immunoblots have been quantified, and relative protein abundance from each independent repeat was labeled as indicated in revised Fig. 3 and supplemental Fig. 3. Also, immunoblot in Fig.3g, Suppl. Fig.3e (revised supplemental. Fig.3f) and Fig. 3n derived from each independent repeat has been quantified. As for IF analyses Fig.3h,

we consider it as kind of qualitative analysis, and it is accepted for general qualitative analysis (Hi-Jai R. Shin, Nature, 2016).

Moreover, the analyses of dose response of CUL4-DDB1 overexpression are not clear (Suppl. Fig.3f,3g). To this end, while the Flag-PROX1 levels respond to the overexpression of E3 ligase complex, the overexpression of E3 enzyme seems not gradual. Thus, how to interpret this gradual effect on PROX1 levels? How exogenous PROX1 protein is degraded if AMPK is not activated? The analyses of PROX1 polyubiquitination could be informative in this case.

Response: We apologize for misleading data presentation. For Suppl. Fig.3f (revised Suppl. Fig.3g), band density quantification indeed indicates statistical significance of gradual increase of PROX1. We consider it as technical reasons, such as oversaturated exposure when it comes to Suppl. Fig.3g of original supplemental. Fig. 3. We have provided another immunoblot derived from an independent technical repeat of the original samples in the revised supplemental. Fig.3h, and it is a better representative image.

As far as we know, it is kind of concept that the turnover of certain protein is controlled by multiple post-translation modifications and numerous E3 ligase complexes. In this manuscript, we have found that AMPK-dependent PROX1 phosphorylation enhances its degradation by CUL4-DDB1 E3 ligase complex. It is not clear whether other kinases are potentially involved in PROX1 degradation by CUL4-DDB1 complex and/or other E3 ligase complex. Definitely, PROX1 degradation occurs in part though CUL4-DDB1 although AMPK is not activated. We appreciate the reviewer's insightful comments. Also, we have presented new data showing lower ubiquitination levels of PROX1-S79A than WT PROX1 or PROX1-S79E under glucose starvation (revised Fig. 3m). More importantly, PROX1 ubiquitination was present under normal glucose supply and was increased upon glucose insufficiency (revised supplemental Fig. 3i).

Finally, the heat-map on Fig.3p is derived from analyses on Fig.3o, however it is not clear what tumoral area and how it was quantified. It would be interestingly to discuss what are those HCC tumors in which PROX1 levels are high.

Response: As mentioned in Materials and Methods and also our previous work (5). All IHC staining was independently assessed by two experienced pathologists. The staining intensity was graded from 0 to 2 (0, no staining; 1, weak; 2, strong). The staining extent was graded from 0 to 4 based on the percentage of immunoreactive tumor cells (0%, 1%-5%, 6%-25%, 26%-75%, 76%-100%). A score ranging from 0 to 8 was calculated by multiplying the staining extent score with the staining intensity score, resulting in a low (0-4) level or a high (6-8) level for each sample. Data shown in Fig. 3o is representative images. For those HCC tumors with high PROX1 levels, we previously reported that high PROX1 expression in patients with HCC is associated with unfavorable prognosis, early tumor relapse, angiogenesis and resistance to sorafenib therapy (3, 5, 6). We have not discussed this in this manuscript, due to unavailability of clinical information of TMA used in this study, such as tumor relapse and sorafenib

sensitivity. However, the correlation of prognosis and PROX1 expression in patients with HCC has been analyzed and shown in Fig. 7 and supplemental Fig. 10.

11) The authors employed a murine model of liver specific knockout of PROX1 to perform RNA seq (Fig.4) in order to understand the physiological consequence of AMPK dependent PROX1 phosphorylation. It is not clear how those are related to metabolic rearrangements in cancer under AMPK activation (lung cancer and liver cancer evoked in previous analyses). In the same line, the gene signature and pathway analyses in ATAC-Seq should be discussed (Fig.4j)? How specific it would be to BCAA metabolism? The relevance of these analyses to metabolic adaptation in cancer or under glucose deprivation challenge or under AMPK activation should be explained. To this end, the correlative analyses presented in Fig.4s should be extended to pAMPK as the upstream player of PROX1 in the mechanism that is studied.

Response: Thanks for the reviewer's insightful questions. Absolutely, RNA-seq analysis should be done in the context of AMPK activation or inhibition, such as HCC *de novo* liver cancer model in mice, in which tumor cells are exposed to various stresses such as energy insufficiency and hypoxia. Due to the fact that establishment of liver cancer model needs 24 weeks, we have performed RNA-seq analysis of WT and PROX1-deficient livers to find out some clues before the use of samples derived from liver cancer model in mice. Also, these data were cross-referred to the ChIP-seq datasets publicly available. This analysis was informative, and identified BCAA metabolism and mTOR signaling as potential physiological consequences of PROX1 deficiency. We reasoned that BCAA metabolism and mTOR signaling could link AMPK to PROX1 to interpret what have been done in mechanistical studies and in vitro investigation. Although we have not re-performed RNA-seq analysis using *de novo* tumors tissues in mice that are more relevant to AMPK activation, this model has been used for subsequent verification of BCAA metabolism and mTOR activation by PROX1. In addition, lung cancer model of KL mice, in which LKB1, the upstream kinase of AMPK, is genetically deficient, has been a well-documented model for studies of metabolic incompetence in lung cancers (7). Importantly, it should be noted that data in this manuscript clearly indicates that PROX1 regulates BCAA metabolism regardless of AMPK activation or inhibition. This function is impaired by AMPK-mediated PROX1 phosphorylation, which ultimately alters its protein abundance and activity, or in the context of cancers in which AMPK is activated.

We have ATAC-Seq dataset analyzed for gene signature and pathway enrichment, and indeed BCAA catabolism was enriched. This data has been included in the supplemental Fig. 4f. The clinical relevance of pAMPK and BCAA-related enzymes has been analyzed, and the results have been shown supplemental Fig. 5f.

12) It would be important to at least to discuss what are the mechanisms of differential methylation of H3K4me3, H3K43me and H3K27Ac in PROX1-depleted cells/livers? How selective it is to the genes involved in degradation of BCAA?

Response: Since we have not performed relevant studies with regard to the epigenetic events, we think that PROX1 possibly recruits certain specific histone

methyltransferases or de-methyltransferases to mediate histone modifications and subsequent gene activation or repression. More specifically, PROX1 was previously identified as the component of the Nucleosome Remodeling and Deacetylase (NuRD) complex (8). Possibly, PROX1 acts through NuRD complex that remained to be confirmed. For the concern of selectivity of the epigenetic events to the regulation of BCAA degradation, we consider that the epigenetic tags, such as H3K4me3, H3K43me and H3K27Ac, are not selective to the genes involved in degradation of BCAA. However, certain writers, such as methyltransferases or acetyltransferases, might be specifically recruited to the promoters of genes in BCAA metabolism. We have now provided relevant discussion, please see line 652-664.

13) What is the subcellular localization and chromatin recruitment of PROX1 S79 point mutants especially the resistant to degradation PROX1-S79A? What is their effect on BCAA metabolic gene expression (recruitment and modification in promoters)?

Response: Either WT or PROX1-S79A are nuclear distributed (supplemental Fig.2j). Their effects on BCAA metabolic gene expression have been evaluated by ChIP and qPCR in cells with PROX1-S79A or PROX1-S79E. The data has been included in the revised Fig.4n, 4o and revised supplemental Fig.5e.

14) Given the diurnal control of the AA degradation, authors should specify the time of tissue collection for metabolic and molecular analyses. It is not clear how authors derived the graph on Fig.5c if on graph Fig.5a none of the BCAA reach differences of 50% decrease. The methodology of relative BCAA level calculation should be better described. As previous analyses with the point mutants of PROX1 show that its degradation was promoted by the glucose starvation and AMPK activation, the similar experimental settings should be employed in these analyses.

Response: We thank the reviewer for the detailed comments. We are sure that we collected samples during daytime. For the reviewer's concern, we consider it as technical reasons. Data in Fig.5a was measured by targeted metabolomics using LC-MS. Data in Fig.5c was evaluated by BCAA assay kit (MAK003, sigma). These two different assays might give rise to the similar conclusion, but not always to the same extent. Determination of relative BCAA levels was done as described in the manual provided by the manufacture. Detailed procedure has been provided in the Materials Methods, please see "Determination of BCAA". As requested, we have provided additional data of BCAA levels in PROX1-depleted Huh7 cells with PROX1-S79A or PROX1-S79E expression upon glucose starvation, which was shown in the revised Fig. 5g and 5j.

15) The important analyses in HUH7 cells presented on Fig.5e should be complemented with the controls of PROX1 expression.

Response: We thank the reviewer for the kind reminder. As suggested, we have added that relevant assays are derived from PROX1 stably-expressed cell lines, and immunoblot of PROX1 expression was shown in the revised supplemental Fig.5d.

16) In the metabolic flux analyses, the scale in graphs of Fig.5g and 5i are not clear. Were the values of heavy isotopologues normalized to the total metabolite levels? The glutamine levels should be measured to understand if the uptake is different. The conclusion that authors make about PROX1 required for BCAA production through transamination needs further support regarding the expression of GLS2 and the role of PROX1 in its control. Given that the focus of this study is BCAA degradation upon AMPK activation, the flux analyses should be performed under glucose starvation and KIC, C5-Coa and the TCA intermediates should be analyzed.

Response: Thanks for the comments. The relative labeled metabolites were normalized to the total metabolite levels. Originally, we have had noted the expression of GLS2 in RNA-seq dataset, and indeed, the level of GLS2 was not significantly altered in the Huh7 cells upon PROX1 depletion. Data have now been showed in the revised supplemental Fig.6c. Also, results in Fig.5a, revised Fig.5e and 5f indicate increased of intracellular glutamine levels and reduced glutamine catabolism upon PROX1 depletion. To further support this observation, we have examined glutamine levels in cells during a serial time courses, and no obvious alteration was found. Data has been shown below for your reference. As requested, the flux analyses were performed in cells with PROX1-S79A or PROX1-S79E upon glucose starvation. Data were presented in the revised Fig.5g and Fig.5j.

17) The suggestion that findings of lower BCAA levels in Fig.5g would impact mTORC1 activation need further support. Given that the PROX1-related mechanism will be triggered by active AMPK, the contribution of other AMPK driven mechanisms upstream of mTORC1 should be studied/discussed. Overall, the analyses of TORC1 activation should be quantified (e.g. Fig.5k, 5l, Supp Fig. 6b and 6c) and they should be complemented by analyses of AMPK status.

Response: We have improved this in the discussion in regard to AMPK-mTORC1 crosstalk. These experiments have been repeated additionally, and immunoblots was shown in the revised Fig. 5l and 5m (original Fig.5k and 5l). Also, immunoblots of p-S6K has been quantified, and relative ratio has been labeled as indicated in the revised Fig. 5n and Fig. 5p and 5q (original Supp Fig. 6b and 6c).

18) The description of Fig.5m is missing. It is not clear how the treatment was done and what question this experiment addresses. It would be important to elaborate on how the rescue with exogenous Leu or BCAA would work if it is degradation of BCAA that is activated in cells depleted of PROX1. These metabolite rescue analyses should be complemented by metabolomics analyses.

Response: We apologize for missed data description of Fig.5m (revised Fig. 5n), which was intended to determine whether BCAA could rescue mTOR signaling in Prox1-deficient mice. We have examined BCAA levels of mice livers using a BCAA assay kit. Considering dense data in Fig.5, this result was only shown below for your reference.

19) The link of mTORC1 to glycolysis in the context of “AMPK->hPROX1->PROX1 degradation->increased BCAA catabolism” needs clarification. The experiments presented on Fig.5o-x require controls of treatments (levels of PROX1 expression/knockdown/rescue). The metabolic flux with labelled glucose should be performed as from available in material and methods information, the enzymatic kits that were used are not adapted for measurement of consumption.

Response: As suggested, relevant assays are derived from PROX1 stably-expressed cell lines, Immunoblots of PROX1 expression upon PROX1 depletion or rescue have been included in the revised supplemental Fig.5d, revised Fig. 5p and 5q. As for glucose consumption, the enzymatic kits (Sigma) have been used in many top papers, we assume that the cost of isotope tracing is high, and this might be more suitable for determine where glucose goes.

20) What would be the explanation of increased proliferation in cells expressing S79E PROX1 mutant (Fig.6)? Importantly, the proliferation was assessed by CCK8 assay which is based on the formazan reduction and, given the metabolic changes that occur upon PROX1 downregulation, it is not an adequate approach. Should be supplemented with BrdU labeling and by cell counting.

Response: PROX1 phosphorylation impairs its protein stability and activity compared with that of WT PROX1. We assume that it is reasonable that S79E PROX1 mutant displays increased proliferative activity than con group. As mentioned in response to “Comments 3”, the results of CCK8 assay and BrdU labeling were comparable.

21) The histological images on Fig.6y should be increased in size and the description of the quantification methodology should be detailed. The scale bar should be included. The quantification of lesions needs to be normalized to area of tissue analyzed and the grading of lesions must be explained (Fig.6z).

Response: The lesions were quantified as described previously (9). Briefly, lung metastases were classified into four grades, based on the number of HCC cells in the maximal section of the metastatic lesion. Grade I was defined as having <10 tumor cells; grade II had 10 to 20 tumor cells; grade III had 20 to 50 tumor cells; grade IV had >50 tumor cells. based on the number of HCC cells in the maximal section of the metastatic lesion. We agree with the other reviewer’s suggestion, to avoid dense data

in figures and improve storyline of this manuscript, we have removed irrelevant data (Fig.5u, 5v, 5w, 5y, 5z and supplemental 6h-6J and Fig. 6c, 6d of original figures) related to cell migration in vitro and in vivo, considering that we mainly focus on evaluating the effect on cell proliferative capability by PROX1 phosphorylation.

22) Leu treatment in vivo, needs further controls including the effect on TORC1 signaling, proliferation, metabolite levels. The analyses in Supp Fig 7h-7m are mentioned in one sentence, thus it is difficult to understand what they represent. The metabolic profiling in the tumors should be performed to characterize how PROX1-S79E and metformin treatment affect BCAA metabolism. The phosphorylation of p-Ser79 PROX1 and its total protein levels as well as polyubiquitination should be analyzed in course of tumorigenesis together with status of AMPK to draw clear parallels between these molecular events.

Response: As recommended. IHC staining of p-S6K and Ki-67 has been done to determine TORC1 signaling and proliferative activity during tumorigenesis (revised supplemental Fig. 8b). We have improved data description of original Supp Fig 7h-7m. Since that the effects of PROX1-S79E and AMPK or metformin on BCAA have already been addressed above, it is not essential to try to fit everything into a small space, it might make this manuscript extremely difficult to be readable, considering that it has already been overloaded with data in figures. IHC staining of p-Ser79 PROX1, its total protein levels and p-AMPK have been included in the revised supplemental Fig. 8g. Measuring the levels of PROX1 ubiquitination derived from xenografts are technically difficult and not accurate, due to the absence of mg132 treatment and ubiquitination enrichment. Also, it might be less informative by doing so, because PROX1 ubiquitination ultimately affects its protein abundance. The total levels of PROX1 have been determined and data has been shown in the revised supplemental Fig. 8g.

23) The model sgPROX1 on Fig 6r-u should be introduced. As for any other treatment, the controls of phenformin efficacy and analyses of PROX1 phosphorylation as well as protein levels should be included. In relation to this main figure, the choice of the enzymes in Suppl Fig.8a-d should be explained. The correlation with the TORC1 signaling should be established in the sgPROX1 conditions and in tumors that are characterized by “high” expression of metabolic enzymes in BCAA metabolism.

Response: We feel sorry for confusing data presentation. Experiments in Fig 6r-u (revised Fig. 6n-6r) were performed similarly as it did in Fig.6m-6p (revised Fig. 6i-6l). Briefly, KL mice was given Ad-sgPROX1-Cre infection by nasal inhalation, followed by histological analyses. For space limitation, the model was not included, but it has briefly described in figure legend of the original manuscript. The phenformin efficacy was evaluated by tumor burden, tumor number and Ki-67 reactivity. IHC analyses of PROX1 phosphorylation, PROX1, p-AMPK and p-S6K have been shown in the revised supplemental Fig.9a. The correlation analyses of the levels of p-S6K, PROX1 and enzymes in BCAA metabolism have been shown in the revised supplemental Fig.9a in tumors of sgCon and sgPROX1. For the choice of the enzymes in Suppl Fig.8a-d , these were selected as representative genes of BCAA metabolic pathway with no particular

consideration. As requested by another reviewer, IHC staining of PROX1, p-S6K and several enzymes in BCAA metabolism in sections derived from tumors in original Fig. 6m and 6n should be done and packed into main figure. Fig. 6a and 6b of the original Fig. 6 have been moved to the revised supplemental Fig. 8a, 8c.

24) Given that the authors link the pro-tumorigenic effect of PROX1 expression to repression of BCAA catabolism, it would be important to perform the correlation analyses with TORC1 activation by analyzing its phospho-targets (Fig7).

Response: As requested, this analysis has been done, and data has been shown in the revised Fig. 5o.

25) The schema of the study should be modified to better convey the take home message.

Response: We thank the reviewer for the suggestion. We have done our best to improve our manuscript.

Comments on the text and formatting:

1) The authors are encouraged to revise the English to define the terms and concepts that they use. They also should employ the action verbs that are specific. To this end, avoid the unclear verbs such as “regulate, impact, control” and instead use “activate, repress, increase/decrease” so that the message is evident.

The vague phrasing examples that need definitions in different parts of the text are: broadly coordinate (line 12), essential determinant (line 138), considerably impaired (line 391), aggressive phenotype of Prox1 (line 422); central switch that orchestrates fine-tuning (line 94-95), aberrant BCAA metabolism (line 112), malignant phenotypes (line 113), intrinsic cancer properties (line 114), property of metabolic plasticity (line 131), the metabolic fitness (line 165), BCAA metabolism reprogramming, metabolic vulnerability to therapeutic drugs, reprogramming switch... etc.

The examples of terms that need definition are: metabolic plasticity, tumor transformation, metabolism reprogramming, property of cell metabolic plasticity (line 128), metabolic therapy (line 140), metabolic invulnerability (line 534)

Response: As suggested, we have provided sufficient explanations of the terms and concepts in the introduction section. Although some of them were widely-used in research articles without particular explanations, we feel it would affect the readability of the manuscript if all these terms were included with sufficient explanations, particular in the result section. Therefore, we have removed most of them and replaced with the wording that will be readily intelligible to any scientist.

2) the abbreviations in the title and the abstract should be limited and explained (PROX1, LKB1, BCAA, CUL-DDB1, HCC, AMPK, mTOR). The take-home message of the story should be clarified. In the title, it would be advisable to make clear what is PROX1, what is BCAA and how exactly it is impacted by LKB-AMPK. In abstract, it should be clear what is the cellular function of PROX1, how it is relevant to tumor metabolism in general and BCAA metabolism specifically (current state of the art). Their relevance in

cancer should be stressed.

Response: We appreciate the reviewer's detailed suggestion to improve our manuscript. We have improved the writing of the title and abstract as suggested.

3) The introduction would benefit from including more information on what is known about the role of PROX-1 in tumorigenesis (expand section lines 120-126). The take-home message in the concluding paragraph of the introduction should be edited as it is not clear in its current formulation.

Response: We have improved this as suggested.

4) Figure legends should be revised and when missing added (e.g. Fig4j and 4g). The minimal required information to understand the experimental conditions such as type of treatments, measurements and the type of cell line/tissue should be indicated in the legend.

Response: We have included detailed information in the Figure legends so that it will be easier for the readers to follow.

5) The presentation of histology analyses should be ameliorated as captures are hardly visible. Scale bar and the insets with higher magnification should be added.

Response: We have improved histological images with higher magnifications.

6) As for discussion, the main conclusion that "PROX1 is an essential determinant for LKB-AMPK axis that mediates tumor metabolic plasticity to control therapeutic response (line 536-538)" needs further experimental evidence. To this end no flux metabolomics analyses were performed to define the metabolic rearrangements downstream of activated or inhibited AMPK in cancer cells. The selectivity of this identified mechanism to the cancer context should be established. Authors also write that they have discovered "the molecular mechanism of LKB-AMPK axis mediated BCAA metabolism reprogramming" (line 540). "PROX1 controls the intracellular BCAAs concentrations in an AMPK-dependent manner" (line 554). However, the analyses presented in Fig.5 do not address the role of AMPK in BCAA degradation in cancer context and the authors do not measure the levels of BCAA under conditions of increased or inhibited AMPK. Another conclusion that authors discuss is that "PROX1 deletion in HCC and lung tumor significantly increases expression of the most majority of key enzymes for BCAA metabolism through mediating epigenetic modifications" (line 552-554). This is an overstatement as, first, epigenetics marks were analyzed only in HCC cell model (HUH7 cells). Second, although ATAC-seq in non-tumoral PROX1 LKO mice (Fig.4) showed marked changes in chromatin accessibility (suggestive of active transcription) this was less evident in HUH7 cells depleted of PROX1 (Suppl.Fig.4). As for epigenetic defects in PROX1 depleted cells, those were suggested by ChIP-qPCR in some regions of selected genes with very modest enrichment around 10folds and about 20-30% increase/decrease upon knockdown or knockout of PROX1. Thus, if it is the point that the authors want to stress, they should perform unbiased genomic analyses of epigenetic landscape in PROX1 mutant cells and investigate what is the

functional link between PROX1 and epigenetic control of transcription.

Response: We sincerely appreciate the reviewer for the constructive comments to improve our manuscript. We have addressed these concerns with careful writing related to the results and conclusions in this manuscript.

Reviewer #2 (Remarks to the Author):

In this manuscript, the authors discover PROX1 transcription factor as a substrate of AMPK, and that AMPK-dependent phosphorylation of PROX1 on Ser79 results in its ubiquitin-dependent degradation, which requires the Cul4A/Cul4B/DDB1 complex. The authors go on to demonstrate that PROX1 is suppressing mRNA expression of core branch amino acid metabolic enzymes, and in PROX1 KO cells and tissues, there is a upregulation of those same enzymes at the protein level (ACADSB, ACADM, BCKDHB, etc). IHC on human liver samples supports an inverse relationship between PROX1 levels and levels of these BCAA enzymes. The authors next examine whether the changes in BCAAs is sufficient to alter mTOR signaling, which is known to be very reliant on amino acid levels. siRNA to PROX1 modestly lowers P-S6K and 4ebp1 indicative that mTOR signaling is impacted by the lowered levels of BCAAs. The authors go on to define the impact of sgPROX1 in Kras Lkb1 mutant GEMM models of NSCLC and the impact

of enforced expression of S79A and S79E cDNAs of PROX1 in Kras Lkb1 mutant mice. Finally IHC and comparison of gene expression in human samples make this a very thorough and very compelling study.

The authors have made what appears to be a significant finding in decoding new targets of LKB1 and new regulators of cancer metabolism more broadly. The extensive data across many systems makes the paper almost acceptable in its current form, but there are a few minor experiments the authors could do which would greatly enhance the manuscript and will not require that much effort.

Response: We greatly appreciate the reviewer for his/her positive and enthusiastic comments.

Major points:

1. In Figure 1, the authors use glucose starvation and metformin as AMPK activators and their data is convincing but in the last 5 years there are now a number of selective on-target synthetic AMPK activators that work readily in cells, i.e. 991 or MK8722. The authors should examine whether these direct pure AMPK activators can also trigger PROX1 degradation in cells. They almost certainly will but this would be an important point for the authors to experimentally address.

Response: We thank the reviewer for the thoughtful comments. As recommended, we selected MK8722 (MedChemExpress) to test its effect. As expected, MK8722 treatment triggers PROX1 degradation in Huh7. For space limitation, this result has been shown in revised supplemental Fig.1g and also below for a quick reference.

8

2. The effects of the 79A and 79E mutants used throughout (e.g. across Figure 5) is presumably due to the protein stability of these two PROX1 mutants. However the data on the mutants in Figure 3M is not very convincing there is a major difference, in fact the data in Figure 3N is more convincing. WT vs 79A vs 79E PROX1 levels are needed for interpreting Figure 5E, 5Q, 5R, 5S, 5T, 5U, etc. A western blot of PROX1 levels is needed from the cells used in Figure 5 panels.

Response: We thank the reviewer for the insightful comments. The proteins stability of WT vs 79A vs 79E PROX1 has been determined for many times, as requested by another reviewer, we have assessed the half-life of S79A and S79E mutants compared to WT PROX1 under glucose starvation, and showed this results in the revised Fig.3i (replaced the original Fig. 3m). Also, it should be noted that PROX1 phosphorylation impairs its activity (see Fig. 4 and Fig. 5). Western blot of WT PROX1 and its mutants in Figure 5 had been determined, but not all included in Figures due to dense data presentation. However, these should have been included. Now the immunoblots have been presented in the revised supplemental Fig.5d, 6f and revised Fig. 5p and 5q.

3. IHC for PROX1 or its targets in Figure 6M, 6N (like authors did in Supplemental Figure 8A) would also help explain the behavior of the S79A and S79E alleles in the mouse tumors in vivo. Such IHC should go in the main figure of the paper.

Response: As recommended, we have performed IHC staining of PROX1, p-S6K and several enzymes in BCAA metabolism in sections derived from tumors in original Figure 6M and 6N (revised Fig. 6i and 6j). We agree with the reviewer, and these data have been packed into the revised main fig. 6m, while Fig. 6a and 6b of the original Fig. 6 have been moved to the revised supplemental Fig. 8a, 8c.

4. In Figure 6 regarding the phenformin experiments, while it is correct that sgPROX1 is rendering the tumor cells less sensitive to phenformin, a complicating factor is that the tumors lacking PROX1 are not dividing as much to begin with. I would simply change the wording from “dictates therapeutic response” to “blunts therapeutic response”. There are likely many targets of AMPK that play a role in mediating phenformin response. This data with PROX1 is surprising and exciting but given that the tumors are growing slower and are smaller to begin with, one cannot so directly compare them.

Response: Definitely, we agree with reviewer’s concerns. We have corrected the expression from “dictates therapeutic response” to “blunts therapeutic response” as suggested by reviewer.

Minor points:

1. A lighter exposure of the PROX1 immunoblot in Figure 1M would help make the point more clearly. The AMPK α 1 shRNA did not work very well in this experiment, authors might want to redo the experiment.

Response: Thanks for reviewer's kind comments. This experiment has been reperformed, and new data has been presented in the revised Fig. 1m.

2. In Fig. 3f the HA-Cul4 input is not shown. The figure legend nor the figure panel mention what is HA-tagged.

Response: We thank the reviewer for the critical reading of our manuscript. It was mislabeled with HA. Untagged-DDB1 was cotransfected with FLAG-PROX1. CUL4A and CUL4B are HA-tagged. We have corrected this mistake. Also, we have included description in the figure legend with regard to the vectors used.

Reviewer #3 (Remarks to the Author):

The manuscript by Wang et al. investigates the role of Prospero-related homeobox 1 (PROX1) in coordinating hepatic and lung tumor metabolic plasticity through the modulation of the BCAA metabolism in response to AMPK signaling. The authors first observed that PROX1 levels are decreased due to impaired protein stability in response to glucose starvation, P-AMPK (LKB1-AMPK) or metformin. They went on to show mechanistically that these metabolic perturbations lead to the AMPK-dependant phosphorylation of Serine 79 on PROX1, which targets PROX1 for degradation through the CLU4-DDB1 E3 ubiquitin-ligase complex. The loss of PROX1 transcriptionally modulates the expression of multiple enzymes that belong to various metabolic pathways, including branch chain amino acid metabolism (BCAA). They show that high levels of PROX1 in response to LKB1-AMPK axis increases BCAAs levels in hepatic and lung cancer models, contributing to mTOR activity, cancer progression and resistance to metformin or phenformin treatment and contributes to clinical outcomes.

The data reported in this manuscript is very relevant to the field of tumour metabolism and therapeutics and the conclusions improve the mechanistic understanding of AMPK-dependent metabolism. However, some mechanistic studies lack essential controls, and some conclusions are not fully supported experimentally as currently reported. The data presented also focusses on a specific metabolic aspect of the observed phenotype and would benefit from efficiently integrating the findings with the previously reported metabolic roles of PROX1. The data presented is also very dense and the description of results sometimes lacks precision: the manuscript would benefit from a thorough and more detailed description of the rationales and results. Below are some major and minor points that would need to be addressed to improve these aspects.

Major points

1- The cell and tumour models used throughout the manuscript get interchanged between lung and hepatic tumour models. It should be specified in the text why the focus is put on these two cancer models and the rationale for investigating one or the other at specific instances (do they expect same PROX1 functions if different tumour types?). In addition, 2 different cell lines are used as models for hepatic cancers (none for the lung models) and they are being used alternatively without justifications, with several of the results only reported for one of the two cell lines. It is important that the authors justify reason for switching between models (or cell lines from a same model) and report differences observed between them, as it seems that some differences are indeed observed.

Response: We appreciate the reviewer's positive and insightful comments. Liver cancer is always the focus of our previous and ongoing works (3, 5, 6, 10). This study was initiated in liver cancer, and most of the work in vitro and in vivo has been done in liver cancer models. Considering the frequent somatic mutation of LKB1(15-30%), the upstream kinase of AMPK, in lung cancers, the use of LKB1-deficient KRAS-driven lung cancer model (KL), which defines a lung cancer subtype deficient in sensing metabolic stress, will greatly reinforce the findings and conclusions in this manuscript. Also, it may also establish the connection of PROX1 to the clinical relevance of lung cancer treatment. This KL model is clinically relevant, and is available to us by our collaborator as a kind gift. Therefore, cell-based biochemical and phenotypic studies were done in liver cancer, but not in lung cancer cell lines. KL model was only intended to extend and confirm what we have concluded in liver cancer model. We apologize for misleading schema and storyline of this manuscript, but we will try our best improve this manuscript. For your concerns of inconsistent cell line used, we noted that most of experiments have been done in both two cell lines and also in other cells such as liver cancer cell SMMC-7721 and non-transformed MEFs and 293T cells. For instance, in Figure 1, the effect of glucose starvation on PROX1 expression and cell apoptosis was determined in Huh7, HepG2, MEFs cells and in vivo. Considering the similar effect, we used HepG2 cells for determining PROX1 levels upon AMPK depletion, Huh7 cells following AMPK overexpression, and MEFs with AMPK deficiency (This cell line is a gift by our collaborator). Also, IF analyses were only done in one liver cell line (Huh7) and nontransformed MEFs. The detailed mechanistical study is mainly done in 293T and in vitro (see Figure 2). In Figure 3c and Figure 3d, Co-IP was performed in Huh7 and HepG2 under normal condition and glucose starvation, respectively. In Figure 4 and supplemental Figure 4, RNA-seq analysis, ChIP-qPCR, qPCR and immunoblot verification of genes in BCAA metabolic pathway were done in Huh7. In Figure 5, most of the experiments were done in both cells, except for the flux analyses that were only done in Huh7. Indeed, these two cells are different, and data collected from these two cells of all experiments might be similar. We assume that data from these two are similar, and some of the experiments were not done in one of two cell lines without bias. More importantly, we consider that the data would be doubled if all these experiments were replicated in these two cell lines, considering this manuscript covers a great deal of ground. Lastly, it is not surprising and will be accepted by readers if data from these two cells is different in some of the experiments.

However, by trying to fit everything into a small space, it has made it extremely difficult (and at times impossible) to be confident of many of the conclusions. Honestly, we have no additional data to report the potential difference between these two cell lines. However, we totally agree with the reviewer's concern that we should have done relevant assays in Huh7 and HepG2 throughout the manuscript, and we are happy to perform these assays if it is indeed essential for this manuscript.

2- In figure 3, the authors report that the CUL4-DDB1 E3-ubiquitin ligase complex mediates decrease in PROX1 protein levels/stability upon glucose starvation or AMPK activation. Yet it is not clear whether this complex is expressed/active in all models studied, as they need to ectopically overexpress the complex in Huh7 cells to detect it and to study its effect on PROX1 levels (was the interaction study done in HEK293 cells, it is not specified?). Why not kd the endogenous CUL4-DDB1 in Huh7 cells as they did in HepG2? Is it because it is not expressed endogenously in Huh7 cells? If that is the case, it partly invalidates this mechanism as the main ligase responsible for PROX1 degradation in response to low glucose (since PROX1 decrease upon low glucose is also observed in Huh7 cells). Also, ectopic expression could lead to overexpression of proteins and facilitate non-functional interactions. The levels of ectopic proteins should be quantified, compared between each other and related to endogenous protein levels. Also are endogenous levels of CUL4-DDB1 proteins modulated upon glucose starvation in the different models? The authors should clarify and comment on this.

Response: We thank the reviewer for the comments. It is not clear whether CUL4-DDB1 E3-ubiquitin ligase complex are expressed in all models studied here. We are sure that CUL4 and DDB1 are expressed in 293T, Huh7, HepG2 cells and HCC tissues. Please see immunoblots in Figure 3c and 3d, expression of CUL4A, CUL4B and DDB1 was detectable in 1% input of cell lysate. The interaction was done in 293T (Figure 3a), HuH7(Figure 3c) and HepG2 (Figure 3d). In Figure 3, CUL4-DDB1 is clearly expressed in Huh7 as mentioned above in Figure 3c. Also, we have found that the dominant-negative mutants of CUL4A and CUL4B (dnCUL4A and dnCUL4A) significantly up-regulated PROX1 protein level in the Huh7 cells in the original manuscript Fig. 3b, the dominant-negative mutants of CUL4A and CUL4B have a similar role with the kd the endogenous CUL4 and CUL4B. We totally agree with the reviewer's concern with regard to the potential artificial effect of forced expression of CUL4 and DDB1 to facilitate non-functional interactions. So, we have repeated the interaction experiment, and found that PROX1 has interaction with CUL4A, CUL4B and DDB1, respectively, but not bound to CUL1, CUL2, CUL3 and CUL5 in the Huh7 cells (revised Fig. 3c and Supplementary Fig. 3c). For your concern of CUL4-DDB1 levels upon glucose starvation, we actually had noted the potential effect of metabolic stress or AMPK on CUL4 and DDB1 expression, yet data in our hands indicates no significant change (data not shown). Also, please see immunoblots in Figure 3d.

3- None of the western blots is quantified and none of them shows molecular weight markers. This should be corrected. Many conclusions are drawn from small changes in band intensity that are not always obvious (eg. Fig.3h; 5k-l-m; 6b-c; Supp Fig.3e, etc).

Response: As recommended, the immunoblots have now quantified, and relative band density and molecular weight have been labeled as indicated. Data as mentioned has been repeated for at least three times. Although data in Fig.3h appears moderately altered, this conclusion has been supported by data of immunoblot. Data in Fig. 5m and Supp Fig.3e were quantified in the revised Fig. 5n and revised Supp Fig.3f, respectively. For immunoblots in Fig. 5k-l, we have more representative data from independent repeat. As requested by another reviewer of the manuscript, data here should include p-AMPK, so these experiments have been independently repeated for three times. New data from one repeat has been shown in the revised Fig. 5l, 5m, 5p and 5q.

4- Confirmations of KD or KO should be shown for each si-, sh- and sg-RNA used. Eg. Figure 1j and k first introduce PROX1 kd but no confirmation that the protein is actually depleted is shown at that point. (shown later in the manuscript for some models, but not for all the lines, etc). The same is expected for all proteins that are depleted or deleted.

Response: As suggested, we have included all data required for confirmations of KD or KO efficiency in the revised Figure and Supplementary Figure.

5- It is not shown in figure 2o whether glucose starvation induces S79 phosphorylation on endogenous PROX1 in MEF (as it is an IP) so the statement on p, 9 line 13 is not supported by figure 2o. To support this statement, figure 2p (blot for p-S79-Prox1) should be repeated including the mice treated conditions from figure 1h (fasted and metformin treated). In figure 2p, IHC for AMPK-P and AMPK and PROX1 total should be shown as well.

Response: As recommended, for Figure 2p, the IHC staining for p-AMPK, AMPK and PROX1 have been shown in Fig. 1r and revised supplemental Fig.1m. Also, p-S79-Prox1 levels of liver sections in mice with or without fast or metformin treatment have been shown in the revised Fig. 2q.

6- In Figure 3c, blotting for other CULs factors (eg CUL5) should be included as it was done in Figure 3b to show these are specific interactions.

Response: This experiment has been repeated as requested, and shown in the revised Fig.3c and supplemental Fig.3c.

7- The co-localization experiments only show that these proteins are located in the same organelles and should not be used to conclude endogenous interactions (the text implies that).

Response: We have corrected the inappropriate wording to “colocalization”.

8- In Figure 3c, the DDB1 levels are affected (decreased) by glucose starvation. How does this affect the role of this complex in regulating PROX1? In Figure 3b is this at basal levels? What about in low glucose? In figure 3f, Why blot DDB1 in input and not only HA as it is done in the supplemental? Is this a mistake?

Response: We feel a little bit confused about your concern that DDB1 levels are affected (decreased) by glucose starvation in Figure 3c? Did you mean the immunoblot in Figure 3d? If it is in this case, immunoblot in Fig.3c showed the basal level of DDB1. It appears increased, but this observation cannot be replicated under glucose starvation. Data showed in Fig.3b was determined at basal levels. We have done additional experiment in Huh cells with DDB1 depletion at basal condition or upon glucose starvation. The result was showed below for your reference, which indicates that PROX1 was increased in cells with DDB1 depletion with or without glucose starvation compared with Control group. Fig. 3f was mislabeled with HA. Untagged-DDB1 was cotransfected with FLAG-PROX1. CUL4A and CUL4B are HA-tagged. We have corrected this mistake.

9- In figure 4, there are a few issues with the transcriptional/epigenetic analyses:

- a different set of genes regulated by PROX1 is shown for each model used. The same list of genes should be used in all the models to show expression.

Response: Relevant genes were selected based on the RNA-seq datasets derived from WT vs PROX1-deficient livers in mice and Control vs PROX1-depleted Huh7 cells. Indeed, the differential-expressed genes involved in BCAA metabolic pathway of these two datasets were not exactly the same, we thus selected those with significant changes of each dataset for further verification by qPCR.

- The GSEA result implies downregulation of the BCAA pathway in PROX1 KO condition (4c) but the heatmap (4b), figure 4d and qPCR validations (4e-f-g, etc) and epigenetic data imply that low PROX1 upregulates a subset of these genes. Is the NES really negative in 4c?

Response: It is indeed negative as indicated which means upregulation in the KO group. We have shown original GSEA Results Summary below for your reference.

Dataset	WT-KO - mouse_collapsed_to_symbols.WT-KO - mouse.cls#WT-mouse_versus_KO-mouse
Phenotype	WT-KO - mouse.cls#WT-mouse_versus_KO-mouse
Upregulated in class	KO-mouse
GeneSet	KEGG_VALINE_LEUCINE_AND_ISOLEUCINE_DEGRADATION
Enrichment Score (ES)	-0.5968905
Normalized Enrichment Score (NES)	-2.5061264
Nominal p-value	0.0
FDR q-value	0.0
FWER p-Value	0.0

- Are these transcriptional/epigenetic reprogramming of BCAA genes modulated by AMPK itself? It is not shown directly but is implied in the text and should be

experimentally demonstrated in this figure.

Response: Indirect data can be seen in Fig. 4d, 4f and 4g. We have provided direct qPCR data of Huh7 cells upon AMPK depletion. This data has been included in the revised supplemental Fig. 4d.

- What is going on with BCAT1-2 (Supp Fig4c and Figure 4s)? These genes show a different trend than other genes (downregulated upon loss of PROX1). What is the consequence of this different trend as they are the enzymes that initiate the catabolism of BCAA? This should be shown in main figure and discussed.

Response: As the reviewer commented below, BCAT1/2-mediated BCAA catabolism is reversible. Detailed explanations can be seen below. It would be packed in main figure, yet there is no enough space, even if Fig. 4f and 4g were moved to supplemental figures. Also, the whole Supp Fig. 4 is to focus of BCAA metabolism in Huh7 cells.

10- In figure 4j, include stats about differences in accessible chromatin between WT and Prox1 KO for upper panels. Importantly, the authors should show a similar panel restricting the analysis only to the ATAC results at the promoters of the BCAA genes that they study. The genomic representations shown in figure 4 and in supplemental 4k do not convincingly show that chromatin is “more opened” at promoters of genes coding for BCAA enzymes. Do the sites that show increased opening in figure 4j actually include BCAA genes?

Response: We have ATAC-Seq dataset analyzed for gene signature and pathway enrichment, and indeed BCAA catabolism was enriched. This data has been included in the revised supplemental Fig. 4f.

11- In figure 5, can the author better describe why the BCAA-related metabolites are decreased upon loss of PROX1 while most genes regulating them are upregulated? Could this be linked to the trend of BCAT1-2? Is it because the BCAT enzymes catabolism deplete BCAA-derived metabolites? The authors should discuss this. Few points that could help this discussion: In 5f-1: Decreased incorporation of labelled Gln into BCAA but increased KIC from labelled leucine. Blots or RNA should be provided to show GLS and BCAT1/2 levels upon PROX1 inhibition to better understand this result. BCAT1/2 is bidirectional but the breakdown of BCAA to KIC seems to be preferred upon PROX1 ablation (according to the data 5f-i)? Is PROX1 acting to increase intracellular BCAA levels due to inhibited BCAA catabolism activity but also due to increasing BCAA synthesis? (From 5f-g?)? In Fig 5I-M: Are extracellular BCAAs contributing to mTORC1 activation? Could they be taken up by LAT1? Gln level shows opposite trend to Leu level in Fig 5A, which could suggest LAT1 activity upon modulation of PROX1?

Response: We thank the reviewer for the comments. Loss of PROX1, on one hand, releases the repression of genes in BCAA catabolism. On the other hand, PROX1 depletion results downregulation of certain genes like BCAT1-2 and suppresses BCAA synthesis. We have discussed this in Discussion section, please see line 643-652. We have examined GLS, GLS, SLC7A5 and BCAT1/2 levels in Huh7 cells upon PROX1 depletion. Data has been shown in the supplemental Fig. 4c and the revised

supplemental Fig.6c. Absolutely, we agree with the reviewer's comments that PROX1 acting to increase intracellular BCAA levels due to inhibited BCAA catabolism activity but also due to increasing BCAA synthesis. We have examined LAT1 expression in cells upon PROX1 ablation, and no alteration was found (revised supplemental Fig.6c). We thought that increased Gln level in cells with PROX1 depletion was due to Gln catabolism to BCAA as Fig.5f-g indicated.

12- The conclusion of Figure 5 does not seem to be fully supported: "PROX1 activates the mTOR pathway via BCAA to promote tumour progression". It is clear that altered PROX1 levels modulates mTOR signalling (which occurs at least partly through modulation of BCAA levels) but the link that this is responsible for tumour aggressiveness is not clear. Supplemental Fig 6d is the only figure actually testing mTOR modulation. However its conclusion is not convincing: "rapamycin, a mTOR inhibitor, considerably reversed Huh7 cell death under glucose-deprived conditions, an effect comparable to that of cells with PROX1 depletion (Supplementary Fig. 6d), suggesting that inactivation of mTOR by AMPK-PROX1 axis is critical for tumor cells to establish metabolic adaptation" the fact that it is "comparable" to PROX1 depletion does not mean that the effect of PROX1 on cell growth is actually going through mTOR inhibition. Also, how does rapamycin affect PROX1 levels? What is the status of PROX1, pS6 and p4EBP upon glucose starvation? Glucose starvation should decrease PROX1 (and mTOR?) so then why further inhibiting them prevents the apoptosis? This part is confusing (more rationale needed?). Maybe an approach using ectopic expression of PROX1 in PROX1KO cells and rapamycin treatment would better support this? (blots for pS6 should be quantified, eg. effect in 5l is not clear). Also, 5X-Z: Staining of pS6 (on top of Ki67 in 56M) on these tumours would better link the aggressiveness with PROX-1 dependent mTORC1 activation.

Response: We appreciate the reviewer's insightful comment and constructive suggestion to improve the understanding of this manuscript. Since inefficient inhibition of mTOR signaling and anabolism upon glucose starvation, cells undergo apoptosis. Suppression of mTOR by rapamycin efficiently shut down cellular anabolism and rescues cell apoptosis under glucose starvation. It is reasonable and is reported previously (11). The ability to adapt to energetic stress (the ability to shut down mTOR signaling) differs in cancers and cancer subtypes, and is compromised in certain subtype of cancers, such as LKB1-mutant lung cancer, which is vulnerable to energetic stress. We have provided additional data as suggested in the revised supplemental Fig.6j, which indicates that re-expression of PROX1 renders PROX1-depleted cells sensitive to glucose starvation, and was further compromised upon rapamycin treatment. Also, we have included IHC staining of p-S6K in supplemental Fig.7e, 9a and revised Fig. 6m as the reviewer requested.

13- In figure 6k-l, the PROX1 KD already has a very significant effect on tumor growth (orange and green curves in 5k) so it is very difficult to conclude whether loss of PROX1 makes HCC cells resistant to metformin - can the growth be even further decreased? Same observations in r-s-t-u. Can the authors show that PROX1 is necessary for the

metformin effect instead?

Response: Indeed, PROX1 depletion markedly reduced tumor cell growth. However, to figure out whether PROX1 is linked to metformin response, we feel that it is reasonable and commonly recognized to evaluate tumor reduction in PROX1-depleted cells with metformin treatment relative to those without metformin treatment. To our best knowledges, cellular sensitivity to metformin is not roughly affected by smaller tumor sizes or burdens which have been already suppressed by the presence of certain oncogenes. Instead, take TET2 for instance, expression of TET2 in A2058 cells dramatically repressed tumor xenograft growth while a further tumor reduction was observed in TET2-expressed cells upon metformin treatment (12). Similar observation was seen in cells with HK depletion as described (13). In our viewpoint, PROX1 is linked to mTOR signaling that is correlated with energetic stresses, such as metformin or phenformin. To further support this, we actually have already done the cellular response of WT, S79A and S79E to metformin in nude mice, and have not shown in Fig. 5x due to space limitation. We have done two separate sets of tumor xenograft model with control, WT, S79A and S79E with or without metformin treatment. We have now included the whole figure below for your reference, and we will consider whether it is needed to be included in Fig.5r of the revised Fig.5 as the reviewer will suggest. Actually, in the revised supplemental Fig. 8i-8n of original supplemental Fig.7, metformin responsiveness in cells with WT and S79E has been done both in vitro and in vivo. We think these parts of data together with the de novo experiments are generally sufficient to support the conclusions.

- In the same line of thoughts, metformin decreases PROX1 as shown in figure 1, so does this imply that the loss of PROX1 is necessary for metformin effect on mTOR and tumor growth? This is not clearly shown here but should be the purpose of these experiments. Can they re-express PROX1 as rescue upon metformin treatment? Or use the S79A mutant to show this?

Response: In fact, it is the effect on mTOR activation by PROX1 that decides the cellular sensitivity to metformin or phenformin. We actually have already done the cellular response of WT, S79A and S79E to metformin in nude mice, and have not shown in the original Fig. 5x (revised Fig.5r) due to space limitation. We have done two separate sets of tumor xenograft model with control, WT, S79A and S79E with or without metformin treatment. We have now included the whole figure for your reference (upper), and we will consider whether it is needed to be included in Fig.5r of the revised Fig.5 as the reviewer will suggest. Actually, in the revised supplemental Fig. 8i-8n of original Fig.7, metformin responsiveness in cells with WT and S79E has been

done both in vitro and in vivo. We think these parts of data together with the de novo experiments are generally sufficient to support the conclusions.

14- Regarding the previously reported role of PROX1 as a transcriptional co-repressor interacting with ERR-PGC1 axis (in bioenergetic metabolism and circadian clock), or NuRD complex (to repress Notch pathway) and other reported mechanisms, can the authors discuss whether these TFs might be involved in mediating the PROX1 transcriptional activity on BCAA metabolic enzymes? Is the effect directly due to PROX1 recruitment to the BCAA gene promoters or could this happen indirectly as well? Can the effect on mTOR be linked to the PROX1-ERR-PGC1 axis through modulation of bioenergetic capacity? Likewise, could other metabolic pathways/functions altered by the loss of PROX1 (eg fatty acid metabolism, glutathione, peroxisome, etc as shown in figure 4a) be affected by the AMPK-PROX1 axis and contribute to the PROX1-mTOR effect? Better integration of previously described PROX1 roles should be discussed.

Response: We thank the reviewer for the very helpful suggestion and discussion. Probably, PROX1 works with other transcriptional factors that remains to be identified to modulate BCAA genes and mTOR signaling. We have discussed ERR-PGC1 axis and other potential mechanisms related to PROX1-mediated BCAA gene expression, please see line 664-673 in the Discussion section.

Minor points

1- Some parts of intro would benefit from clarifications or rephrasing:

- Introduction line 21: “(...) imposing a metabolic invulnerability to therapeutic drugs”. Do they mean resistance to therapeutic drugs?

Response: We aim to express the meaning “cells that are metabolically invulnerable” or show metabolic invulnerability. Now it seems confusing, and we have corrected it to “resistance to therapeutic drugs”.

- Also in introduction, the part “we uncovered that PROX1 phosphorylation by AMPK drove BCAA metabolism reprogramming to suppress mTOR signaling activity and overall energy metabolism” is also confusing as it suggests that PROX1 activity suppresses mTOR activity (we don't know yet that phosphorylation of PROX1 targets it for degradation). However, mTOR activity is known to be linked to drug resistance in several tumor types, which makes the rest of the sentence about invulnerability to therapeutic drugs confusing.

Response: We have changed this to “we uncovered that PROX1 phosphorylation by AMPK drove BCAA metabolism reprogramming to suppress mTOR signaling activity and overall energy metabolism”

2- The cell models used for some experiments are not systematically reported (eg. Co-IP-MS in figure 3, etc). This should be corrected throughout the manuscript.

Response: We have corrected this as indicated.

3- Mistake in Figure 2f? Where is AMPK? Is it FLAG AMPKa2? Or blotted AMPK? The

text refer to AMPK2alpha.

Response: We apologize for confusing label. It was FLAG-AMPK α 2, we have corrected this mistake.

4- In Supp Figure 2f, DAPI imaging is not scaled according to the other images. Include the right image.

Response: We have included the right image.

5- The expression “therapeutic drugs” should be changed to refer to metformin or phenformin throughout the manuscript.

Response: We have changed “therapeutic drugs” to “metformin” or “phenformin” throughout the manuscript.

6- p6 line 3, define TMT (undefined acronym).

Response: We have defined the abbreviation of TMT in the text.

7- In figure 4a, the reference for PROX1 ChIP-seq is Charest-Marcotte et al., 2010 and not “Alexis et al., 2011”. Also Fatty acid and not “Fathy acid”.

Response: We have corrected this mistake and typo as well.

8- Gene names when referring to mouse models should not be all capitalized. Change in Supplementary Figure 7B and supplementary Figure 8A.

Response: Thank the reviewer for pointing out the mistakes. We have now corrected these mistakes.

9- Wording for some conclusions is very strong (eg. “drive” and “essential event”). Eg. in “Collectively, these findings demonstrate that downregulation of PROX1 by AMPK is an essential event for tumor metabolic adaptations during nutrient deficiency.” This figure suggests that increase in PROX1 contributes in part to the effect of decreased AMPK on survival and apoptosis as kd of PROX does not completely rescue this effect. AMPK likely signals through other factors that also contribute to the survival and this should be nuanced in the conclusion/text.

Response: We have improved the wording throughout the manuscript.

10- Supplementary Figure 2g is important and links figure 2 to figure 3: it should be in the main figures and part of Figure 3.

Response: We have moved this figure into Figure 2.

11- In all the GSEA plots, the p-values should not be 0. Actual adjusted p-value should be reported.

Response: We have checked GSEA result and also shown the FDR q-value together.

12- in figure 4j: the legend does not report “j” panel; mistaken with “g”?

Response: It was mistakenly labeled with “g”. We have corrected it to “j”.

13- In Figure 5 c-d, what does BCAA means in the graph axis? Does it mean all BCAAs compiled together? That should be detailed.

Response: As described in the kit (MAK003, sigma), it is suitable for the colorimetric detection of branched-chain amino acids, leucine, isoleucine, and valine. Yet the reference standard was leucine standard. We assume that it means all BCAAs. Giving that this kit was widely used in several literatures of top journals (14, 15), we used it as well. It is not mentioned in these publications whether the values represent all BCAAs or leucine only.

14- Figure 5e should show levels of PROX1 (ectopic) as they state that they reached a level equivalent to endogenous but it is not shown.

Response: It have now been included in the revised supplementary Fig.5d as requested.

15- On p.6 line 26-27: the authors state that “Given the role of PROX1 in AMPK-mediated metabolic plasticity (...)”. The authors have not shown a role of PROX1 on metabolic “plasticity” yet so it is not clear what they are referring to. Rephrase?

Response: According to the definition of “metabolic plasticity”, we assume that PROX1 play a role in “metabolic plasticity” as metabolic fitness ultimately impacts cellular survival. Indeed, we have not shown the role of PRXO1 in mediating metabolic reprogramming upon energetic stress, we have rephrased this statement to “Given the role of PROX1 in AMPK-mediated cellular survival upon energetic stress”

16- in general, figures are very dense and convey multiple messages (eg. Figure 5 starts with BCAA metabolomics and ends with pS79 effect on migration, invasion and tumor volume (and going though glucose consumption/mTOR activation). The link between these multiple messages is not always clear or supported and figures could be split (or re-ordered) to better convey the main conclusions. In line with this, better explanations of the large variety of results obtained is needed.

Response: We totally agree with the reviewer’s comments. We have improved the schema, reordered the figures and removed some of non-essential data. Detailed explanations have been included above. We sincerely appreciate the reviewer for your efforts to improve our manuscript.

1. Martinez-Reyes I, Chandel NS. Cancer metabolism: looking forward. *Nat Rev Cancer* 2021;21:669–680.
2. Cairns RA, Harris IS, Mak TW. Regulation of cancer cell metabolism. *Nat Rev Cancer* 2011;11:85–95.
3. Liu Y, Ye X, Zhang JB, Ouyang H, Shen Z, Wu Y, Wang W, et al. PROX1 promotes hepatocellular carcinoma proliferation and sorafenib resistance by enhancing beta-catenin expression and nuclear translocation. *Oncogene* 2015;34:5524–5535.
4. Xie X, Hu H, Tong X, Li L, Liu X, Chen M, Yuan H, et al. The mTOR–S6K pathway links growth signalling to DNA damage response by targeting RNF168. *Nat Cell Biol* 2018;20:320–331.
5. Liu Y, Zhang JB, Qin Y, Wang W, Wei L, Teng Y, Guo L, et al. PROX1 promotes hepatocellular carcinoma metastasis by way of up-regulating hypoxia-inducible factor 1alpha expression and protein stability. *Hepatology* 2013;58:692–705.
6. Liu Y, Zhang Y, Wang S, Dong QZ, Shen Z, Wang W, Tao S, et al. Prospero-related homeobox 1 drives angiogenesis of hepatocellular carcinoma through selectively activating interleukin-8 expression. *Hepatology* 2017;66:1894–1909.
7. Ji H, Ramsey MR, Hayes DN, Fan C, McNamara K, Kozlowski P, Torrice C, et al. LKB1 modulates lung cancer differentiation and metastasis. *Nature* 2007;448:807–810.
8. Hogstrom J, Heino S, Kallio P, Lahde M, Leppanen VM, Balboa D, Wiener Z, et al. Transcription Factor PROX1 Suppresses Notch Pathway Activation via the Nucleosome Remodeling and Deacetylase Complex in Colorectal Cancer Stem-like Cells. *Cancer Res*

2018;78:5820-5832.

9. Lu M, Zhu WW, Wang X, Tang JJ, Zhang KL, Yu GY, Shao WQ, et al. ACOT12-Dependent Alteration of Acetyl-CoA Drives Hepatocellular Carcinoma Metastasis by Epigenetic Induction of Epithelial-Mesenchymal Transition. *Cell Metab* 2019;29:886-900 e885.
10. Liu Y, Tao S, Liao L, Li Y, Li H, Li Z, Lin L, et al. TRIM25 promotes the cell survival and growth of hepatocellular carcinoma through targeting Keap1-Nrf2 pathway. *Nat Commun* 2020;11:348.
11. Rowe I, Chiaravalli M, Mannella V, Ulisse V, Quilici G, Pema M, Song XW, et al. Defective glucose metabolism in polycystic kidney disease identifies a new therapeutic strategy. *Nat Med* 2013;19:488-493.
12. Wu D, Hu D, Chen H, Shi G, Fetahu IS, Wu F, Rabidou K, et al. Glucose-regulated phosphorylation of TET2 by AMPK reveals a pathway linking diabetes to cancer. *Nature* 2018;559:637-641.
13. DeWaal D, Nogueira V, Terry AR, Patra KC, Jeon SM, Guzman G, Au J, et al. Hexokinase-2 depletion inhibits glycolysis and induces oxidative phosphorylation in hepatocellular carcinoma and sensitizes to metformin. *Nat Commun* 2018;9:446.
14. Li JT, Yin M, Wang D, Wang J, Lei MZ, Zhang Y, Liu Y, et al. BCAT2-mediated BCAA catabolism is critical for development of pancreatic ductal adenocarcinoma. *Nature Cell Biology* 2020;22:167-+.
15. Han J, Li E, Chen L, Zhang Y, Wei F, Liu J, Deng H, et al. The CREB coactivator CRTC2 controls hepatic lipid metabolism by regulating SREBP1. *Nature* 2015;524:243-246.

REVIEWER COMMENTS

Reviewer #1 (Remarks to the Author):

I would like to thank authors for an extensive work that they have conducted in the revision process. I also take a note of their efforts to streamline the presentation of their interesting story. However, I still find the manuscript massively loaded with data and lacking of detailed explanations in the text. I believe the text editing could be still improved to promote use of specific verbs and to avoid overstatements. There are still many instances when the non-specific verbs are used. Although suggested in previous review, the abstract is largely unchanged with just 2 abbreviations explained. The same for the title which still does not convey a specific message. One of suggestions could be: AMPK phosphorylates transcriptional repressor PROX1 to activate Branched amino acid degradation in cancer. Throughout the text there is lack of details on how the analyses were conducted and the figure legends contain little information.

Another major concern is inadequate quantification of the data presented in WBs (and other analyses with the quantitative conclusions, especially when authors mention "significantly"). The quantifications should represent the average of the biological replicates with the statistical analyses conducted on pooled data and not just numbers on WB panels of one band quantification. The examples are the conclusions on half-life of PROX1 or generally changes of PROX1 protein levels in WBs or IF intensity. In my opinion, the argument that there are already too many data, so it is difficult to put more graphs is not relevant because a lot of representative data does not equal to fewer quantified observations. The authors must make a choice of what data are essential for conclusions and remove all other panels to free the space required for presenting quantifications. The same relates to histological images, as they are small and the structures on them hardly visible. We basically have to take the word of authors on what they conclude and it should not be the case. One of the examples of need for quantification and editing to avoid the overstatement is description of Fig.5p and 5q. Although in the text the authors claim that "In keeping with the existence of AMPK-PROX1 axis, S79A, but not S79E mutant, stimulated mTOR activation, indicating that such a phosphorylation tag caused compromised PROX1 protein activity and subsequent mTOR signaling activation (Fig. 5p and 5q)." the data do not support this statement as the OE of S79E mutant also activated mTORC1 signaling judging from pS6K WB panel (line 449-452).

The explanations in the rebuttal letter on why some of the experiments cannot be conducted are not fully clear as for example, the authors mention that they cannot investigate in vitro Ubiquitination of PROX1 because the recombinant protein is not available, yet they produced it for interaction assay by translation in vitro (line 211-212).

Moreover, I still do not understand what is presented on Fig5e-j for metabolic flux. The figure legend (and the explanation in rebuttal letter) does not provide the methodological details (line1232-1240): h, i Schematic of isotope tracing in the Huh7 cells were traced 24h with [13C]-Leu (h), followed by LC/MS analysis of the labeled metabolites (i)(n=4).

Finally, given the omnipresent through this study claim of PROX1 "driving metabolic plasticity" in cancer

cells the authors should either tune down this claim or provide biochemical evidence of the plasticity in form of metabolic flux with labeled Glu in cancer cells devoid of Prox1/glucose starvation/under metformin treatment.

Reviewer #2 (Remarks to the Author):

In this revised manuscript by Wang and colleagues, the authors have addressed all of this reviewers concerns. New data and new controls for PROX1 IHC in lung tumors and use of modern AMPK agonists like MK8722 enhance and clarify the authors findings. Manuscript has been heavily edited in response to other reviewers, and should be acceptable in its current form for publication in Nature Communications in this reviewer's opinion.

Reviewer #3 (Remarks to the Author):

Report for revised manuscript by Wang et al., Prospero-related homeobox protein 1 orchestrates tumor metabolic plasticity by integrating AMPK signaling and BCAA catabolism for Nat Commun.

The authors have addressed most of the major comments that were raised at the initial submission, and they included a considerable amount of new data addressing the comments in this revised version of their manuscript. I believe that the following comments were not properly or fully addressed in the rebuttal and would benefit from additional clarifications and adjustments. Some of these points may be addressed and explained in the main text.

Point #1: I agree that using the various liver cancer cell lines as well as the KRAS driven LKB1 deficient lung cancer models are all interesting (and available) models, but I still don't understand why the authors go back and forth using different cell models for different experiments. While a large part of the data was carried out in the Huh7 cell line, it is known that different cancer cells display different basal metabolic profiles. Given the conclusion of this study, this should be an incentive to show whether conclusions drawn from this study may apply to cancer cells with a specific metabolic profile or can be generalized. Validation of the main findings in a second cell model should not be omitted simply because it would "double" all the experiments. The authors do not need to redo all the experiments in all the cell lines, but they should show some consistency and validate main findings in other models (for eg in figure 4, qPCR and immunoblots could be validated in additional models, no need to redo the RNA-seq). Whenever this is not possible, the use of one specific model for a specific experiment should be justified in the text.

Point #9:

- I am still confused with the GSEA plot for the PROX1 KO (4b and S4b). The figure legend of 4b is incomplete: does not indicate the system used (liver KO, cell models?). The GSEA plot is opposite than

what is shown in all other panels (ie. BCAA metabolic gene expression increases upon PROX1 KO). Please include the full GSEA graphs (with labeling of samples to avoid confusion). How do the authors explain the negative NES of the GSEA plot in 4c? The table provided in the rebuttal tend to show that order of samples in 4c might be mislabeled.

- Evidence for the regulation of the BCAA metabolic genes by AMPK has now been shown by the authors (S4d) but should be included in the main text. This is an important data showing the role of AMPK regulation in this BCAA effect (via prox1 or independently).
- The reasons and consequences of different trend observed for BCAT1 should be discussed.

Point #13: I believe that the figure presented in the rebuttal (of tumor xenograft model with control, WT, S79A and S79E with or without metformin treatment) should be included in the main text as it supports the observations about the role of PROX1 with metformin and it is more complete than Figure S8n.

Response to reviewers:

We thank the reviewers for their interests in our work and their insightful and constructive comments and criticisms. Accordingly, we have performed new experiments and extensively revised the manuscript. We have also expanded the Discussion sections. Please find below our responses to all comments and criticisms. The data generated during the revision are highlighted in red color in the figures and the text.

REVIEWER COMMENTS

Reviewer #1 (Remarks to the Author):

I would like to thank authors for an extensive work that they have conducted in the revision process. I also take a note of their efforts to streamline the presentation of their interesting story. However, I still find the manuscript massively loaded with data and lacking of detailed explanations in the text. I believe the text editing could be still improved to promote use of specific verbs and to avoid overstatements. There are still many instances when the non-specific verbs are used. Although suggested in previous review, the abstract is largely unchanged with just 2 abbreviations explained. The same for the title which still does not convey a specific message. One of suggestions could be: AMPK phosphorylates transcriptional repressor PROX1 to activate Branched amino acid degradation in cancer. Throughout the text there is lack of details on how the analyses were conducted and the figure legends contain little information.

Response: We are grateful to this reviewer for taking the time to read our manuscript and give us very helpful comments. As suggested, we have extensively revised the manuscript, including title, abstract, figure legends, the use of specific verbs and so forth.

Another major concern is inadequate quantification of the data presented in WBs (and other analyses with the quantitative conclusions, especially when authors mention "significantly"). The quantifications should represent the average of the biological replicates with the statistical analyses conducted on pooled data and not just numbers on WB panels of one band quantification. The examples are the conclusions on half-life of PROX1 or generally changes of PROX1 protein levels in WBs or IF intensity. In my opinion, the argument that there are already too many data, so it is difficult to put more graphs is not relevant because a lot of representative data does not equal to fewer quantified observations. The authors must make a choice of what data are essential for conclusions and remove all other panels to free the space required for presenting quantifications. The same relates to histological images, as they are small and the structures on them hardly visible. We basically have to take the word of authors on what they conclude and it should not be the case. One of the examples of need for quantification and editing to avoid the overstatement is description of Fig.5p and 5q. Although in the text the authors claim that "In keeping with the existence of AMPK-PROX1 axis, S79A, but not S79E mutant, stimulated mTOR activation, indicating that

such a phosphorylation tag caused compromised PROX1 protein activity and subsequent mTOR signaling activation (Fig. 5p and 5q).” the data do not support this statement as the OE of S79E mutant also activated mTORC1 signaling judging from pS6K WB panel (line 449-452).

Response: We thank the reviewer for the comments. As suggested, we have performed the quantification on half-life of PROX1 or generally changes of PROX1 protein levels in WBs and histological images. We have also quantified the results of Fig.5p and 5q. Indeed, these results indicate that S79E mutant compromised PROX1 function for mTOR signaling activation compared with the wide-type of PROX1 and S79A mutant. In order to avoid misleading, we have also revised the description of these results in the manuscript.

The explanations in the rebuttal letter on why some of the experiments cannot be conducted are not fully clear as for example, the authors mention that they cannot investigate in vitro Ubiquitination of PROX1 because the recombinant protein is not available, yet they produced it for interaction assay by translation in vitro (line 211-212).

Response: We apologize for this confusing explanation. CUL4 uses DDB1 as a linker to interact with a subset of WD40 proteins that serve as substrate receptors, forming as many as 90 E3 complexes in mammals. Currently, we did not identify the substrate receptor of CUL4-DDB1E3 ubiquitin complex for PROX1. Therefore, there is no currently suitable way to perform a meaningful in vitro Ub assay. Absolutely, we appreciate this constructive comment from biochemical perspective which is a very interesting topic for study in future. We have also discussed this in the revised Discussion section, please see line 636-638.

Moreover, I still do not understand what is presented on Fig5e-j for metabolic flux. The figure legend (and the explanation in rebuttal letter) does not provide the methodological details (line1232-1240): h, i Schematic of isotope tracing in the Huh7 cells were traced 24h with [¹³C]-Leu (h), followed by LC/MS analysis of the labeled metabolites (i)(n=4).

Response: We apologize for the confusion. The relative labeled metabolites were calculated using the ration of the counts of labeled and the total of each metabolite. For example, the relative level of [¹⁵N]-Leu (M+1) labeled is the count of [¹⁵N]-Leu (M+1) versus ([¹⁵N]-Leu (M+1) + [¹⁴N]-Leu (M+0)). We have now revised the figure legends.

Finally, given the omnipresent through this study claim of PROX1 “driving metabolic plasticity” in cancer cells the authors should either tune down this claim or provide biochemical evidence of the plasticity in form of metabolic flux with labeled Glu in cancer cells devoid of Prox1/glucose starvation/under metformin treatment.

Response: We thank the reviewer for the comments. As suggested, we have tuned down the claim in the revised manuscript.

Reviewer #2 (Remarks to the Author):

In this revised manuscript by Wang and colleagues, the authors have addressed all of this reviewer's concerns. New data and new controls for PROX1 IHC in lung tumors and use of modern AMPK agonists like MK8722 enhance and clarify the authors' findings. Manuscript has been heavily edited in response to other reviewers, and should be acceptable in its current form for publication in Nature Communications in this reviewer's opinion.

Response: We greatly appreciate the reviewer for his/her positive and enthusiastic comments.

Reviewer #3 (Remarks to the Author):

Report for revised manuscript by Wang et al., Prospero-related homeobox protein 1 orchestrates tumor metabolic plasticity by integrating AMPK signaling and BCAA catabolism for Nat Commun.

The authors have addressed most of the major comments that were raised at the initial submission, and they included a considerable amount of new data addressing the comments in this revised version of their manuscript. I believe that the following comments were not properly or fully addressed in the rebuttal and would benefit from additional clarifications and adjustments. Some of these points may be addressed and explained in the main text.

Response: We are grateful to this reviewer for taking the time to read our manuscript and give us very helpful comments.

Point #1: I agree that using the various liver cancer cell lines as well as the KRAS driven LKB1 deficient lung cancer models are all interesting (and available) models, but I still don't understand why the authors go back and forth using different cell models for different experiments. While a large part of the data was carried out in the Huh7 cell line, it is known that different cancer cells display different basal metabolic profiles. Given the conclusion of this study, this should be an incentive to show whether conclusions drawn from this study may apply to cancer cells with a specific metabolic profile or can be generalized. Validation of the main findings in a second cell model should not be omitted simply because it would "double" all the experiments. The authors do not need to redo all the experiments in all the cell lines, but they should show some consistency and validate main findings in other models (for eg in figure 4, qPCR and immunoblots could be validated in additional models, no need to redo the RNA-seq). Whenever this is not possible, the use of one specific model for a specific experiment should be justified in the text.

Response: We thank the reviewer for the comments. As recommended, we have performed qPCR and immunoblots to test the expression of several genes involved in BCAA metabolism, and found a similar result in the HepG2 cells. These results have been shown in revised supplemental Fig.4d and 5f and also below for a quick reference.

Point #9:

- I am still confused with the GSEA plot for the PROX1 KO (4b and S4b). The figure legend of 4b is incomplete: does not indicate the system used (liver KO, cell models?). The GSEA plot is opposite than what is shown in all other panels (ie. BCAA metabolic gene expression increases upon PROX1 KO). Please include the full GSEA graphs (with labeling of samples to avoid confusion). How do the authors explain the negative NES of the GSEA plot in 4c? The table provided in the rebuttal tend to show that order of samples in 4c might be mislabeled.

Response: We have improved the figure legend, “Prox1-cKO” indicates Prox1 liver specific knockout mice (Alb-Cre; *Prox1*^{fl/fl}).

The GSEA plot was performed using WT group versus Prox1-cKO group, and it is indeed negative NES as indicated which means this Gene Set is upregulated in the KO group. If we used the Prox1-cKO group versus WT group to perform GSEA, it will be a positive NES of the GSEA plot which also means upregulation in the KO group. We have shown the two original full GSEA Results Summary below for your reference.

WT group versus Prox1-cKO group

Table: GSEA Results Summary

Dataset	WT-KO - mouse_collapsed_to_symbols.WT-KO - mouse.cls#WT-mouse_versus_KO-mouse
Phenotype	WT-KO - mouse.cls#WT-mouse_versus_KO-mouse
Upregulated in class	KO-mouse
GeneSet	KEGG_VALINE_LEUCINE_AND_ISOLEUCINE_DEGRADATION
Enrichment Score (ES)	-0.5968905
Normalized Enrichment Score (NES)	-2.5061264
Nominal p-value	0.0
FDR q-value	0.0
FWER p-Value	0.0

Prox1-cKO group versus WT group

Table: GSEA Results Summary

Dataset	WT-KO - mouse_collapsed_to_symbols.WT-KO - mouse.cls#KO-mouse_versus_WT-mouse.WT-KO - mouse.cls#KO-mouse_versus_WT-mouse_repos
Phenotype	WT-KO - mouse.cls#KO-mouse_versus_WT-mouse_repos
Upregulated in class	KO
GeneSet	KEGG_VALINE_LEUCINE_AND_ISOLEUCINE_DEGRADATION
Enrichment Score (ES)	0.5974971
Normalized Enrichment Score (NES)	2.4402003
Nominal p-value	0.0
FDR q-value	0.0
FWER p-Value	0.0

- Evidence for the regulation of the BCAA metabolic genes by AMPK has now been

shown by the authors (S4d) but should be included in the main text. This is an important data showing the role of AMPK regulation in this BCAA effect (via prox1 or independently).

Response: We thank the reviewer for the comments. As suggested, we have now moved this figure into the main Figure 4.

- The reasons and consequences of different trend observed for BCAT1 should be discussed.

Response: We thank the reviewer for the comments. Loss of PROX1, on one hand, releases the genes implicated in BCAA degradation. On the other hand, PROX1 depletion results in downregulation of certain genes like BCAT1-2 which leads to the BCAA synthesis suppression. Absolutely, we agree with the reviewer's comments that PROX1 is acting to increase intracellular BCAA levels due to inhibited BCAA catabolism activity but also increased BCAA synthesis through upregulating the expression of BCAT1-2. We have discussed this in Discussion section, please see line 598-605.

Point #13: I believe that the figure presented in the rebuttal (of tumor xenograft model with control, WT, S79A and S79E with or without metformin treatment) should be included in the main text as it supports the observations about the role of PROX1 with metformin and it is more complete than Figure S8n.

Response: We appreciate the reviewer's insightful comment. As suggested, we have now moved this figure into the main Figure 6.

REVIEWERS' COMMENTS

Reviewer #1 (Remarks to the Author):

The authors addressed my major concerns regarding the experimental work. As a suggestion, it would be great if the text of the manuscript was edited by the professional to improve the comprehension (grammar and use of specific terms).

Reviewer #3 (Remarks to the Author):

Revised manuscript by Wang et al., now entitled: "AMPK phosphorylates transcriptional repressor PROX1 to activate Branched amino acid degradation in cancer"

The authors have appropriately addressed the major concerns that were raised during the initial and rebuttal submissions. Although the manuscript is still heavily loaded with data, I believe that the authors have made a considerable effort to better explain and organize the results and not overdraw conclusions.

Minor comment:

At line 650, the sentence "In keeping with this role, higher expression level of PROX1 in HCC shows therapeutic sensitive to metformin.", the word "sensitive" should be replaced by "sensitivity"

REVIEWERS' COMMENTS

Reviewer #1 (Remarks to the Author):

The authors addressed my major concerns regarding the experimental work. As a suggestion, it would be great if the text of the manuscript was edited by the professional to improve the comprehension (grammar and use of specific terms).

Response: We are grateful to this reviewer for taking the time to read our manuscript and give us very helpful comments. As suggested, we have revised the manuscript by Nature Research Editing Service.

Reviewer #3 (Remarks to the Author):

Revised manuscript by Wang et al., now entitled: "AMPK phosphorylates transcriptional repressor PROX1 to activate Branched amino acid degradation in cancer"

The authors have appropriately addressed the major concerns that were raised during the initial and rebuttal submissions. Although the manuscript is still heavily loaded with data, I believe that the authors have made a considerable effort to better explain and organize the results and not overdraw conclusions.

Response: We are grateful to this reviewer for taking the time to read our manuscript and give us very helpful comments.

Minor comment:

At line 650, the sentence "In keeping with this role, higher expression level of PROX1 in HCC shows therapeutic sensitive to metformin.", the word "sensitive" should be replaced by "sensitivity"

Response: We have corrected this word.